# A modular circuit coordinates the diversification of courtship strategies

Rory T. Coleman[1,2], Ianessa Morantte[1,2], Gabriel T. Koreman[1,2], Megan L. Cheng[1,2], Yun Ding[3] & Vanessa Ruta[1,2✉]

Mate recognition systems evolve rapidly to reinforce the reproductive boundaries between species, but the underlying neural mechanisms remain enigmatic. Here we leveraged the rapid coevolution of female pheromone production and male pheromone perception in *Drosophila*[1,2] to gain insight into how the architecture of mate recognition circuits facilitates their diversification. While in some *Drosophila* species females produce unique pheromones that act to arouse their conspecific males, the pheromones of most species are sexually monomorphic such that females possess no distinguishing chemosensory signatures that males can use for mate recognition[3]. We show that *Drosophila yakuba* males evolved the ability to use a sexually monomorphic pheromone, 7-tricosene, as an excitatory cue to promote courtship. By comparing key nodes in the pheromone circuits across multiple *Drosophila* species, we reveal that this sensory innovation arises from coordinated peripheral and central circuit adaptations: a distinct subpopulation of sensory neurons has acquired sensitivity to 7-tricosene and, in turn, selectively signals to a distinct subset of P1 neurons in the central brain to trigger courtship. Such a modular circuit organization, in which different sensory inputs can independently couple to parallel courtship control nodes, may facilitate the evolution of mate recognition systems by allowing novel sensory modalities to become linked to male arousal. Together, our findings suggest how peripheral and central circuit adaptations can be flexibly coordinated to underlie the rapid evolution of mate recognition strategies across species.

Sensory evolution has been proposed to fuel behavioural diversification across species[4], allowing animals to capture and perceive distinct features of their environment. Although the rapid expansion and diversification of sensory receptors is thought to represent a potent force in the evolution of behaviour[4–6], changes to central circuit processing must also have a role[7,8], by acting in concert with a diversifying periphery to translate novel sensory inputs into coherent behavioural responses. Yet, how peripheral and central circuit adaptations are coordinated to give rise to the emergence of adaptive behavioural traits remains unclear.

Reproductive behaviours provide a powerful inroad to explore how evolution acts at multiple levels within a sensory-processing circuit. As species diverge, the sensory signals that females convey to males rapidly diversify under strong selection to prevent interspecies mating[9]. In turn, the sensory pathways that males use to detect and interpret female cues must coevolve. Indeed, changes in mate preference between closely related species are thought often to rely on reciprocal switches in the behavioural valence of mating signals[10]—conspecific cues must be made arousing whereas heterospecific cues made aversive—a process probably relying on concurrent changes in peripheral detection and central circuit processing.

Across the *Drosophila* genus female pheromones have repeatedly diversified, both in their chemical composition and the apparent logic of how they control mate choice[3,11–15] (Fig. 1a), offering an opportunity to examine how the rapid diversification of mate recognition systems is coordinated. In some species, females produce unique cuticular hydrocarbons thought to mediate mate recognition by triggering a conspecific male's arousal. *Drosophila melanogaster* females, for example, produce 7,11-heptacosadiene (7,11-HD), a pheromone that differs from the chemical cues carried by conspecific males and heterospecific females that cohabitate within the same environments[11]. Thus, 7,11-HD signals both sex and species identity, enabling it to serve as a potent excitatory cue to promote courtship in *D. melanogaster* males. Sexually instructive pheromones such as 7,11-HD, however, are rare as the females of most *Drosophila* species carry no chemicals that uniquely distinguish them from their conspecific males[3] (Fig. 1a). For instance, *Drosophila yakuba* and *Drosophila simulans*, close relatives believed to have evolved in geographic separation[16], have independently lost the biosynthetic enzymes necessary to produce female-specific pheromones[15]. Consequently, females of both species produce 7-tricosene (7-T) (Fig. 1a), the same cuticular compound as their males.

[1]Laboratory of Neurophysiology and Behavior, The Rockefeller University, New York, NY, USA. [2]Howard Hughes Medical Institute, New York, NY, USA. [3]Department of Biology, University of Pennsylvania, Philadelphia, PA, USA. ✉e-mail: ruta@rockefeller.edu

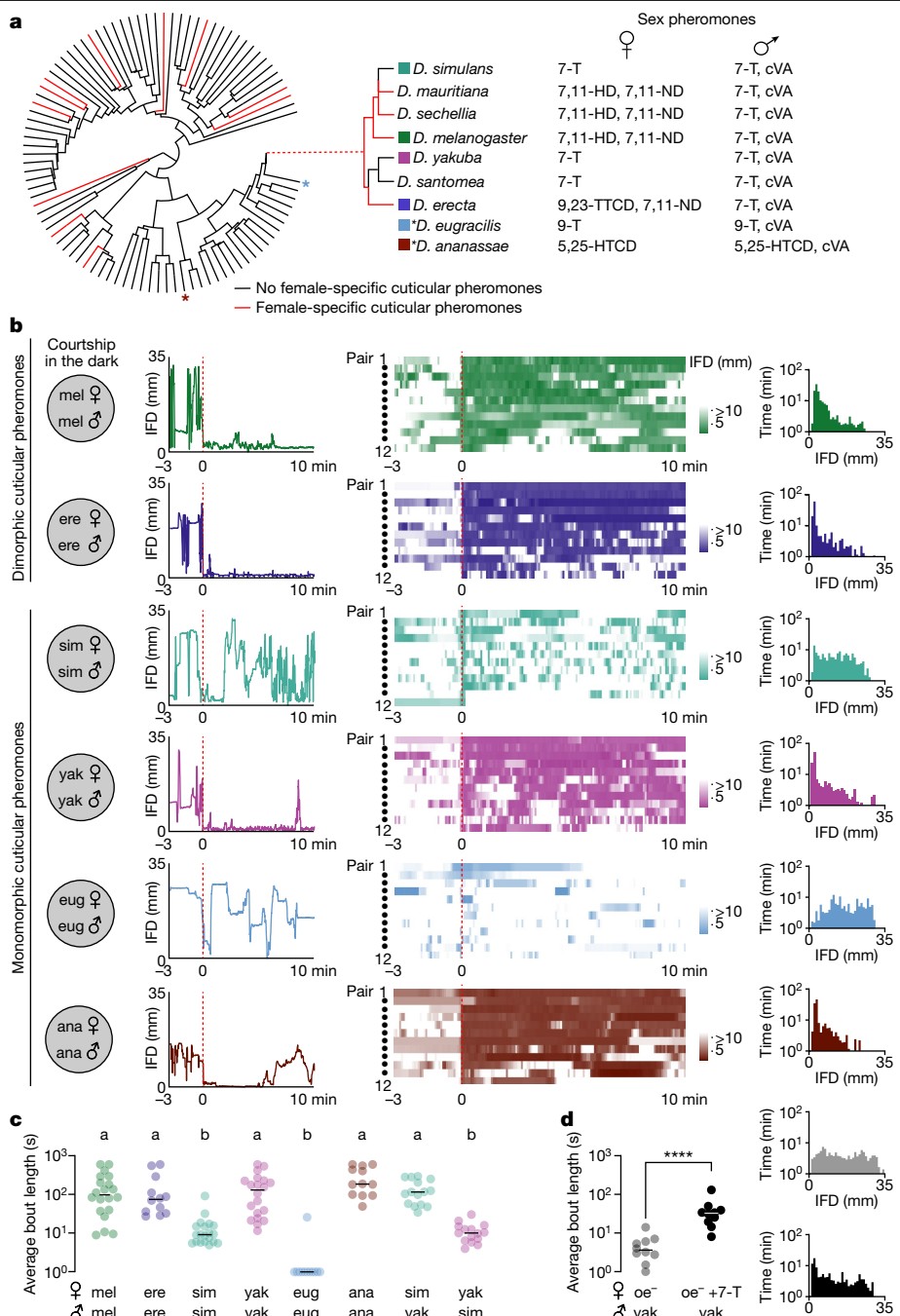

**Fig. 1 | Sexually ambiguous pheromones do not preclude courtship in the dark. a**, Phylogeny of 99 *Drosophila* species for which cuticular pheromones have been characterized (left) and primary sex pheromones of select species (right)[3,11]. 7-T, 7-tricosene; cVA, *cis*-vaccenyl acetate; 5,25-HTCD, 5,25-hentriacontadiene[61]; 7,11-ND, 7,11-nonacosadiene; 9-T, 9-tricosene (M. Khallaf, personal communication); 9,23-TTCD, 9,23-tritriacosadiene. **b**, Courtship as captured by the inter-fly distance (IFD) between a *D. melanogaster* (mel), *D. erecta* (ere), *D. simulans* (sim), *D. yakuba* (yak), *D. eugracilis* (eug) or *D. ananassae* (ana) male with a conspecific female in the dark. Left, IFD traces over time for a single representative pair as courtship proceeds. Middle, heatmaps for 12 pairs, aligned to courtship initiation for all species except *D. eugracilis*, for which videos were aligned to the time of first interaction. Red, dotted lines indicate the time of first interaction (*D. eugracilis*) or courtship initiation (all other species). Right, histograms of time as a function of IFD for the same 12 courting pairs. **c**, Average courtship bout length in the dark following courtship initiation for males paired with a conspecific female (mel/sim/yak *n* = 20; ere/eug/ana *n* = 12), *D. yakuba* males paired with a *D. simulans* female (*n* = 15) or *D. simulans* males paired with a *D. yakuba* female (*n* = 15). **d**, Left, average courtship bout length following courtship initiation for *D. yakuba* males paired with oe⁻ females mock perfumed (grey) or perfumed with the *D. yakuba* pheromone 7-T (black) in the dark (*n* = 10) and (right) histograms (as in **b**). Data points represent individual males; bars are median. Statistics: Kruskal–Wallis test (**c**) or unpaired Mann–Whitney (**d**). Letters denote statistically different groups (*P* < 0.05). ****P* < 0.0001. Diagram in **a** adapted from ref. 3, Springer Nature Limited.

Despite the convergent use of the same sexually monomorphic pheromone in *D. simulans* and *D. yakuba*, we find that males of these species employ a distinct logic in the chemosensory control of courtship, underscoring the evolutionary flexibility of mate recognition systems. Although *D. simulans* males appear insensitive to 7-T and use vision or other sensory cues to become aroused[13], *D. yakuba* males use this same

pheromone as an excitatory cue to promote courtship. This sensory innovation confers *D. yakuba* males with the ability to become aroused and faithfully pursue their conspecific females in the dark, thereby expanding the range of potential sensory environments available for mating. The distinct behavioural sensitivity of *D. yakuba* males to 7-T arises from coordinated peripheral and central circuit adaptations: 7-T activates a subset of sensory neurons, which selectively signal to one of two molecularly defined subsets of P1 neurons that control a male's sexual arousal[17–26]. A similar sensory specialization is also apparent in the P1 subpopulations of *D. melanogaster* males, underscoring how the modular organization of this circuit node may provide a facile evolutionary substrate for the rapid diversification of pheromone preferences and courtship strategies.

## Divergent mate recognition strategies

In many *Drosophila* species, the perception of a moving fly-sized target serves as a potent courtship-promoting cue, suggesting that vision can act redundantly with female pheromones to arouse males[12,24,25,27]. To compare the role of pheromonal signalling in mate recognition across species, we examined the courtship dynamics of single male–female pairs in the dark, where identification of a conspecific female becomes more reliant on chemical cues[28]. We focused our comparison on two species that produce sexually dimorphic pheromones—*D. melanogaster* and *Drosophila erecta*—and several in which females lack any chemical signatures that differentiate them from their males—*D. simulans* and *D. yakuba*, two close relatives within the *D. melanogaster* species subgroup; *Drosophila eugracilis*, a close outgroup species; and *Drosophila ananassae*, a more distant relative (Fig. 1a). Given that a male's pursuit of a female represents a conserved hallmark of courtship across species[2,29], we used inter-fly distance (IFD) as a behavioural readout that is independent of any potential variation in courtship motor displays (Fig. 1b and Extended Data Fig. 1a,b).

*D. melanogaster* males court proficiently in the dark[28,30–33] due to their ability to use 7,11-HD, the pheromone carried by their conspecific females, to sustain their arousal in the absence of vision[28,31,32]. Indeed, once *D. melanogaster* males initiated courtship, they closely tracked a female (IFD < 8 mm) for extended bouts (Fig. 1b,c, Extended Data Fig. 1a,c and Supplementary Video 1). *D. erecta* males displayed similarly faithful pursuit, suggesting that the unique pheromonal cues carried by their conspecific females have an instructive role comparable to that of 7,11-HD (Fig. 1b,c and Extended Data Fig. 1a). By contrast, the courtship dynamics of monomorphic species were far more variable in the dark. *D. eugracilis* males rarely performed any pursuit, and proximity to the female was only transient and incidental (Fig. 1b,c and Extended Data Fig. 1a). *D. simulans* males displayed brief bouts of courtship, but these frequently broke off as soon as the female walked away (Fig. 1b,c, Extended Data Fig. 1a,b,c and Supplementary Video 2). Males of both these species therefore appear unable to initiate or sustain courtship pursuit in the absence of vision. By contrast, *D. yakuba* and *D. ananassae* males engaged in extended periods of courtship, resembling the persistent tracking displayed by species with sexually dimorphic pheromones (Fig. 1b,c, Extended Data Fig. 1a,b,c and Supplementary Video 3). Males of some monomorphic species thus seem to have evolved distinct sensory strategies to pursue conspecifics in the absence of visual feedback or a female-specific pheromone, pointing to species-specific variation in the sensory signals used to guide mating decisions[34].

In *D. melanogaster* males, the excitatory effect of 7,11-HD is thought to compensate for the absence of vision[28,31,32]. The persistent courtship pursuit exhibited by *D. yakuba* males in the dark suggests that they may rely on female pheromones to become aroused, despite the sexually monomorphic nature of 7-T. Indeed, in the dark, *D. yakuba* males were largely indifferent to *D. melanogaster* females that lack oenocytes (oenocyte-less (oe⁻)) and produce no cuticular pheromones[12]. Perfuming oe⁻ females with 7-T, however, rendered them attractive

(Fig. 1d and Extended Data Fig. 1d). The detection of 7-T is therefore sufficient to arouse *D. yakuba* males, an adaptation that could expand the diurnal periods accessible for mating, consistent with their preference to court before dawn[30]. Aligned with the excitatory role of 7-T, *D. yakuba* males courted *D. simulans* females as vigorously as their conspecifics (Fig. 1c). 7-T thus seems to have divergent roles to hone mate selection—serving as an inhibitory cue to prevent heterospecific courtship in *D. melanogaster* males[35], a neutral signal that *D. simulans* males seem indifferent to[13], and an excitatory cue that promotes courtship in *D. yakuba* males—reflecting multiple switches in the behavioural valence of a single pheromone across closely related species.

## Divergent P1 neuron tuning across species

P1 neurons serve as a central node within the male courtship circuit that integrates from multiple sensory pathways to encode the suitability of a prospective mate and trigger persistent courtship displays[17–26]. Given the distinct reliance of different *Drosophila* species on pheromone signals to guide their courtship (Fig. 1b), we generated neurogenetic tools to compare how the chemosensory tuning of the P1 population has accordingly diversified. Anatomic labelling revealed that across species, P1 neurons display rich projections in the lateral protocerebral complex (LPC), a sexually dimorphic neuropil shown to receive inputs from pheromone-processing pathways and extend outputs to descending neurons that drive the component behaviours of courtship[19,20,36] (Fig. 2a). Consistent with a conserved role in mate recognition, optogenetic activation of P1 neurons in *D. erecta* and *D. yakuba* males was sufficient to trigger persistent pursuit of otherwise unattractive heterospecific females (Extended Data Fig. 2a,b), replicating the courtship-promoting function of these neurons in *D. melanogaster* and *D. simulans*[13,17,19,22,23,25,26]. P1 neurons thus represent a common circuit node regulating courtship arousal across species.

We compared the pheromonal responses of P1 neurons across species by performing functional calcium imaging of their projections within the LPC, as tethered males walked on an air-supported ball and were offered an array of female targets, replicating the sensory assessment males perform as they tap females with their forelegs to sample their cuticular pheromones (Fig. 2b). In *D. melanogaster*[13,19] and *D. erecta* males, P1 neurons were robustly activated each time a male tapped a conspecific female, with minimal excitation elicited by heterospecific females (Fig. 2b). Such selective chemosensory tuning supports the notion that the unique female pheromones of sexually dimorphic species serve as instructive cues for mate recognition, exciting the P1 neurons to promote courtship. By contrast, the P1 neurons of *D. simulans* males were largely unresponsive to any female targets (Fig. 2b), concordant with evidence that in this species, pheromone pathways serve to inhibit rather than promote courtship[12,13]. The P1 neurons of *D. yakuba* males, however, rather than displaying the attenuated chemosensory responses of *D. simulans*, instead were strongly activated by the taste of both *D. yakuba* and *D. simulans* females (Fig. 2b), aligned with behavioural data showing that 7-T serves as an excitatory cue to arouse males (Fig. 1d). Thus, although P1 neurons have a conserved role in promoting courtship across species, they display divergent pheromonal tuning, highlighting the intrinsic evolutionarily flexibility of sensory circuits that control a male's arousal and recognition of an appropriate mate.

## Diversification of peripheral pathways

*D. yakuba* males seem to have evolved the ability to use the sexually ambiguous chemical 7-T as a courtship-promoting cue via the diversification of pheromone circuits that impinge onto the P1 neurons. To shed light on how alterations in peripheral pheromone detection contribute to this distinct chemosensory strategy, we compared the sensory neurons in the male foreleg, a heterogenous population marked

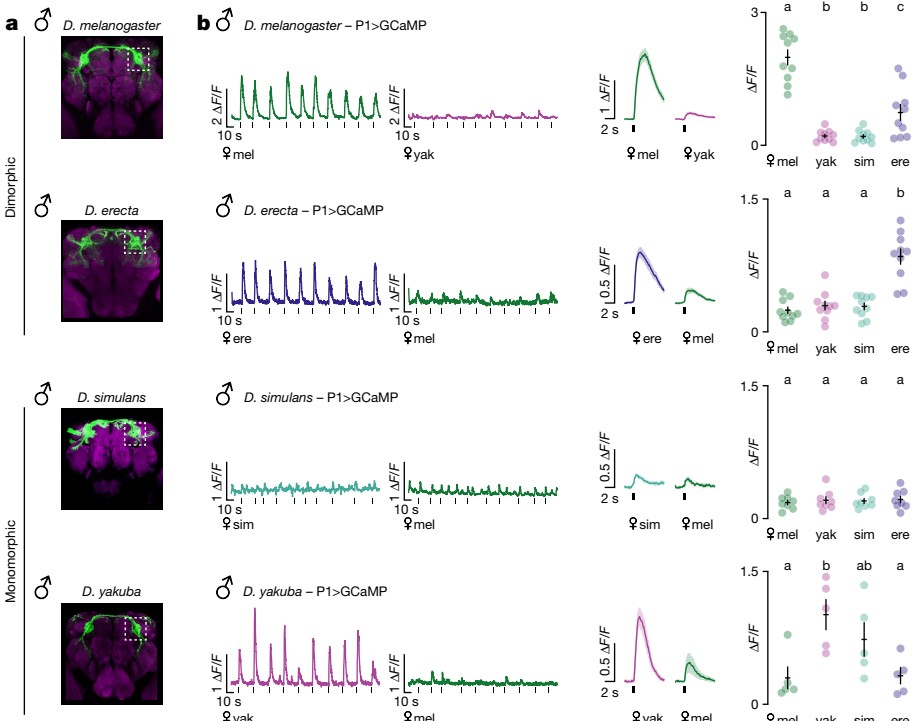

**Fig. 2 | P1 neurons of *D. yakuba* males share conspecific tuning pattern of dimorphic species. a**, P1 neurons labelled by 71G01>CD8::GFP (*D. melanogaster*, *D. erecta*), 71G01>GCaMP6s (*D. simulans*) or SplitP1>CD8::GFP (*D. yakuba*; Methods) expression, stained for GFP (green) and neuropil counterstain (magenta). Images were masked to remove glial fluorescence from ie1 marker and non-P1-specific labelling for clarity. Imaging location used for experiments in **b** is indicated by white, dotted box. **b**, Left, representative traces of P1 responses in the LPC in males of each species evoked in response to the taste of a conspecific or heterospecific female (black ticks indicate time of foreleg taps). Middle, averaged tap-evoked functional responses ($\Delta F/F_0$, black ticks) of the P1 neurons across all males. Right, average peak response ($\Delta F/F_0$) for each male evoked by a given female target (sample sizes: mel $n = 10$; ere $n = 10$; sim $n = 8$; yak $n = 5$). Data points represent individual males; error bars are mean ± s.e.m. Statistics: analysis of variance (ANOVA) with Tukey's post hoc. Letters denote statistically different groups ($P < 0.05$).

by expression of the DEG/ENaC channel Ppk23 (refs. 13,32,37–39). In *D. melanogaster*, Ppk23[+] neurons adopt a paired organization within each sensory bristle[20,31,32,37–39], in which one neuron co-expresses the DEG/ENaC channel Ppk25 and is tuned to 7,11-HD to promote pursuit of conspecific females, whereas its Ppk25[−] partner is responsive to heterospecific pheromones, including 7-T, to suppress interspecies courtship (Fig. 3a,h). To explore the role that Ppk23-mediated pheromonal signalling plays in mate recognition in *D. yakuba*, we used CRISPR–Cas9 genome editing to generate males mutant for this receptor (Extended Data Fig. 3a). We found that these mutants lost their characteristic aversion to heterospecific females, supporting that, as in *D. simulans*[13], this receptor mediates the detection of inhibitory pheromones to curb inappropriate visual pursuit (Fig. 3b,h). *D. yakuba ppk23* mutant males continued to vigorously pursue their conspecific females in the light, probably due to the redundant role that conspecific pheromones and visual cues have in promoting courtship (Fig. 3b and Extended Data Fig. 3c), but were unable to sustain courtship in the dark, yielding the same saltatory dynamics displayed by *D. melanogaster ppk23* mutant males in the absence of visual feedback (Fig. 3c,d and Extended Data Fig. 3b,d). The total time that *ppk23* mutant males spent courting was not significantly reduced, however, probably because they were often able to rapidly re-encounter their female (Extended Data Fig. 3b,d). Ppk23 signalling in *D. yakuba* thus seems required both to promote courtship of conspecific females and to inhibit pursuit of heterospecific targets, suggesting that, as in *D. melanogaster*[32,37,38], Ppk23 receptors mark a heterogeneous neuronal population that plays opposing roles in honing mate recognition.

Imaging the aggregate activity of Ppk23[+] foreleg sensory afferents (Fig. 3a) in *D. yakuba* males revealed that they were robustly activated by all target flies, with equivalent responses evoked by the taste of conspecific and heterospecific females and males (Fig. 3e). Such broad tuning contrasts with the selective responses to 7,11-HD-carrying females observed in both *D. melanogaster* (Fig. 3e) and *D. simulans* males[13], to alternatively promote or deter courtship to *D. melanogaster* females (Fig. 3h). Ppk23[+] sensory responses in *D. yakuba* males were chemosensory in origin, as oe[−] females lacking pheromones evoked minimal activity but robust responses could be restored by perfuming these females with either 7,11-HD or 7-T (Extended Data Fig. 3f). Moreover, pheromone responses were largely abolished in *D. yakuba ppk23* mutant males (Fig. 3e), substantiating that Ppk23-mediated signalling plays a conserved and essential role in pheromone detection across species[13,32,37,38].

To determine whether the broad chemosensory tuning apparent at the level of the Ppk23[+] population reflects the activity of heterogenous Ppk25[+] and Ppk25[−] subsets (Fig. 3a), we generated a *ppk25* mutant (Extended Data Fig. 4a). Recording from foreleg sensory afferents in *ppk25* mutant males revealed that responses to *D. yakuba*, *D. simulans* females and males were selectively attenuated, whereas responses evoked by other heterospecific targets remained intact (Fig. 3e and Extended Data Fig. 4c), suggesting that Ppk25-mediated signalling underlies 7-T detection to regulate a male's sexual arousal. Consistent with this, *D. yakuba ppk25* mutants displayed diminished courtship towards conspecifics in the dark (Fig. 3f and Extended Data Fig. 4b), phenocopying the disrupted pursuit observed in *ppk23* mutants (Fig. 3d). *D. yakuba ppk25* mutants nevertheless remained averse to courting *D. melanogaster* females (Fig. 3f), substantiating that *ppk25* is required for detection of 7-T without impairing the recognition of heterospecific pheromones such as 7,11-HD that suppress inappropriate courtship.

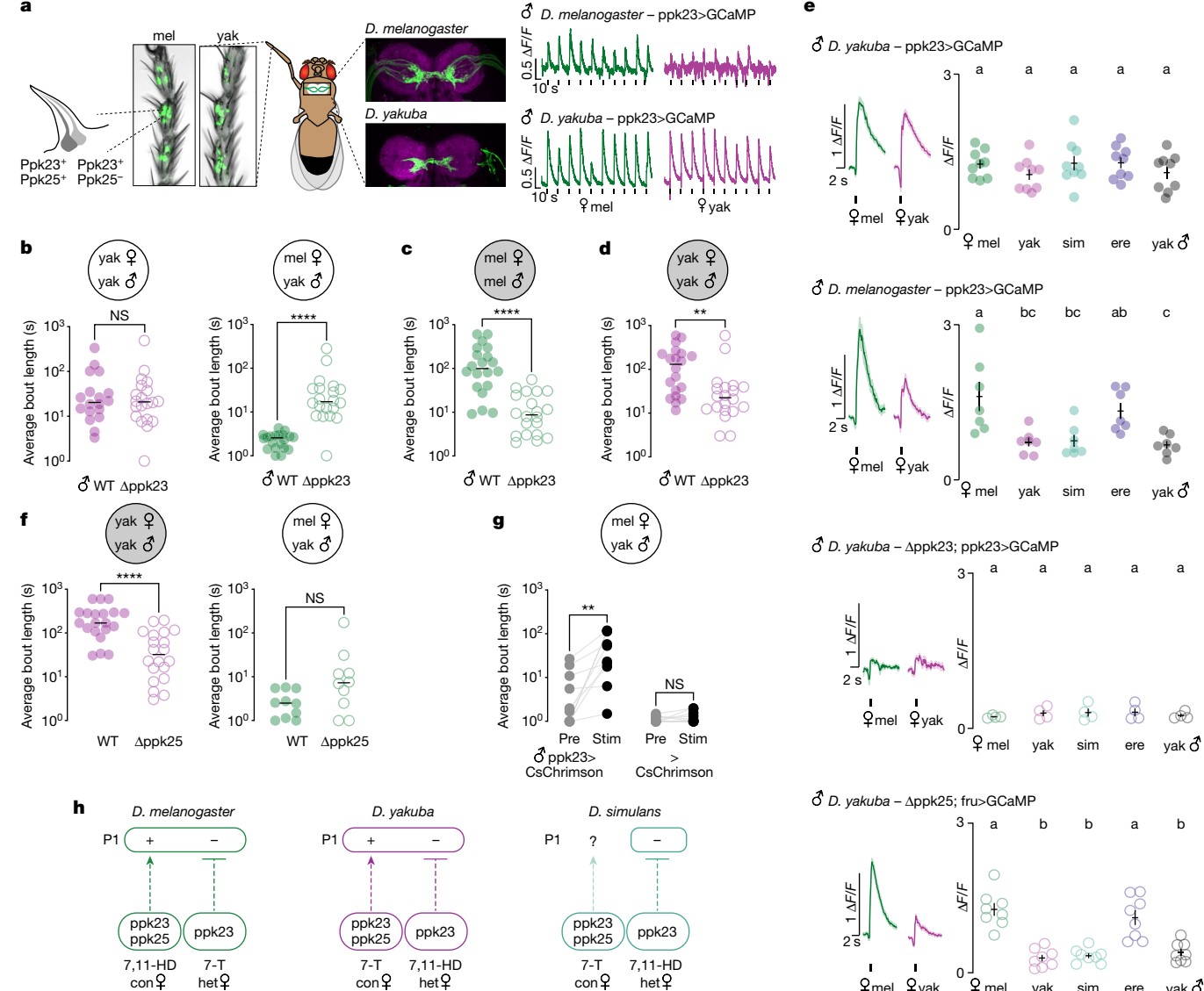

**Fig. 3 | Altered pheromone sensitivity in *D. yakuba* sensory neurons. a**, Left, Ppk23$^+$ sensory neurons in *D. melanogaster* and *D. yakuba* male foreleg tarsal segments, marked by CD8::GFP (green), distal end up. Middle, Ppk23$^+$ sensory afferents in the first thoracic segment of the ventral nerve cord expressing CD8::GFP (green) with neuropil counterstain (magenta), anterior side up. Right, functional responses of Ppk23$^+$ afferents evoked by indicated females (black ticks indicate foreleg taps). **b**, Average courtship bout length in the 10 min following courtship initiation for wild-type males paired with conspecific female (*n* = 18); *D. melanogaster* female (*n* = 20) or *ppk23* mutant (*n* = 20 each) in the light. **c**,**d**, Average courtship bout length of *ppk23* mutant *D. melanogaster* (*n* = 20 each) (**c**) or *D. yakuba* (*n* = 20 each) (**d**) males paired with conspecific females in the dark. **e**, Left, average functional responses ($\Delta F/F_0$) recorded from foreleg sensory afferents aligned to time of a tap and (right) average peak response ($\Delta F/F_0$) per male for a given female target. (sample sizes: yak ppk23>GCaMP *n* = 9; mel ppk23>GCaMP *n* = 7; yak Δppk23;ppk23>GCaMP *n* = 4;

yak Δppk25;fru>GCaMP *n* = 8). **f**, Average courtship bout length of wild-type males paired with conspecific females in the dark (left) (*n* = 10 each) and *ppk25* mutant males paired with *D. melanogaster* females in the light (right) (*n* = 10 each). **g**, Average courtship bout length before (Pre) or during (Stim) periods of optogenetic stimulation of Ppk23>CsChrimson or >CsChrimson control animals paired with a *D. melanogaster* female (*n* = 10 each). **h**, Model summarizing the inferred changes in the sensitivity of sensory populations and effect on P1 neuron activity. *D. melanogaster* and *D. simulans* diagrams on the basis of previous reports[13,19,20,31,35,37,39]. For behavioural tests (**b**–**g**), points represent individual males; bars are median. For imaging (**e**), shading is mean ± s.e.m., points are individual males and error bars are mean ± s.e.m. Statistics: ANOVA with Tukey's post hoc (**e**), Mann–Whitney (**b**–**d**,**f**), Wilcoxon test (**g**). Letters denote statistically different groups (*P* < 0.05). **P* < 0.01, ****P* < 0.0001; NS, not significant. WT, wild type.

The sensory neurons that detect conspecific and heterospecific pheromones thus seem to have undergone a reciprocal switch in their chemical specificity across species. Although *D. melanogaster* and *D. yakuba* females carry distinct pheromones, males of both species detect their cognate conspecific chemical cues via the Ppk25$^+$ subset of sensory neurons[20,39], whereas the Ppk25$^-$ subset underlies the detection of heterospecific cues. This swap in the chemical tuning of peripheral sensory neuron subtypes suggests a potentially facile mechanism to alter the behavioural meaning of pheromones, whereby the ascending

circuits that promote or suppress courtship are conserved but their pheromone sensitivity is altered. In *D. simulans* males, *ppk25* has been reported to promote courtship[40], suggesting it may be similarly involved in 7-T detection. However, this peripheral sensitivity does not seem to be translated to P1 neuron excitation (Figs. 2b and 3h), consistent with the tepid courtship that males of this species display in the dark (Fig. 1b,c). The behavioural valence of pheromones therefore seems to depend not only on which subset of sensory neurons are activated, but also on how these peripheral signals are conveyed to the

P1 population. Indeed, using optogenetics to exogenously activate the Ppk23$^+$ sensory neurons and bypass pheromone detection revealed that although stimulation of this population suppresses courtship in *D. simulans*[13], it triggered both *D. melanogaster*[13,32,37] and *D. yakuba* males to court otherwise unattractive targets, including heterospecific females (Fig. 3g). The opposing behavioural valence of Ppk23$^+$ sensory neuron activation across species suggests that further neural adaptations underlie how these diversified peripheral signals are integrated by the P1 neurons to control a male's mating decisions.

## Sensory specialization of P1 subtypes

How might P1 neurons accommodate the flexible integration of different ascending pheromone pathways to underlie their species-specific tuning? The modular organization of sensory circuits has been proposed to facilitate their evolutionary diversification, as their segregated nature allows for the independent retuning of sensory inputs[7,41]. Notably, P1 neurons can be divided into two discrete subsets on the basis of their expression of the sexually dimorphic transcription factors Doublesex (Dsx) and Fruitless (Fru)[18,42]. Although all P1 neurons express Dsx and belong to the larger population of pC1 neurons, a subset also expresses Fru[18] (Extended Data Fig. 5a), raising the possibility that these genetically distinct P1 subsets may represent modular units that integrate from different sensory pathways. We therefore devised an intersectional genetic strategy using a P1 neuron-specific driver to label a Fru$^+$ (Fru∩P1) or Dsx$^+$ (Dsx∩P1) subset of the P1 population (Extended Data Fig. 5a). Each intersection marked a comparable number of neurons in *D. yakuba* and *D. melanogaster* males, with Fru∩P1 labelling only Fru$^+$ somata and Dsx∩P1 labelling a mixture of Fru$^+$ and Fru$^−$ neurons (Extended Data Fig. 5b–e). Although the morphology of the Fru∩P1 and Dsx∩P1 subpopulations was similar both within and across species, they displayed minor variation in their ventral and dorsal lateral projections (Fig. 4a,b and Extended Data Fig. 5b,c,f,g), positioning them to potentially integrate from distinct ascending inputs. Consistent with this, we observed that Dsx∩P1 neurons were strongly activated by the taste of conspecific females (Fig. 4b), which we confirmed using perfumed oe$^−$ females reflects their distinct chemosensory tuning to 7-T (Fig. 4f). Fru∩P1 neurons, by contrast, were unresponsive to all target flies (Fig. 4a). Despite these striking differences in pheromone tuning, optogenetic activation of either P1 subpopulation was sufficient to trigger courtship pursuit and singing towards heterospecific females (Fig. 4a,b, Extended Data Fig. 7a,b and Supplementary Videos 4 and 5). Genetically distinct and functionally specialized subsets of P1 neurons thus seem to have overlapping roles in triggering a male's sexual arousal to drive courtship.

The differential sensory tuning we observe suggests that only the Dsx$^+$/Fru$^−$ P1 subpopulation is excited by 7-T in *D. yakuba* males. To verify this, we recorded pheromone responses from the complete complement of Fru$^+$ or Dsx$^+$ neurons in the LPC where P1 neuron processes reside. We found that the pheromone tuning of these broader Fru$^+$ or Dsx$^+$ populations replicated the selective responses displayed by each P1 neuron subset, with only the Dsx$^+$ processes strongly excited by the taste of *D. yakuba* females. Responses to *D. yakuba* females were lost in *ppk23* and *ppk25* mutants (Fig. 4e and Extended Data Fig. 3g), confirming that they arise from 7-T detection via Ppk25$^+$/Ppk23$^+$ sensory neurons. The Dsx$^+$ P1 subpopulation thus seems to uniquely integrate excitatory pheromone signals from the Ppk25$^+$/Ppk23$^+$ sensory pathways, suggesting that they have a distinct role in translating conspecific pheromone detection to male arousal in *D. yakuba*. Consistent with this, constitutive silencing of the Dsx∩P1 neuronal subset attenuated male courtship in the dark (Fig. 4g), replicating the abortive pursuit of *ppk23* and *ppk25* mutants (Fig. 3d,f). Silencing of the Fru∩P1 neuronal subset, by contrast, had little impact on a male's courtship dynamics (Fig. 4g), aligned with the insensitivity of this population to pheromones. Although we cannot exclude the possibility that Fru∩P1 neurons

might function redundantly with the Dsx$^+$/Fru$^−$ neurons to promote courtship, our functional and behavioural data suggest that Fru$^+$ and Dsx$^+$ P1 subpopulations have distinct roles in sustaining courtship in the dark (Fig. 5b), where males become reliant on pheromonal feedback.

The molecular subdivision of P1 neurons by their differential expression of the Fru transcription factor represents a conserved feature of this circuit node (Extended Data Fig. 5b–e), suggesting that this modular organization may serve as a more general substrate for the evolution of mate recognition in *Drosophila*. Indeed, in *D. melanogaster* males, although activation of either Fru∩P1 and Dsx∩P1 neuron subsets was sufficient to promote courtship (Extended Data Fig. 6a,b and Extended Data Fig. 7c,d), they displayed distinct chemosensory tuning. Both subpopulations responded to their conspecific female pheromone (Extended Data Fig. 6a,b), whereas the Fru$^+$ subset was also excited by the taste of *D. erecta* females (Extended Data Fig. 6a), an expansion of pheromone tuning that may reflect sensitivity to 7,11-nonacosadiene, a minor pheromone shared between *D. melanogaster* and *D. erecta* females[11] (Fig. 1a). Consistent with the use of common excitatory female pheromones in these species, *D. melanogaster* males have been shown to court *D. erecta* females[12]. To further substantiate the functional modularity of P1 neuron subsets, we generated a fru-Gal80 allele in *D. melanogaster* males (Extended Data Fig. 6c), allowing us to selectively record from Dsx$^+$/Fru$^−$ processes in the LPC. This neuronal subset was strongly activated by *D. melanogaster* but not *D. erecta* females, indicating that the Fru$^+$ P1 neuron subset has evolved expanded pheromone sensitivity in this species. The functional specialization of molecularly defined P1 subtypes therefore seems to represent a shared feature of male courtship circuits across species, potentially facilitating the integration of new sensory pathways to allow for the emergence of species-specific mate preferences.

## Multisensory cues shape sex discrimination

Monomorphic pheromones are inherently ambiguous with respect to sex discrimination. Although *D. yakuba* males are less discerning than *D. melanogaster* males, they nevertheless show a significant preference for courting conspecific females over males, suggesting they rely on further sensory cues to aid in mate recognition (Extended Data Fig. 8a). Indeed, perfuming *D. yakuba* females with the male-specific volatile pheromone *cis*-vaccenyl acetate[43] (cVA) suppressed both the responses of P1 neurons and courtship pursuit towards these otherwise attractive female targets (Extended Data Fig. 8d,e). Mutating the canonical cVA receptor *Or67d* in *D. yakuba* males had no apparent impact on a male's propensity to court another male, suggesting that additional olfactory receptors detect this conserved pheromone[43] (Extended Data Fig. 8b,c). Beyond chemosensory cues, behavioural countersignalling—such as the agonistic wing flicks males perform when being pursued by other males[29,44]—also seems to support sex discrimination in *D. yakuba* (Extended Data Fig. 8f,g and Supplementary Video 6), as male-directed courtship was increased if one male was either deafened by surgically removing his aristae or rendered mute by removal of his wings. Thus, *D. yakuba* males seem to counterbalance their use of a sexually ambiguous excitatory pheromone with other male-specific inhibitory cues to hone their courtship towards an appropriate mate (Extended Data Fig. 8h).

## Discussion

In this study, we leveraged the rapid evolution of female pheromones across the *Drosophila* genus to explore how changes in male pheromone detection and preference are coordinated to generate divergent mating strategies. Our work points to at least two discrete sites of diversification within the male courtship circuitry that can modify the behavioural valence of pheromones to underlie species-specific mate recognition. First, the Ppk25$^+$ and Ppk25$^−$ sensory neurons within the

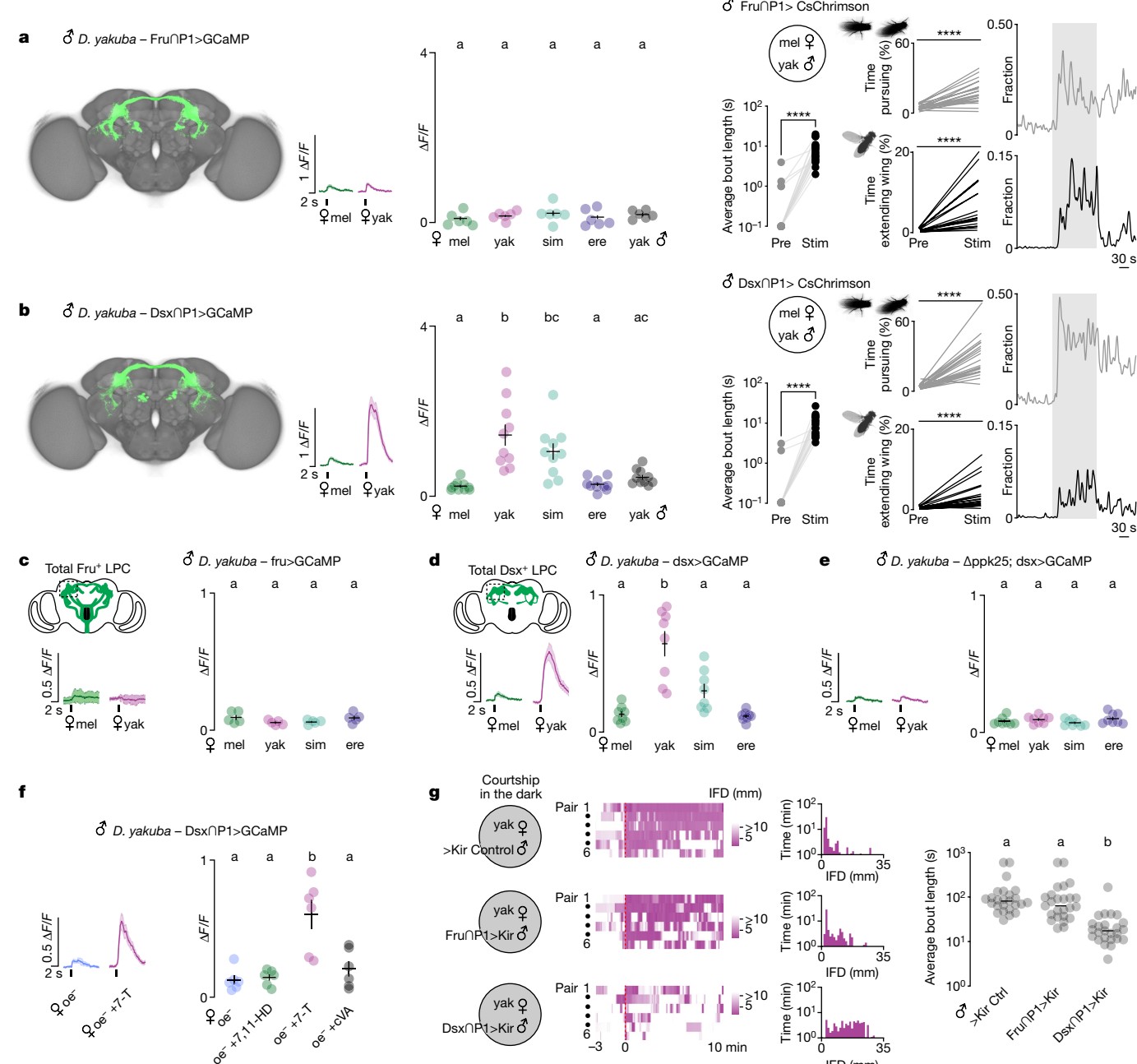

**Fig. 4 | Sensory specialization of Fru⁺ and Dsx⁺ P1 subpopulations in D. yakuba.** **a**,**b**, Left, template brain registrations of neurons labelled by intersection of Fru (**a**; Fru∩P1) or Dsx (**b**; Dsx∩P1) and the P1-driver 71G01 (Methods). Middle, averaged functional responses ($\Delta F/F_0$) aligned to tap of indicated female and average peak response ($\Delta F/F_0$) per male for a given female target. Right, average courtship bout length, total percentage time pursuing or total percentage time extending a unilateral wing towards a *D. melanogaster* female target before (Pre) or during (Stim) optogenetic stimulation of Fru∩P1>CsChrimson or Dsx∩P1>CsChrimson males and the fraction of flies engaging in these behaviours (sample sizes: Fru∩P1>GCaMP $n = 6$; Fru∩P1>CsChrimson $n = 20$; Dsx∩P1>GCaMP $n = 10$; Dsx∩P1>CsChrimson $n = 20$). **c**,**d**,**e**, Functional responses (as in **a**,**b**) of all Fru⁺ (**c**) or Dsx⁺ (**d**,**e**) neurons innervating the LPC in wild-type (**c**, $n = 5$; **d**, $n = 8$) or Δppk25 mutant (**e**, $n = 8$) males. **f**, Functional responses (as in **b**) of Dsx∩P1 to *D. melanogaster*

mock-perfumed oe⁻ (oe⁻), 7,11-HD-perfumed (oe⁻ +7,11-HD), 7-T-perfumed (oe⁻ +7-T) or cVa-perfumed (oe⁻ +cVA). **g**, Courtship in the dark as captured by IFD of a *D. yakuba* male towards a conspecific female with constitutively silenced P1 subsets (Fru∩P1>Kir (middle), Dsx∩P1>Kir (bottom) or genetic control (top; 71G01-DBD; UAS-Kir)) ($n = 6$). Left, heatmaps for six pairs, aligned to courtship initiation (red, dotted line). Middle, histograms of time as a function of IFD for the same six courting pairs. Right, average courtship bout length in the 10 min following courtship initiation of each genotype ($n = 25$ each). For functional imaging (**a**–**f**), shading represents mean ± s.e.m. Points are individual males; error bars are mean ± s.e.m. For behavioural tests (**a**,**b**,**e**), points are individual males and bars are median. Statistics: ANOVA with Tukey's post hoc (**a**–**f**), Wilcoxon test (**a**,**b**), Kruskal–Wallis test (**g**). Letters denote statistically different groups ($P < 0.05$). ****$P < 0.0001$.

male foreleg that detect conspecific and heterospecific pheromones seem to have undergone a reciprocal swap in their chemical tuning and, second, these peripheral signals are differentially conveyed to distinct subpopulations of P1 neurons, each sufficient to promote

courtship (Fig. 5a,b). Such a modular circuit organization, in which rapidly diversifying peripheral sensory populations can independently couple to different P1 courtship control nodes, may facilitate the evolution of mate recognition systems by enabling males to take advantage

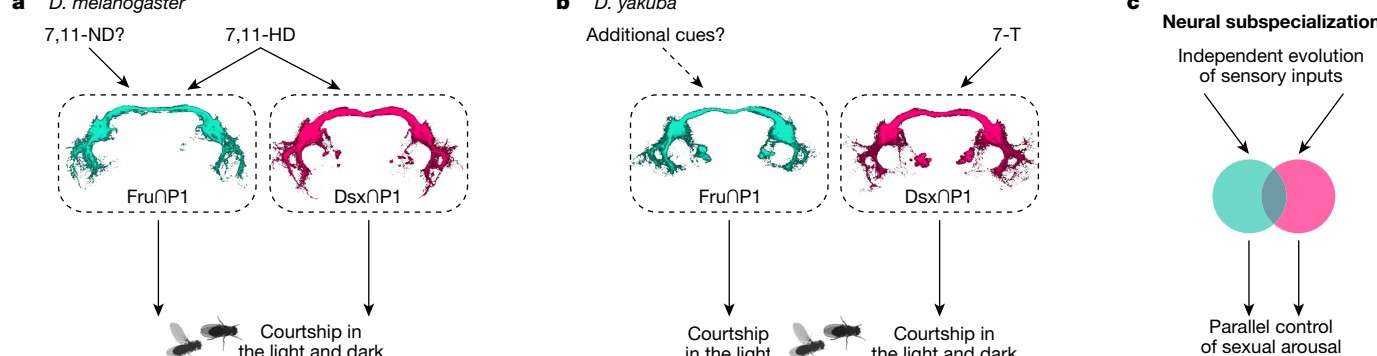

**Fig. 5 | Subspecialization of P1 neuron subtypes. a,b,** Diagrams summarizing proposed sensory specializations of Fru∩P1 and Dsx∩P1 in *D. melanogaster* (**a**) and *D. yakuba* (**b**). Pheromone inputs inferred from functional imaging and behavioural data in Fig. 4 and Extended Data Fig. 6 or previously reported[13,19,24,25] are indicated by solid arrows. Further hypothesized sensory inputs are indicated by dotted arrows. **c,** Model for the diversification of courtship behaviours by neural subspecialization. Independent retuning of sensory inputs to behaviourally redundant but molecularly distinct P1 subtypes may facilitate the rapid evolution of sensory signals that control a male's sexual arousal and courtship.

of novel sensory cues to promote courtship without compromising ancestral pathways for arousal (Fig. 5c). Through the coordination of these peripheral and central circuit modifications *D. yakuba* males have evolved the ability to use 7-T, a sexually ambiguous compound, as an aphrodisiac to promote courtship of their conspecific females.

Female-specific pheromones would seem advantageous to mate recognition by serving as instructive cues that assure males only become aroused when they encounter individuals of the appropriate sex and species[45]. Yet, sexually monomorphic pheromones predominate across the *Drosophila* genus[3]. For many species, the visual perception of a moving, fly-sized target seems innately arousing, potentially rendering excitatory pheromones unnecessary to promote courtship[12,24,25,27]. Nevertheless, as we reveal for *D. yakuba*, excitatory pheromones may confer additional robustness to courtship and expand the sensory environments in which it can occur. Indeed, by using 7-T to become aroused, *D. yakuba* males seem to overcome the strict requirement for vision, providing them with expanded diurnal periods for mating[30]. Notably, of the four monomorphic species we tested, both *D. yakuba* and *D. ananassae* robustly court in the dark, a sensory capacity shared by many species distributed throughout the genus[34]. The use of sexually monomorphic pheromones as courtship-promoting cues therefore seems to have arisen repeatedly, underscoring how the sensory pathways controlling male arousal have probably recurrently diversified to give rise to species with differential dependence on vision (Fig. 1b and Extended Data Fig. 9).

Despite that *D. simulans* and *D. yakuba* have independently converged on the same sexually monomorphic pheromone profiles, our analyses suggest that males of these species rely on divergent strategies for mate discrimination due to concurrent peripheral and central adaptations. Although *D. melanogaster*, *D. yakuba* and *D. simulans* males all maintain robust peripheral responses to 7,11-HD to alternately promote courtship to conspecifics or deter heterospecific pursuit, *D. yakuba* males have evolved a distinct sensory innovation—enhanced sensitivity to 7-T via a switch in the chemical tuning of the Ppk25+ subset of sensory neurons. Such rapid diversification of peripheral sensory neurons, which can be accommodated by either retuning of the receptors that detect pheromone or alterations in their pattern of expression within the male foreleg, must then be integrated with central circuits to allow novel pheromone sensitivity to regulate arousal and mate recognition. Indeed, we find that distinct subpopulations of P1 neurons differentially integrate from ascending pheromone pathways to contribute to the chemosensory control of courtship. Although the structural or functional changes underlying the divergent patterns of P1 neuron integration remain to be elucidated, the homologous ascending

pathways that transmit pheromone signals from the foreleg to the P1 neurons are anatomically identifiable across species[13,19] (Extended Data Fig. 10). In *D. melanogaster* males, Ppk25+ sensory neurons responsive to 7,11-HD relay these signals to both P1 neuron subtypes via the vAB3 ascending pathway to promote courtship towards a conspecific female[19,20]. The switch in pheromone sensitivity of Ppk25+ neurons in *D. yakuba* males suggests that vAB3 neurons may selectively convey 7-T-mediated excitation to the Dsx+/Fru– P1 neuron subpopulation to underlie their distinct tuning. Thus, subtle changes to an otherwise conserved circuit architecture could give rise to divergent P1 pheromone tuning, paralleling how the differential integration of pheromone pathways by P1 neurons underlies the opposing behavioural valence of 7,11-HD in *D. melanogaster* and *D. simulans* males[13]. Together, these observations support the possibility that P1 neurons serve as a site of repeated evolutionary tinkering, whereby reweighting of the sensory input pathways to distinct P1 subpopulations would allow for the rapid diversification in courtship strategies.

An intriguing feature of the functionally specialized P1 neuron subsets is their expression of the master regulatory transcription factors Fru and Dsx, which define the sexually dimorphic features of the nervous system[46–51]. Fru and Dsx act cell-autonomously to specify the anatomy and connectivity of neurons, presenting a powerful mechanism to restructure the functional architecture of mating circuits[18,52–55]. Yet, despite the striking differences apparent in the Fru+ and Dsx+ circuitry between males and females within a species, these pathways appear largely conserved across males of different species, including in the different P1 subtypes. Rather than changing patterns of Fru or Dsx expression, evolution may instead tinker with the anatomic or functional properties of neurons by altering the diverse transcriptional programs that Fru and Dsx direct. The gain or loss of Fru or Dsx binding sites in the regulatory regions of downstream target genes offers a plausible mechanism by which P1 neuron subtypes could independently diversify to generate species-specific patterns of sensory integration[56,57].

Our observations in *Drosophila* strengthen emerging evidence that modular neural circuit architectures, possibly arising from duplication and divergence, may represent a key evolutionary substrate for the diversification of complex behaviours[7,41,58,59]. Similar mechanisms have been proposed to underlie the elaboration of cerebellar nuclei[59] and the evolution of pallial–striatal circuits[58] in vertebrates. Duplication favours modular functional units which, as in gene duplication, relieves evolutionary constraints and allows initially redundant circuit structures to adopt new functions. Cell type duplication is not essential for neural subspecialization to arise. Rather, as our work suggests, all that is required is a population of neurons with shared behavioural roles

but divergent molecular programs that specify the cellular properties of these neurons and provide the genetic substrate on which evolution can act. Here, by taking advantage of the rapid coevolution of female pheromone production and male pheromone preferences in a model clade[7,60], we have begun to map discrete sites of functional divergence to molecularly defined neural subpopulations, revealing neuronal modularity as an important substrate for behavioural evolution.

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

## Methods

### Fly stocks and husbandry

Flies were maintained at 25 °C and 50–65% relative humidity under a 12 h light/12 h dark cycle. *D. melanogaster* stocks Canton-S (catalogue no. 64349), UAS-GCaMP6s (42746 and 42749), UAS-mCD8::GFP (5137), 71G01-Gal4 (39599), fru^LexA (66698) and UAS-CsChrimson-mVenus (55134 and 55136) were obtained from the Bloomington Stock Center. LexAop-SPA-T2A-SPA was generated in a previous study[19]. *D. yakuba* Ivory Coast (14021-0261.00), *D. erecta* (14021-0224.01) and *D. ananassae* (14024-0371.34) were obtained from the Cornell (formerly UCSD) Stock Center. *D. eugracilis* (SHL12) was obtained from the Kyoto Stock Center. *Drosophila suzukii* (WT3) (L. Zhao, Rockefeller); *D. melanogaster* SplitP1-Gal4 (D. Anderson, Caltech); *D. melanogaster* fru^Gal4 and fru^AD/fru^DBD (B. Dickson, Janelia Research Campus); *D. melanogaster* ppk23-Gal4 (K. Scott, UC Berkeley); *D. yakuba* fru^Gal4, fru^DBD and fru^AD inserts[62], 71G01-Gal4 (ref. 63), UAS-GCaMP6s, UAS-CsChrimson, UAS-Kir2.1 and attp1730 (D. Stern, Janelia Research Campus); *D. erecta* 71G01-Gal4 (insertion no. 5), *D. erecta* UAS-GCaMP6s (insertion no. 5), *D. erecta* UAS-CD8::GFP (insertion no. 2), *D. erecta* UAS-CsChrimson::tdTomato (insertion no. 1), *D. yakuba* dsx^Gal4 and dsx^AD inserts[64] (Y. Ding, University of Pennsylvania); *D. yakuba* UAS-mCD8-GFP (insertion nos. 2 & 3), UAS-GFP (insertion no. 8), UAS-SPA-GFP (insertion no. 2), ppk23-Gal4 (insertion no. 2), *ppk23* mutant, *ppk25* mutant and Or67d mutant were generated in this study. See Supplementary Table 1 for detailed genotypes by figure.

### Construct design and generation

The ie1.mCherry and ie1.eGFP minigene reporters used as transformation markers were constructed from the following fragments: the ie1 promoter was PCR amplified from pBac{orco.QF, QUAS.GCaMP, ie1.DsRed} (gift from D. Kronauer), *Drosophila* codon-optimized mCherry and eGFP gene blocks were synthesized (IDT), and the p10-3′ untranslated region (UTR) was PCR amplified from pJFRC81 (Addgene, 36432). These were then cloned by Gibson assembly (NEB) into pBac{3XP3-EGFP, Pactin-Ptrsps} (Addgene, 86861) after digest with BstZ17I and Bsu36I to generate pRC7a (ie1.mCherry) and pRC10a (ie.eGFP).

To create a more flexible multicloning site in pRC10a, a gBlock (IDT) containing the following restriction enzyme sites, HpaI, HincII, HindIII, KpnI, NotI, XhoI, XbaI and FseI, was synthesized and cloned by Gibson assembly into pRC10a. The resulting vector was pIM145. mCherry was then cloned with NheI and HpaI from pRC7a into the pIM145 backbone in place of GFP to generate pIM148.

To generate a P1 driver line, the 3,844 bp fragment from *Vsx2* gene that is present in pGMR71G01 was amplified from *D. melanogaster* genomic DNA. This was then cloned by Gibson assembly into the backbone of pIM145 in front of a Gal4 gene block (IDT), generating pRC12b.

To generate a BAC vector containing UAS-GFP, the following cassette, 10 × UAS-IVS-Syn21-GFP p10 UTR, was excised from pJFRC81 (Addgene, 36432) with HindIII and FseI and ligated into pIM145. The resulting vector was pIM153.

To generate a BAC vector containing UAS-SPA-GFP, a *Drosophila* codon-optimized SPA-GFP was synthesized as a gBlock (IDT) and cloned by Gibson assembly (NEB) into pIM147 in place of GCaMP6s to generated pIM141.

To generate ppk23-Gal4, a DNeasy Blood & Tissue kit (Qiagen) was used to purify wild-type *D. yakuba* genomic DNA. First, 50 mg of flies were collected and snap-frozen in liquid nitrogen. Flies were ground to a fine powder using a mortar and pestle. Powder was transferred to a microcentrifuge tube and processed according to the DNeasy Blood & Tissue kit instructions. The resulting genomic DNA was used as a template for PCR amplification of the ppk23 promoter. The 2.7 kb promoter was cloned into pRC12B to generate pIM169.

To generate a BAC vector containing a blue fluorescent marker for visualization, a *Drosophila* codon-optimized mTagBFP2 was synthesized. Owing to the sequence complexity of mTagBFP2, it was designed as two overlapping gBlocks (IDT) and included a portion of the ie1 promoter for seamless assembly into the destination vector. mTagBFP2 was cloned by Gibson assembly (NEB) into pIM148 in place of mCherry to generate pIM149.

### CRISPR–Cas9-mediated deletion of *ppk23* in *D. yakuba*

Two single guide RNAs (sgRNAs) were designed to direct Cas9-mediated cleavage to the first exon and 3′ UTR of the *ppk23* gene locus in *D. yakuba*. gRNA off-target potential was determined using CRISPR optimal target finder (http://tools.flycrispr.molbio.wisc.edu/targetFinder/index.php). sgRNA sequences, sgRNA1 CATCGGTGCGGTCACCGCAC and sgRNA2 GTGTTGCATACTTAGCGGCG, were PCR amplified with Q5 High-Fidelity master mix (NEB) and cloned into pCFD4 (Addgene, 49411) by Gibson assembly (NEB). The resulting vector was pIM179. pIM148 was digested with HpaI and SmaI to liberate ie1p mCherry p10 UTR. The mCherry cassette was then ligated to pDsRed-attP (Addgene, 51019) which was digested with AgeI and BsiWI to remove the 3 × P3-DsRed cassette then Klenow end-filled. This generated pIM174. The 1 kb homology arms beginning at the predicted Cas9 cut sites in *ppk23* were PCR amplified with Q5 High-Fidelity master mix (NEB) and cloned into pIM174. The resulting vector was pIM175. A cocktail of pIM179, pIM175 and Cas9 protein was injected into wild-type *D. yakuba* embryos by Rainbow Transgenic Flies using standard injection procedures. Viable G0 flies were mated to wild-type male or virgin female flies. F1 progeny were screened visually by mCherry expression. mCherry-positive F1s were individually crossed to wild-type male or virgin female flies then killed for genomic DNA. Deletion of *ppk23* was confirmed in mCherry-positive F1s by genotyping using primers internal to and flanking the targeted genomic region. PCR products from genotyping were sequenced to verify the exact genome modification. mCherry-positive F2s from sequence-verified F1s were self-mated and mCherry-positive F3 virgin females were genotyped by non-lethal methods to identify females homozygous for *ppk23* deletion. As the *ppk23* locus is on the X chromosome, virgin females homozygous for the *ppk23* null mutation were then mated to males hemizygous for the mutation to produce a stable line.

### CRISPR–Cas9-mediated deletion of *ppk25* in *D. yakuba*

Two sgRNAs were designed to direct Cas9-mediated cleavage to the first exon and 3′ UTR of the ppk25 gene locus in *D. yakuba*. gRNA off-target potential was determined using CRISPR optimal target finder. sgRNA sequences, sgRNA1 GUCGGUCGAUGCAACCGGAC and sgRNA2 UAAACUUAACAACAUCGGAG, were synthesized from Synthego. The 1 kb homology arms beginning at the predicted Cas9 cut sites in ppk25 were ordered as gBlocks from IDT. The ppk25 start code in the 5′ homology arm was mutated from ATG to TTG. The homology arms were cloned sequentially into pIM174 by Gibson assembly. The resulting vector was pIM188. A cocktail of pIM188, sgRNA1, sgRNA2 and Cas9 protein was injected into wild-type *D. yakuba* embryos by Rainbow Transgenic Flies using standard injection procedures. Viable G0 flies were mated to wild-type male or virgin female flies. F1 progeny were screened visually by mCherry expression. mCherry-positive F1s were individually crossed to wild-type male or virgin female flies then killed for genomic DNA. Deletion of ppk25 was confirmed in mCherry-positive F1s by genotyping using primers internal to and flanking the targeted genomic region. PCR products from genotyping were sequenced to verify exact genome modification. mCherry-positive F2s from sequence-verified F1s were self-mated and mCherry-positive F3 virgin females and males were genotyped by non-lethal methods to identify flies homozygous for ppk25 deletion.

### CRISPR–Cas9-mediated deletion of *Or67d* in *D. yakuba*

Two sgRNAs were designed to target 22 bp downstream of the start codon (sgRNA1) and 392 bp (sgRNA2) downstream of the stop codon of *D. yakuba Or67d*, removing a total of 1,783 bp of endogenous DNA.

Off-targets were determined using CRISPR optimal target finder. sgRNA1 (GACUUUACGAAAGCGCUCCA) and sgRNA2 (ACUGCUGCUG UCCAAAGGAG) were synthesized by Synthego. A 1,035 bp 5′ homology arm was amplified from *D. yakuba* genomic DNA using Q5 High-Fidelity master mix (NEB) and cloned into pIM174 using XmaI and NdeI restriction enzymes. A 1,174 bp 3′ homology arm of was amplified from *D. yakuba* genomic DNA using Q5 High-Fidelity master mix (NEB) and cloned into pIM174 using AvrII and XhoI restriction enzymes. The resulting vector was pGK1. A cocktail of pGK1, sgRNA1/2 and Cas9 protein was injected into wild-type *D. yakuba* embryos by Rainbow Transgenic Flies using standard injection procedures. Viable G0 flies were mated to wild-type male or virgin female flies. Progeny were screened, mated and genotyped as described for other mutants to produce a stable line.

## Immunohistochemistry

Adult brains were dissected in Schneider's media (Sigma) then immediately transferred to cold 1% PFA (Electron Microscopy Sciences) and fixed overnight at 4 °C. Following overnight incubation samples were washed in PAT3 buffer (0.5% BSA/0.5% Triton/PBS pH 7.4) three times. Brains were blocked in 3% Normal Goat Serum for 90 min at room temperature. Primary antibodies 1:1,000 chicken anti-GFP (Abcam, ab13970), 1:50 mouse anti-brp (Developmental Studies Hybridoma Bank nc82), 1:2,000 rabbit anti-Fru$^{M}$ (generated for this study by Yen-Zyme against a synthesized peptide: HYAALDLQTPHKRNIETDV[70]) and 1:500 guinea pig anti-Fru$^{M}$ (gift from D. Yamamoto, Tohoku University) were incubated for 3 h at room temperature then for 2–3 d at 4 °C. Brains were washed extensively in PAT3 buffer. Secondary Alexa Fluor antibodies (Life Technologies) were incubated for 3 h at room temperature then for 2–3 d at 4 °C. Brains were washed three times in PAT3 buffer then once in PBS. Samples were mounted in Vectashield (Vector Laboratories). Images were captured on a Zeiss LSM 880 using a Plan-Apochromat ×20 (0.8 numerical aperture) objective.

Leg images were taken using the native fluorescence of animals expressing UAS-CD8::GFP (insertion no. 2). Animals were aged approximately 3–5 days and legs were mounted in Vectashield and femurs stabilized using ultraviolet glue. Images were taken at ×25 with ×1.6 digital zoom.

For analysis of P1 projections, images were registered to the JRC2018M template brain using the Computational Morphometry Toolkit (https://www.nitrc.org/projects/cmtk/) and P1 neurons were segmented in VVD Viewer (https://github.com/JaneliaSciComp/VVDViewer).

## Courtship assays in the light

All single choice assays were conducted at 25 °C, 50–65% relative humidity between 0 and 3 h after lights on. Male flies were collected shortly after eclosion and group housed for 4–7 days before assay. Target virgin females were 4–7 days post-eclosion. Assays were performed in 38-mm-diameter, 3-mm-height circular, slope-walled chambers in a 4 × 4 array back-lit using a white light pad (Logan Electric). Fly behaviour was recorded from above the chambers using a PowerShot SX620 camera (Canon) or Point Grey FLIR Grasshopper USB3 camera (GS3-U3-23S6M-C: 2.3 MP, 162 FPS, Sony IMX174, Monochrome). A virgin female was transferred to the chamber by mouth aspiration followed by a test male. Once the male was loaded into the chamber the assay commenced and the activity of the flies was recorded for 10 min. Note that courtship assays in the light were not conducted on food to avoid flies copulating too quickly under these conditions. In the absence of food, females generally move much quicker. This results in shorter male courtship bouts, probably explaining the difference in bout lengths observed between males in the dark on food compared with males in the light off food (that is, bouts were generally longer in the dark when on food).

## Courtship assays in the dark

Courtship assays performed in the dark were conducted at 25 °C, 50–65% relative humidity between 0 and 3 h after lights on. Male flies were collected shortly after eclosion and group housed for 4–7 days before assay. Target virgin females were 4–7 days old. Assays were performed on food in 35 × 10 mm² Petri Dishes (Falcon). Females were transferred to the chamber by mouth aspiration followed by a test male. Once the male was loaded into the chamber the assay commenced and the activity of the flies was recorded for 3 h. The extended recording period was necessary due to the lengthy latency to courtship initiation in the dark, where males must discover females by chance. A manual observer recorded the time of courtship initiation for all videos, and the subsequent 10 min were used for analysis. Assays were back-lit by infrared light-emitting diode (LED) strips (940 nm, LED Lights World). Fly behaviour was recorded from above the chambers using a Point Grey FLIR Grasshopper USB3 camera (GS3-U3-23S6M-C: 2.3 MP, 162 FPS, Sony IMX174, Monochrome) using the Flycapture2 Software Development Kit (v.2.13.3.61) (FLIR).

## Courtship quantification

For all behavioural experiments, variance was assessed by preliminary study and predetermined sample size were chosen. Collection of experimental and control animals was randomized with roughly equal numbers collected on the same days. Courtship was scored either by automated tracking using machine vision (Matlab, FlyTracker, Caltech) or manually by an observer blind to the experimental set-up (for example, species, genotype, perfume-treatment). For manual analysis orienting, wing extension, chasing, mounting and copulation were used to score courtship behaviour. To facilitate plotting on a log scale all 0 values were changed to 0.1. For automated courtship scoring, an 8 mm IFD threshold was found to most accurately capture the courtship bouts scored by the blind observer, producing the lowest number of false positives and negatives. Unfortunately, variations in the food and/or infrared lighting meant that for approximately half of the videos collected we could not obtain high-quality tracking data. Thus, we opted to track a random subset, that is, the first 12 consecutive courting pairs from which quality tracking was possible. This provides a representative visual assessment of courtship dynamics, whereas the complete dataset of all courting pairs was scored manually by blinded observer. Extended Data Fig. 1 illustrates that although IFD tracking and manually scored datasets differed in the absolute quantity of courtship scored, the overall trends in the data were consistent. However, because false detections in the automatically scored data compress the dynamic range of the data, we believe that bout length is the most sensitive indicator of pheromone-dependent courtship in assays conducted in the dark, effectively capturing a male's ability to persistently pursue a female, guided by ongoing pheromonal feedback. Consequently, bout length scored by a blinded observer is used in all figures. The only exceptions to using this metric throughout our study are: (1) in the initial analysis of courtship in the dark (Fig. 1), where automatically quantified IFD is compared with these manually scored courtship metrics (Extended Data Fig. 1) before selecting blinded, manually scored bout length as the most accurate; (2) in the optogenetic analysis of P1 subtypes in *D. melanogaster* and *D. yakuba*, for which it was important to provide a more granular description of courtship dynamics (Fig. 4a,b and Extended Data Figs. 6a,b and 7), and thus JAABA (Janelia) behavioural classifiers were employed to quantify courtship pursuit and song; and (3) in the extended data, where in addition to bout length, more common measurements of courtship (for example, latency to court, total percentage time courting) were plotted to provide a comprehensive view (Extended Data Figs. 3b–d, 4b and 8a,c,d,f,g).

## Generation of oe⁻ animals

Oenocyte ablation was performed genetically by crossing male +;PromE(800)-Gal4, tub-Gal80$^{TS}$;+ flies to female +;UAS-StingerII, UAS-hid/CyO;+ at 18 °C. Newly eclosed virgin females were collected and kept at 25 °C for 1 d. Females were then shifted to 30 °C for 2 d and then allowed to recover at 25 °C for 2 d before use in experiments. Females were screened for GFP expression to confirm oenocyte ablation[12].

## Perfuming

Mock, 7-T and 7,11-HD perfuming of oe⁻ or *D. yakuba* females was performed by adding heptane, 2 µg of 7-T or 2 µg of 7,11-HD (Cayman Chemicals), respectively, to 1 ml of heptane in a 32 mm scintillation vial (ThermoScientific). A vacuum was then applied to evaporate the heptane from the vial. We added 5–8 flies to each vial and vortexed at low speed for 30 s, three times. cVA perfuming was perfumed by dissolving cVA to a concentration of 5 mg ml⁻¹ in ethanol. Then, 0.5 µl was then applied directly to each fly's abdomen by pipette. All flies were returned to food to recover for at least 1 h before experimentation.

## Photoactivation

For photoactivation experiments fru^Gal4 was used to express SPA-GFP. Photoactivation was performed on adult flies aged 24–48 h after eclosion. Brains were imaged at 925 nm to identify suitable sites for photolabeling while not stimulating photoconversion. Using PrairieView, a region of interest (ROI) was then drawn around projections unique to the cell type of interest and photoconversion was stimulated in a single *z*-plane by brief exposure of the ROI to 710 nm laser light. Power was 5–35 mW at the back aperture of the objective, depending on the depth of the neurite being converted. This process was repeated 50–100 times, interposed with rests to allow for diffusion of the photoconverted molecules, until the cell type of interest was uniformly above background levels of fluorescence.

## Functional imaging

All imaging experiments were performed on an Ultima two-photon laser scanning microscope (Bruker) equipped with galvanometers driving a Chameleon Ultra II Ti:Sapphire laser. Emitted fluorescence was detected with GaAsP photodiode (Hamamatsu) detectors. All images were collected using PrairieView Software (v.5.5) at 512 × 512 pixel resolution. Fluorescence time-series were extracted using FIJI (v.2.14.0/1.54f). Ventral nerve cord and LPC preparations were performed as previously reported[13,19]. For all imaging experiments, the presentation order of female or male stimuli was randomized and the strongest responding stimulus was presented again the end of an experiment to confirm the continued health of experimental male. Sample sizes were not predetermined.

For the *D. yakuba* P1 imaging, 'splitP1' (71G01-AD 15A01-DBD intersection) had to be used rather than 71G01-Gal4 because this line was weak and bleached before an experiment could be completed. These splitP1 animals showed inter-animal variability in responses. Only about 1 in 5 (5 of 27) animals exhibited P1 responses to conspecifics, although in responding animals the responses were uniform and predictably evoked each time a male tapped a conspecific female (as shown in Fig. 2). We discovered this probably resulted from stochastic labelling of the Fru⁻Dsx⁺ P1 subset. Specifically, splitP1 predominantly labels Fru⁺ P1 neurons, and labels 2–3 Fru⁻Dsx⁺ neurons in only some animals, as previously found in *D. melanogaster*[65]. Indeed, when we subsequently imaged Dsx∩P1 (71G01-DBD dsx-AD intersection), no inter-animal variability was observed and all animals responded to conspecifics with each tap. The same was found to be true when imaging for all Dsx⁺ projections in the LPC. Therefore, for simplicity and understanding we have presented just responding *D. yakuba* splitP1 males in Fig. 2b.

## Imaging analysis

For each GCaMP recording, an ROI was drawn in the LPC neuropil where axonal projections are densest for P1 (indicated in Fig. 2a) or in the ventral nerve cord neuropil where Ppk23⁺ neurons have their sensory afferents (indicated in Fig. 3a). For all experiments, 3–5 s were recorded before the stimulus presentation to create a baseline. Twenty frames (approximately 2 s) of this pre-stimulus period were then averaged to determine baseline fluorescence ($F_0$), and $\Delta F/F_0$ was calculated as $\Delta F_t/F_0 = (F_t - F_0)/F_0$, where $t$ denotes the current frame.

## Optogenetic set-up

Optogenetic assays were performed in a 38 mm diameter, 3 mm height circular chamber with sloping walls. The chamber was placed in the middle of a 3-mm-thick acrylic sheet suspended on aluminium posts above a 3 × 4 array of 627 nm LEDs (Luxeon Star LEDs). LEDs were attached to metal heat sinks (Mouser Electronics) which were secured at 5 cm intervals to a 30 × 30 cm² aluminium wire cloth sheet (McMaster-Carr). LEDs were driven by Recom Power RCD-24-0.70/W/X2 drivers, which were powered by a variable DC power supply. Infrared LED strips (940 nm, LED Lights World) attached to the wire cloth between the heat sinks provided back-illumination of the platform. LED strips were covered with 071 Tokyo blue filter (Lee Filters) to remove potential activating light emitted from the illumination source. LED drivers were controlled by the output pins of an Arduino running custom software. Fly behaviour was monitored from above the chamber using a Point Grey FLIR Grasshopper USB3 camera (GS3-U3-23S6M-C: 2.3 MP, 162 FPS, Sony IMX174, Monochrome) outfitted with 071 Tokyo blue filter (Lee Filters) to avoid detection of light from the high-power LEDs. Flies were recorded at 30 frames per second. Custom software was used for data acquisition and instrument control during assays. Light intensity was measured with a photodiode power sensor (Coherent, 1212310) placed at the location of the behavioural chamber. The peak wavelength of the LED (627 nm) was measured across a range of voltage inputs. Measurements were repeated three times and averaged. The baseline intensity before LED illumination was subtracted.

## Optogenetic assays

Flies were reared on standard SY food in the dark at 25 °C and 50–65% relative humidity. P1>UAS-CsChrimson and UAS-CsChrimson control male flies were collected shortly after eclosion, group housed for 3 d, then shifted to food containing 0.4 mM all *trans*-retinal (Sigma) for 48 h before being assayed. ppk23-Gal4>UAS-CsChrimson and UAS-CsChrimson control male flies were collected shortly after eclosion, group housed for 3 d, then shifted to food containing 0.4 mM all *trans*-retinal for 24 h before being assayed. Fru∩P1>UAS-CsChrimson, Dsx∩P1>UAS-CsChrimson and UAS-CsChrimson control males were collected shortly after eclosion, grouped housed for 2 d and then shifted to food containing 0.4 mM all *trans*-retinal for 24 h (Fru∩P1) or 48 h (Dsx∩P1) before being assayed. Target virgin females were 4–6 d old. Flies were added to a 38 mm diameter, 3 mm height circular courtship chamber by mouth aspiration. Once a male was transferred to a courtship chamber containing a virgin female, the assay commenced and the activity of the flies was recorded for 10 min. Neurons expressing CsChrimson were activated by 627 nm wavelength LED stimulation. To activate splitP1 (w; 71G01-AD/+; 15A01-DBD/UAS-CsChrimson) neurons in *D. yakuba* the following stimulation protocol was used: 2 min dim white light followed by 2 min 627 nm LED (5 Hz, 100 ms pulse-width, 10 µW mm⁻²) alternating for 10 min total. For activation of P1 (UAS-CsChrimson::tdTomato/+; 71G01-Gal4/+) neurons in *D. erecta* the stimulation protocol was: 2 min dim white light followed by 2 min 627 nm LED (5 Hz, 100 ms pulse-width, 3.4 µW mm⁻²) alternating for 10 min total. To activate Ppk23⁺ neurons in *D. yakuba* the following stimulation protocol was used: 2 min dim white light followed by 2 min 627 nm LED (5 Hz, 100 ms pulse-width, 8 µW mm⁻²) alternating for 10 min total. All assays were conducted at 25 °C, 50–65% relative humidity between 0 and 3 h after lights on. To activate Fru∩P1 and Dsx∩P1 neurons in *D. yakuba* the following stimulation protocol was used: 2 min dim white light followed by 2 min 627 nm LED (5 Hz, 100 ms pulse-width, 14 µW mm⁻² for Fru∩P1 and 37 µW mm⁻² for Dsx∩P1) alternating for 10 min total. The 48 h retinol stimulation protocol was used in *D. melanogaster* males for each P1 population, and the light intensity used was 3.4 µW mm⁻². A blinded observer scored orienting, wing extension, chasing, mounting and copulation to quantify courtship behaviour for the purposes of calculating overall courtship indices. To quantify

dynamic behaviours, videos were tracked using FlyTracker (Caltech). Behavioural classifiers for courtship pursuit behaviour (consisting or orienting towards the female and moving towards her) and unilateral wing extensions were then trained using JAABA (Janelia) in each species.

## Statistical analysis

Statistical analyses were performed in GraphPad Prism 9. Before analysis, normality was tested for using the Shapiro–Wilk method for determining whether parametric or non-parametric statistical tests would be used. In cases for which multiple comparisons were made, appropriate post hoc tests were conducted as indicated in the figure legends. All statistical tests used were two-tailed. Experimenters were blind to experimental conditions during analysis.

## Reporting summary

Further information on research design is available in the Nature Portfolio Reporting Summary linked to this article.

## Data availability

All data underlying this study are available upon request from the corresponding author. Source data are provided with this paper.

## Code availability

All code used to analyse and present data in this study is available on request from the corresponding author.

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

**Acknowledgements** We thank D. Stern for sharing several *D. yakuba* stocks; T. Hart and D. Kronauer for ie1 promoter DNA; M. Beye for hyperactive piggyBac transposase expression construct; B. Matthews and L. Seeholzer for technical advice; M. Khallaf for information on pheromone compounds produced across species; and D. Stern, B. Datta, B. Noro, J. Ouadah, A. Paul, T. Hindmarsh Sten, P. Brand, J. Rhee, A. Ryba and all the members of the Ruta laboratory for valuable discussion and comments on the manuscript. This work was supported by an NIH NINDS grant (5R35NS111611), the Simons Foundation Collaboration for the Global Brain to V.R., an NIH NIGMS grant (R35GM142678) to Y.D., and a Helen Hay Whitney Foundation Fellowship and an NIH NIGMS grant (K99GM141319) to R.T.C. V.R. is a Howard Hughes Medical Institute Investigator.

**Author contributions** R.T.C. and V.R. conceived of the project. R.T.C., I.M., G.T.K. and M.L.C. conducted and analysed experiments. R.T.C., I.M. and Y.D. designed transgenesis strategies and built neurogenetic reagents. R.T.C. and V.R. wrote the manuscript. All authors provided critical feedback and helped shape the research, analysis and manuscript.

**Competing interests** The authors declare no competing interests.

**Additional information**
**Correspondence and requests for materials** should be addressed to Vanessa Ruta.

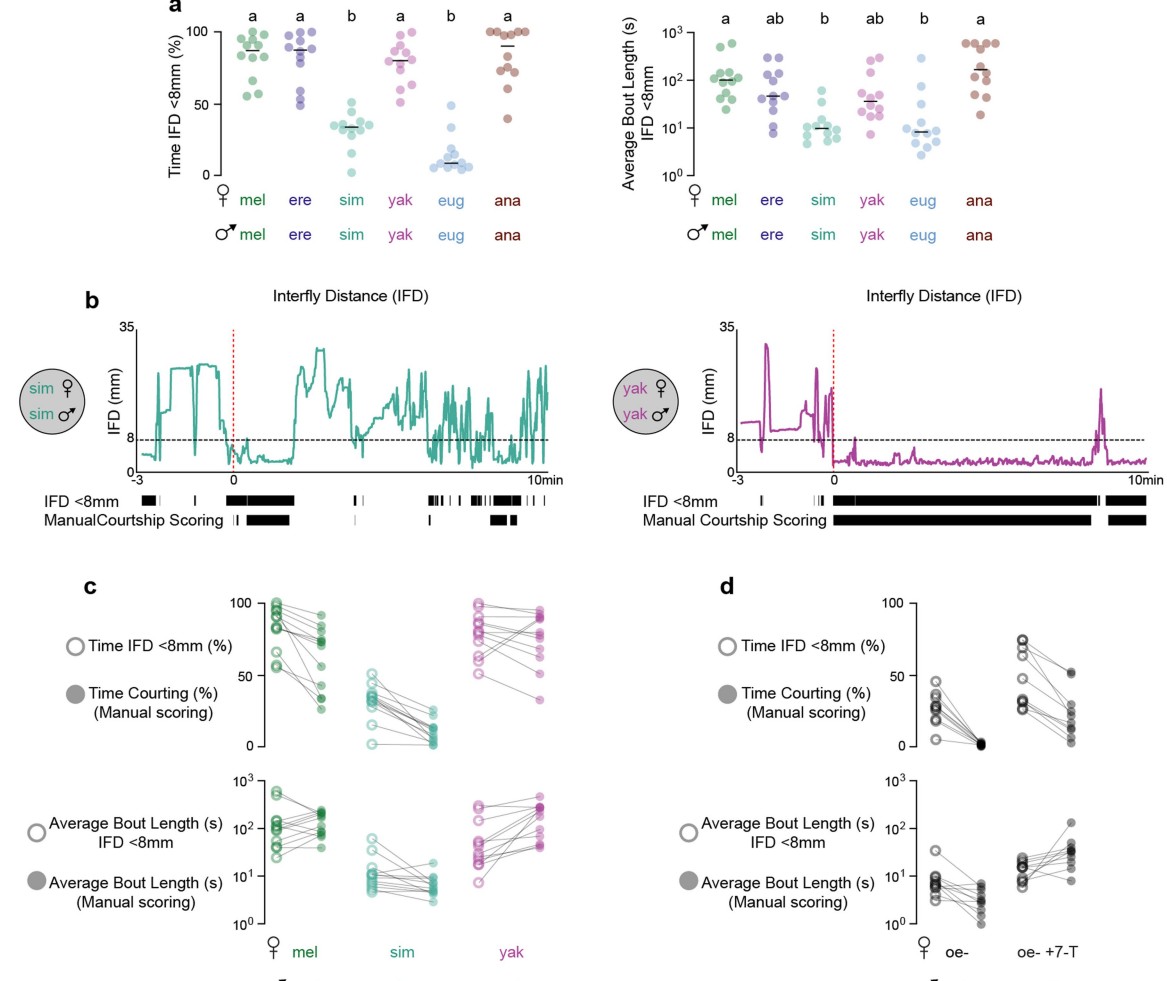

**Extended Data Fig. 1 | Inter-fly distances below 8 mm approximate courtship in the dark. a**, Automated scoring of the samples plotted in Fig. 1b. (Left) percent time pairs spent at IFD < 8 mm and (Right) average bout length at IFD < 8 mm after first interaction (eug) or after courtship initiation (all other species). Each data point represents the individual pairs shown in the heatmaps in Fig. 1b (n = 12 each) **b**, (Top) Inter-fly distance (IFD) traces over time for a single representative pair of *D. simulans* or *D. yakuba* as courtship proceeds. Comparison of manually confirmed courtship bouts with those estimated by IFD thresholding revealed that an IFD threshold of 8 mm most accurately replicated manual scoring, resulting in the lowest incidence of false positives and negatives. (Bottom) Courtship bouts approximated by IFD < 8 mm thresholding compared to manual scoring by a blinded observer. Red dotted lines indicate initiation of courtship. **c,d**, Comparison between the percent time courting and the average courtship bout length scored by automated IFD < 8 mm thresholding versus manually scored by a blinded observer for the same courting pairs across species (c) or for *D. yakuba* male courtship of oenocyte-less females perfumed with the *D. yakuba* pheromone 7-T (d)(n = 12 each). Generally, IFD-thresholding overestimates the total amounts of courtship (due to incidental periods where flies are close but not courting) and underestimates the average courtship bout length (due to transient periods where the female moves away from the male but courtship continues), but captures the overall trends observed in samples manually scored by a blinded observer (Fig. 1c, d). The samples analyzed here in **c** are the same plotted in Fig. 1b and the samples analyzed in **d** are the same plotted in Fig. 1d (n = 10 each). All quantifications are of the 10 min period following courtship initiation. Data points represent individual males, bars are median. Statistics: Kruskal-Wallis tests (a). Letters denote statistically different groups (p < 0.05).

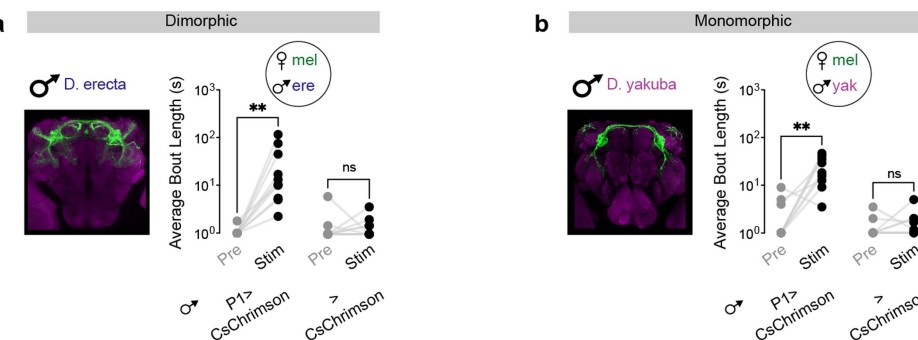

**Extended Data Fig. 2 | P1 neurons play a conserved role in promoting courtship across species. a,b,** Percent time courting prior to (Pre) or during (Stim) periods of optogenetic stimulation of *D. erecta* (a; n = 10 each) or *D. yakuba* (b; n = 9 each) males expressing CsChrimson in P1 neurons (71G01>CsChrimson (a) or splitP1>CsChrimson (b)) or control animals (> CsChrimson) paired with a *D. melanogaster* female. Data points represent individual males. Statistics: Wilcoxon test (a,b). Asterisks: **P < 0.01, ns, not significant.

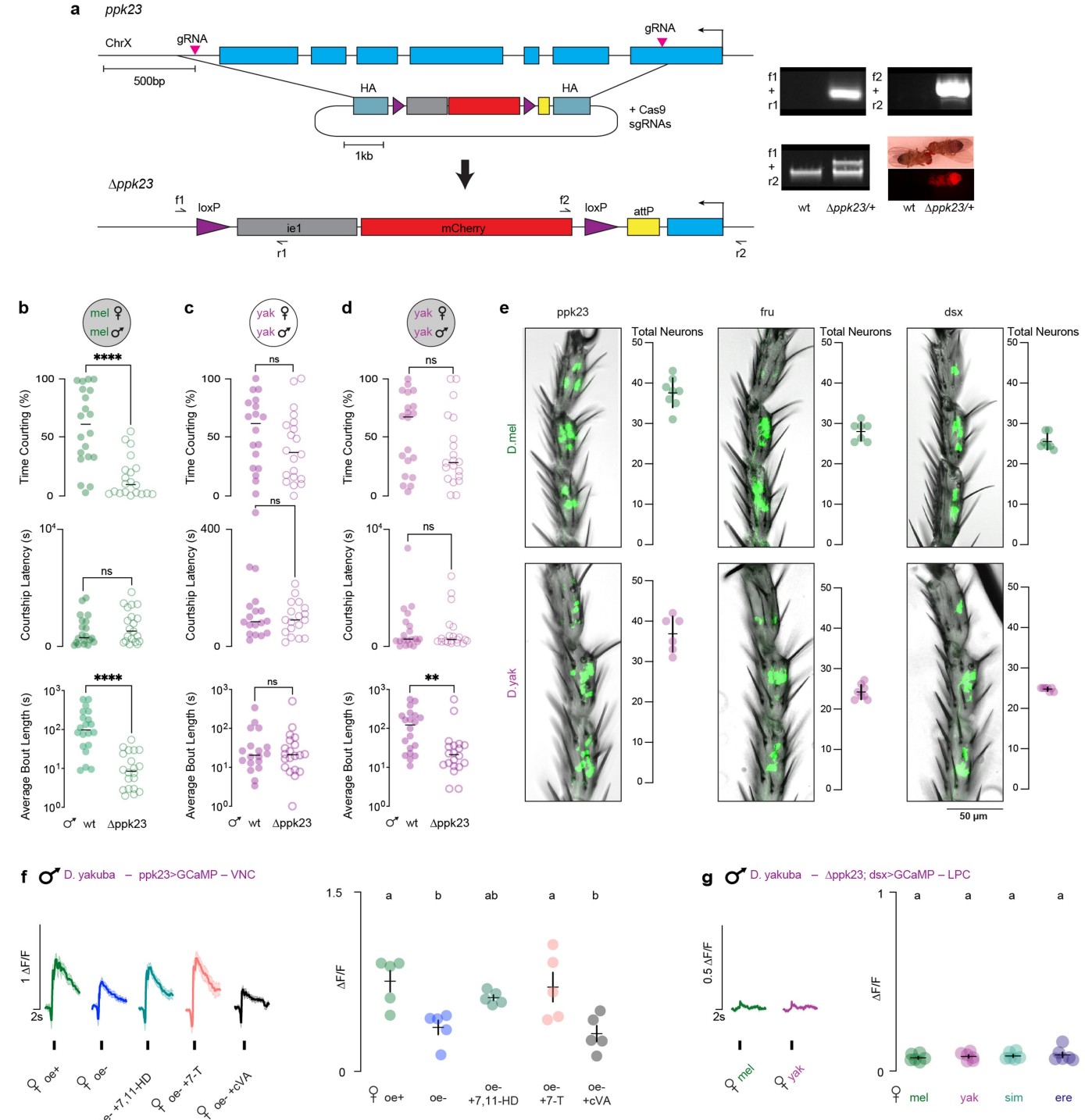

**Extended Data Fig. 3 | Conserved anatomy of pheromone-detecting sensory neurons. a**, CRISPR/Cas9 targeting strategy for the genetic deletion of the *ppk23* locus and validation by PCR and ie1-mCherry marker expression. **b,c,d**, Percent time courting in the 10 min following courtship initiation, courtship latency, and courtship bout length of *ppk23* mutants relative to wildtype for *D. melanogaster* in the dark (b) and *D. yakuba* males in the light (c) or in the dark (d) (n = 20 each). Average Bout Lengths are re-plotted from Fig. 3 for reference. **e**, Representative bright field images and quantification showing expression of CD8::GFP (green) in the soma of the Ppk23+, Fru+, and Dsx+ sensory neurons in the foreleg tarsal segments of *D. melanogaster* and *D. yakuba* males. Images are distal end up. **f**, (Left) Average functional responses ($\Delta F/F_0$) aligned to time of a tap (as in Fig. 3) of indicated *D. melanogaster* (oe+ ctrl),

mock-perfumed oenocyte-less (oe-), 7,11-heptacosadiene-perfumed (oe- +7,11-HD), 7-tricosene-perfumed (oe- +7-T), or cis-Vaccenyl acetate-perfumed (oe- +cVA) female of the foreleg sensory afferents and (Right) average peak response ($\Delta F/F_0$) per male for a given female target (n = 5). **g**, Functional responses of Dsx+ neurons in the LPC (as in Fig. 4d) in *ppk23* mutant males (n = 6)). For behavioral tests (b-c), points represent individual males, bars are median. For anatomy (e), points represent individual forelegs, bars are mean ± SEM. For imaging (f,g), shading is mean ± SEM, points are individual males, error bars are mean ± SEM. Statistics: ANOVA with Tukey's post-hoc (f,g) or unpaired Mann-Whitney (b-d). Letters denote statistically different groups (p < 0.05). Asterisks: **P < 0.01, ****P < 0.0001, ns, not significant.

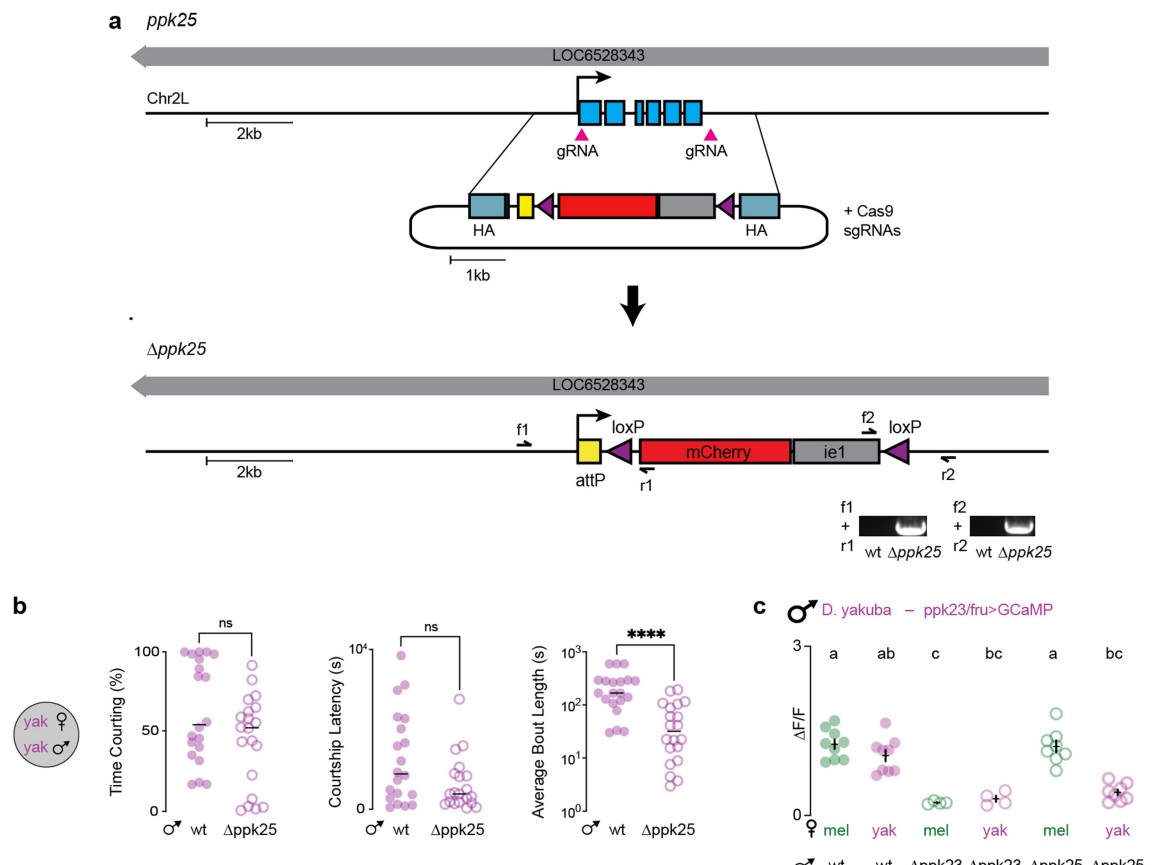

**Extended Data Fig. 4 | Ppk25+ Sensory neurons promote courtship in the dark in** *D. yakuba*. **a**,**b**, CRISPR/Cas9 targeting strategy for the genetic deletion of the *ppk25* locus and validation by PCR (a) and courtship in the dark (as in Extended Data Fig. 3) toward *D. yakuba* females (b). Average courtship bout lengths are re-plotted from Fig. 3 for reference (n = 20). **c**, Replotting of sensory responses in *D. yakuba* wildtype and *ppk23* and *ppk25* mutant males from Fig. 3e for statistical comparison. For behavioral tests (b), points represent individual males, bars are median. For imaging (c), points are individual males, error bars are mean ± SEM. Statistics: Unpaired Mann-Whitney (b) or ANOVA with Tukey's post-hoc (c). Letters denote statistically different groups (p < 0.05). Asterisks: ****P < 0.0001, ns, not significant.

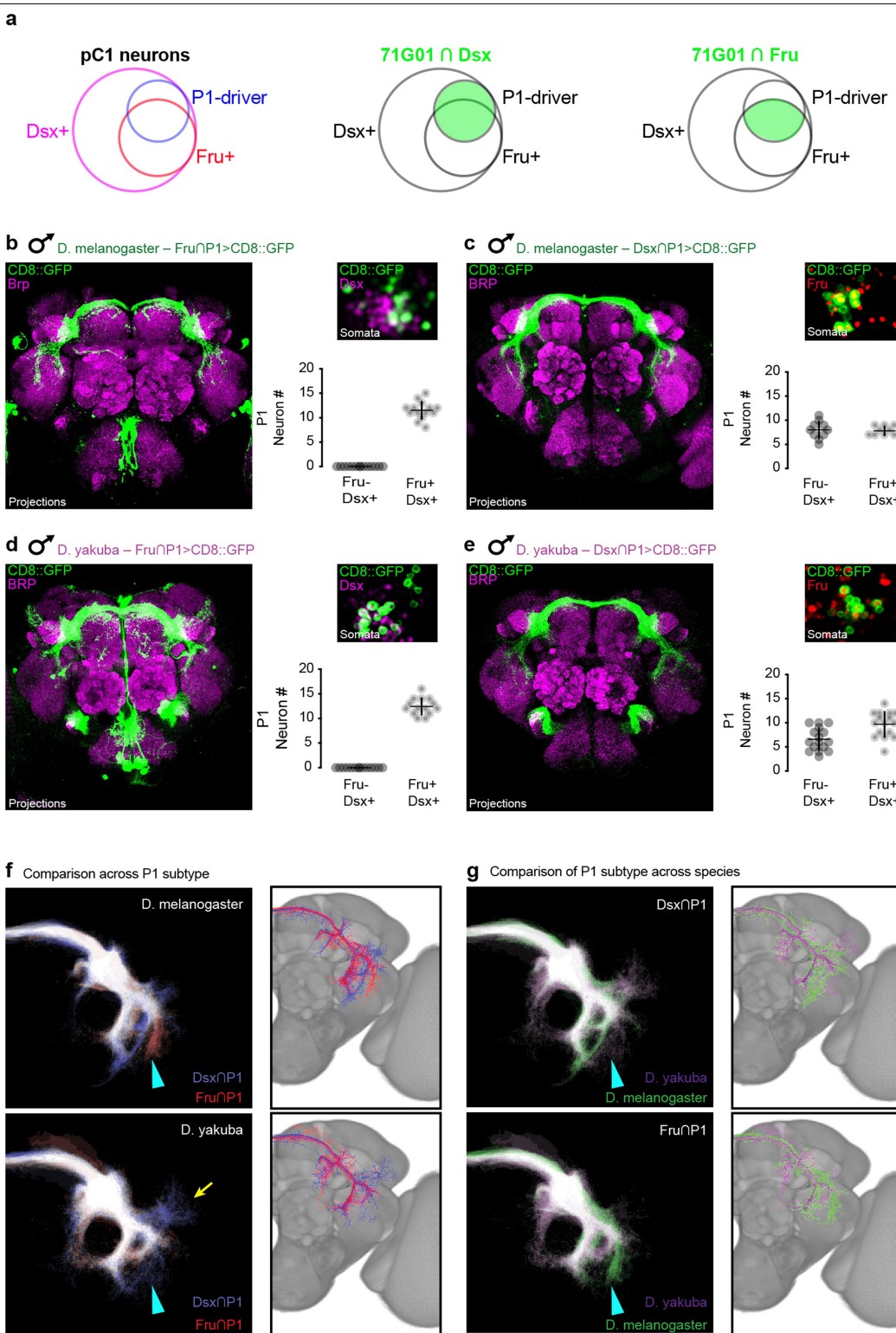

**Extended Data Fig. 5 | Differences in projection patterns of P1 subpopulations. a**, Diagram of the subdivision of the pC1 neurons by Dsx- (purple) and Fru-expression (red). Also noted is the subset of pC1 neurons captured by the P1-specific driver used in this study and the populations labeled through genetic intersection of this P1 driver with Dsx or Fru. Note that the labeled populations are not to scale, but precise neuron counts labeled in each intersection are provided in b-e. **b-e**, (Left) Unmasked images of Fru∩P1 and Dsx∩P1 neurons labeled by intersection of Fru (b,d) or Dsx (c,e) and the P1-driver 71G01 (Fru∩P1 and Dsx∩P1, respectively) in *D. melanogaster* and *D. yakuba*. Stained for GFP (green) and neuropil counterstain (magenta). (Right) Representative images of anti-Dsx (purple) and anti-Fru (red) staining used to

count Dsx+ and Fru+ subsets and quantification of Dsx+ Fru∩P1 neurons and Fru+ Dsx∩P1 neurons within the Dsx+ pC1 cluster, based on anti-Dsx and anti-Fru immunostaining, respectively. Data points represent cell number in one hemisphere and both hemisphere counts are plotted separately (Sample sizes: mel Fru∩P1 n = 7; mel Dsx∩P1 n = 8; yak Fru∩P1 n = 7; yak Dsx∩P1 n = 5). **f,g**, Comparison of the projections of Fru∩P1 and Dsx∩P1 subpopulations across subtype (f) or across species (g). Heatmap (left) and skeletonized overlays (right) of the comparison groups. Regions of greatest difference across both subtype and species indicated (cyan triangle) and region of significant difference between Dsx∩P1 and Fru∩P1 in *D. yakuba* noted (yellow arrow). All images are dorsal side up.

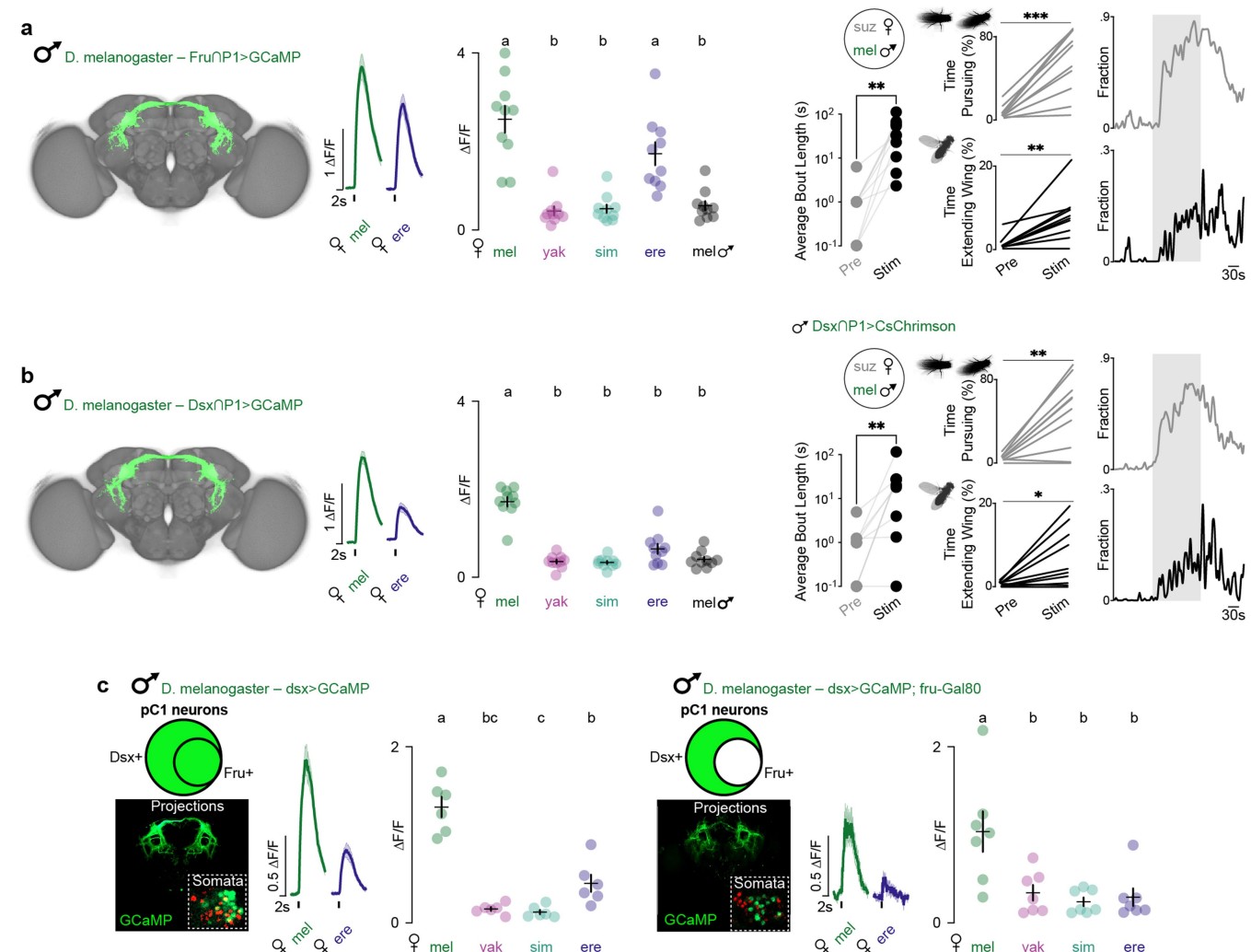

**Extended Data Fig. 6 | Subspecialization of P1 types in *D. melanogaster*.**
**a,b**, (Left) Maximum intensity projections of neurons labeled by intersection of Fru (a) or Dsx (b) and the P1-driver 71G01 (Fru∩P1 and Dsx∩P1, respectively) in *D. melanogaster*, registered to a common template brain (see methods). (Middle) Averaged functional responses ($\Delta F/F_0$) aligned to time of a tap of indicated conspecific or heterospecific female and average peak response ($\Delta F/F_0$) per male for a given female target (n = 10 each). (Right) Average courtship bout length, total percent time pursuing, or total percent time extending a unilateral wing toward a *D. suzukii* female target prior to (Pre) or during (Stim) optogenetic stimulation of Fru∩P1>CsChrimson or Dsx∩P1>CsChrimson males and the fraction of flies engaging in these behaviors over the course of the experiment. (n = 10 each) **c**, Functional responses of the or Dsx+ neurons innervating the LPC where P1 neurons reside with or without fru-Gal80. (Left) Diagram of the P1/pC1 neural cluster illustrating the nested nature of the Fru+ contingent within the broader Dsx+ population. Green highlighting represents

which subsets express GCaMP in the different genetic backgrounds (all Dsx+Fru- and Dsx+Fru+ pC1 neurons in dsx>GCaMP flies, but only the Dsx+Fru- subset in the dsx>GCaMP; fru-Gal80 flies). Also, representative images Dsx+ neural processes and somata expressing GCaMP (green) and counter-stained for Fru (red). Note the denser projections and the presence of Fru+ GCaMP+ somata in the dsx>GCaMP animals, which is absent in animals expressing fru-Gal80. (Right) Averaged functional responses ($\Delta F/F_0$) aligned to time of a tap of indicated conspecific or heterospecific female and (Right) average peak response ($\Delta F/F_0$) per male for a given female target. (Sample sizes: dsx>GCaMP n = 6; dsx>GCaMP;fru-Gal80 n = 7). For functional imaging (a-c), shading represents mean ± SEM. Points are individual males, error bars are mean ± SEM. For behavioral tests (a,b), points are individual males. Statistics: ANOVA with Tukey's post-hoc (a-c) or Wilcoxon test (a,b). Letters denote statistically different groups (p < 0.05). Asterisks: *P < 0.05, **P < 0.01, ***P < 0.001.

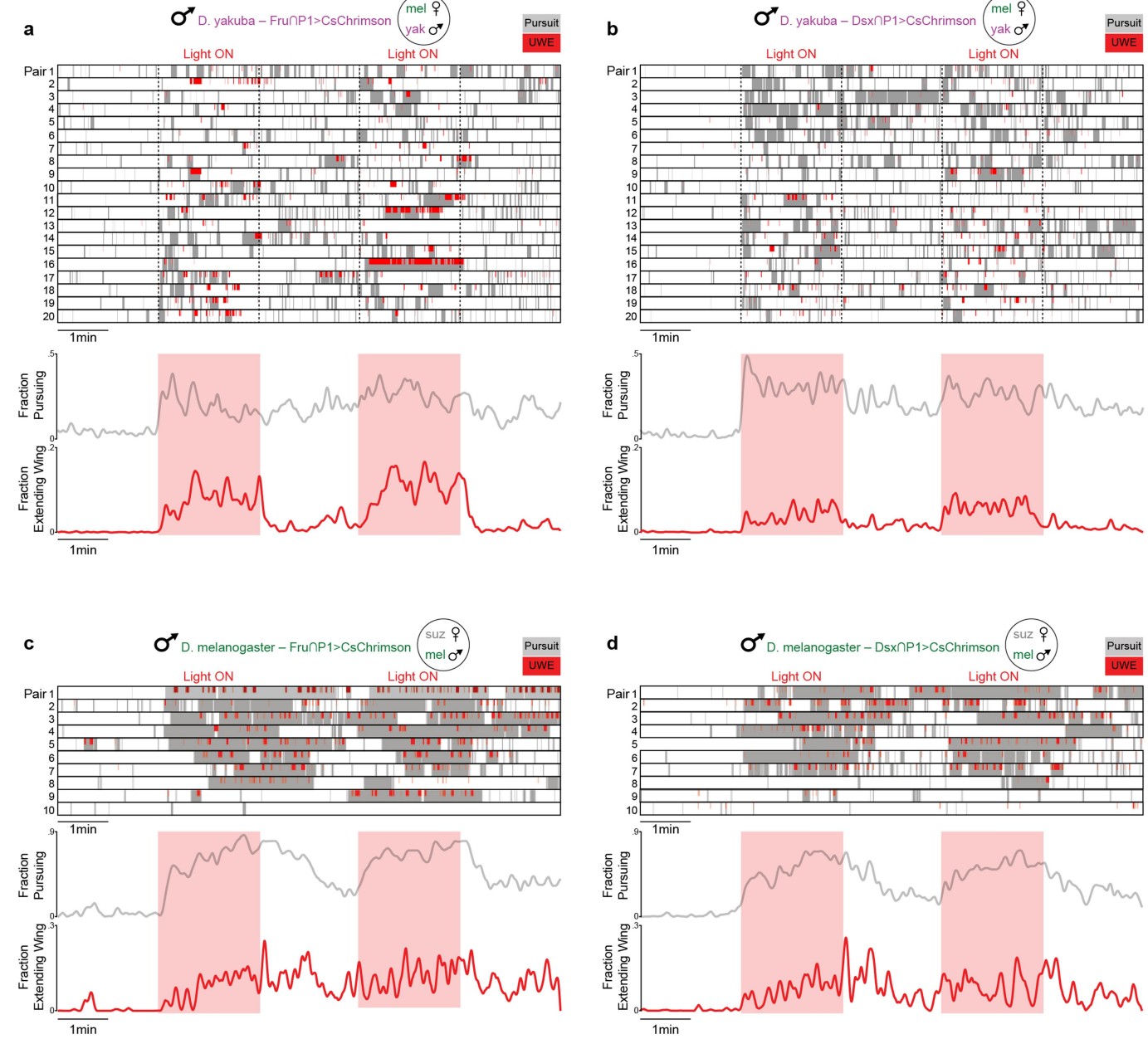

**Extended Data Fig. 7 | Activation of P1 subtypes. a-d**, (Top) Raster plots of female pursuit (gray) or unilateral wing extension behaviors (UWE; red) exhibited upon optogenetic stimulation of Fru∩P1>CsChrimson (a,c) or Dsx∩P1>CsChrimson (b,d) males and the proportion of flies engaging in these behaviors over the course of the experiment in *D. yakuba* (a,b; n = 20 each) or *D. melanogaster* (c,d; n = 10 each). Dotted lines and areas of red shading indicate stimulation periods.

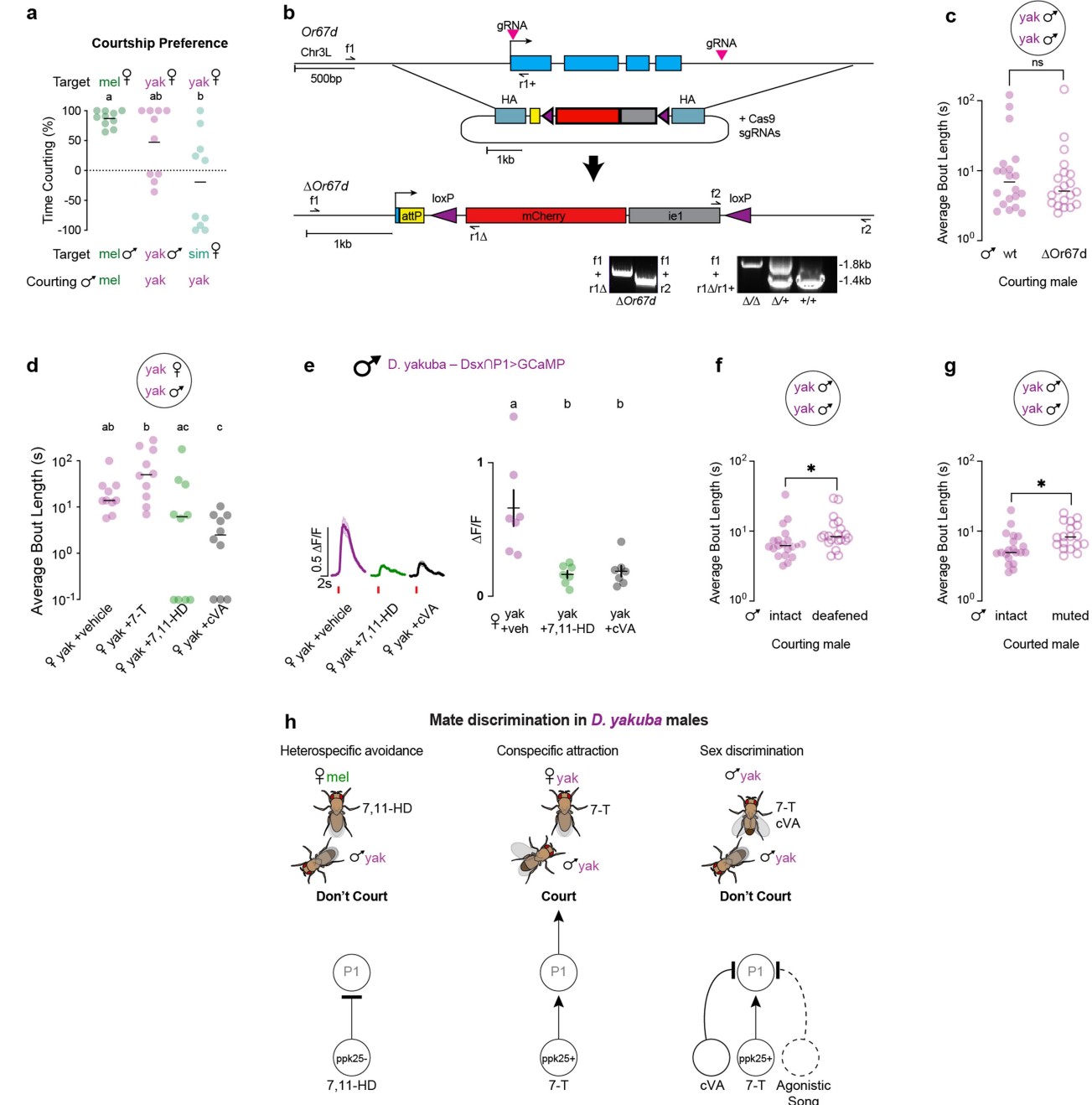

**Extended Data Fig. 8 | Multimodal sensory integration by P1 facilitates sex discrimination in *D. yakuba*. a**, Percent time a *D. melanogaster* or *D. yakuba* male spends courting one target over the other when presented a choice between a conspecific female and male or a conspecific female and a heterospecific female that shares the same pheromone profile (*D. simulans*) (n = 10 each). **b**, CRISPR/Cas9 targeting strategy for the genetic deletion of the *OR67d* locus and validation by PCR. **c**, Average courtship bout lengths of a wildtype or *OR67d* mutant *D. yakuba* male directed toward another *D. yakuba* male (n = 20 each). **d**, Average courtship bout lengths of a *D. yakuba* male toward a *D. yakuba* female mock perfumed or perfumed with 7-T, 7,11-HD, or cVA (n = 10 each). **e**, (Left) Averaged functional responses (ΔF/F₀) aligned to time of a tap of *D. yakuba* females perfumed as in c and average peak response

($\Delta F/F_0$) per male for a given female target (n = 7). **f,g**, Average courtship bout lengths of a wildtype or deafened (arista removed) *D. yakuba* male directed toward another *D. yakuba* male (f; n = 20 each) or a wildtype *D. yakuba* male toward a wildtype or muted (wings removed) *D. yakuba* male (g; intact n = 20, muted n = 18). h, Diagram summarizing the multisensory cues the *D. yakuba* males rely upon to mediate conspecific attraction and inhibit heterospecific and intrasexual courtship. For behavioral tests (a,c,d,f,g), points represent individual males, bars are median. For imaging (e), points are individual males, error bars are mean ± SEM. Statistics: Wilcoxon test (a,d), unpaired Mann-Whitney (c,f,g), or ANOVA with Tukey's post-hoc (e). Letters denote statistically different groups (p < 0.05). Asterisks: *P < 0.05, ns, not significant.

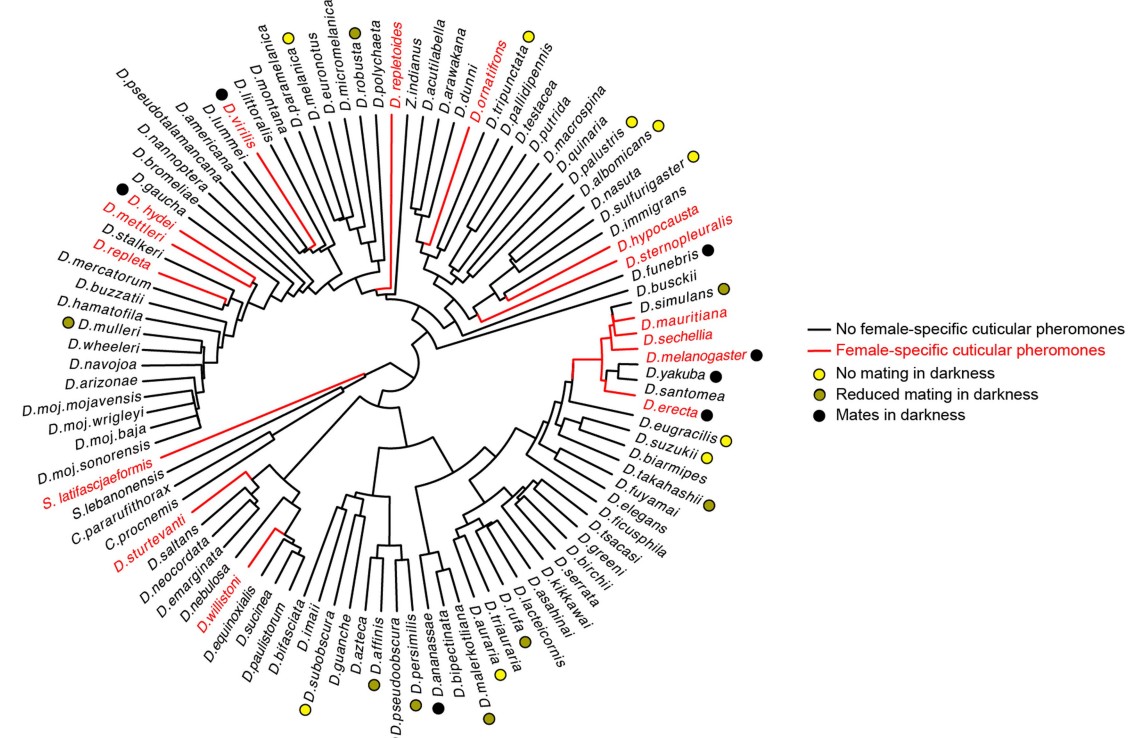

**Extended Data Fig. 9 | Mating in the dark has arisen repeatedly in monomorphic species.** Phylogeny of Drosophila species (as in Fig. 1) with previous reports of various species' capacity to mate in the dark indicated[34]. Diagram in **a** adapted with permission from ref. 3, Springer Nature Limited.

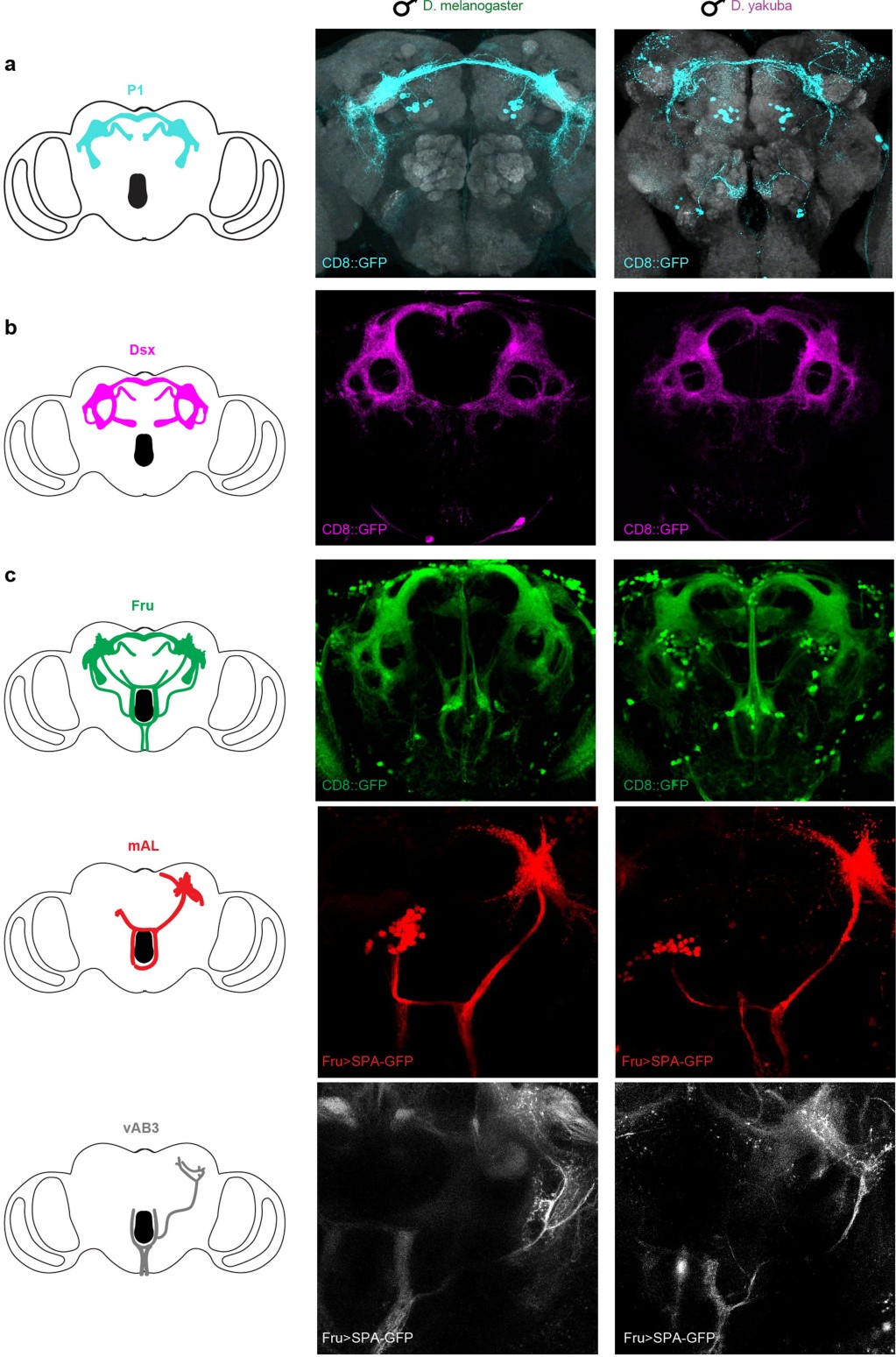

**Extended Data Fig. 10 | Overall anatomical conservation of the Fru+, Dsx+, and ascending circuitry. a**, Unmasked splitP1 > CD8::GFP expression in the brains of *D. melanogaster* (left) and *D. yakuba* (right) males. Stained for GFP (cyan) and neuropil counterstain (gray). **b**, Dsx>CD8::GFP (magenta) expression in the brains of *D. melanogaster* (left) and *D. yakuba* (right) males. **c**, (Top) Fru>CD8::GFP (green) expression, as in (b). (Middle and bottom) Photoactivation labeling of mAL (red) and vAB3 (grey), respectively, using Fru>SPA-GFP. All images are dorsal side up.

# Reporting Summary

## Statistics

For all statistical analyses, confirm that the following items are present in the figure legend, table legend, main text, or Methods section.

| n/a | Confirmed | |
|---|---|---|
| ☐ | ☒ | The exact sample size (*n*) for each experimental group/condition, given as a discrete number and unit of measurement |
| ☐ | ☒ | A statement on whether measurements were taken from distinct samples or whether the same sample was measured repeatedly |
| ☐ | ☒ | The statistical test(s) used AND whether they are one- or two-sided *Only common tests should be described solely by name; describe more complex techniques in the Methods section.* |
| ☐ | ☒ | A description of all covariates tested |
| ☐ | ☒ | A description of any assumptions or corrections, such as tests of normality and adjustment for multiple comparisons |
| ☐ | ☒ | A full description of the statistical parameters including central tendency (e.g. means) or other basic estimates (e.g. regression coefficient) AND variation (e.g. standard deviation) or associated estimates of uncertainty (e.g. confidence intervals) |
| ☐ | ☒ | For null hypothesis testing, the test statistic (e.g. *F*, *t*, *r*) with confidence intervals, effect sizes, degrees of freedom and *P* value noted *Give P values as exact values whenever suitable.* |
| ☒ | ☐ | For Bayesian analysis, information on the choice of priors and Markov chain Monte Carlo settings |
| ☒ | ☐ | For hierarchical and complex designs, identification of the appropriate level for tests and full reporting of outcomes |
| ☐ | ☒ | Estimates of effect sizes (e.g. Cohen's *d*, Pearson's *r*), indicating how they were calculated |

*Our web collection on statistics for biologists contains articles on many of the points above.*

## Software and code

Policy information about availability of computer code

| Data collection | Behavioral data was collected using FlyTracker machine vision software (Caltech) and Flycapture2 Software(2.13.3.61) Development Kit (FLIR). Imagine data was recorded using PrairieView (5.5) software (Bruker). Fluorescence time-series were extracted using FIJI (2.14.0/1.54f, ImageJ, NIH). |
|---|---|
| Data analysis | Data was analyzed using Matlab 2022a, Prism 9, JAABA (0.5.0) machine learning sorftware (Janelia). |

For manuscripts utilizing custom algorithms or software that are central to the research but not yet described in published literature, software must be made available to editors and reviewers. We strongly encourage code deposition in a community repository (e.g. GitHub). See the Nature Portfolio guidelines for submitting code & software for further information.

## Data

Policy information about availability of data

All manuscripts must include a data availability statement. This statement should provide the following information, where applicable:
- Accession codes, unique identifiers, or web links for publicly available datasets
- A description of any restrictions on data availability
- For clinical datasets or third party data, please ensure that the statement adheres to our policy

Source data for all figures is available as supplementary material. All raw data underlying this study are available upon request from the corresponding author.

# Research involving human participants, their data, or biological material

Policy information about studies with [human participants or human data](). See also policy information about [sex, gender (identity/presentation), and sexual orientation]() and [race, ethnicity and racism]().

| | |
|---|---|
| Reporting on sex and gender | n/a |
| Reporting on race, ethnicity, or other socially relevant groupings | n/a |
| Population characteristics | n/a |
| Recruitment | n/a |
| Ethics oversight | n/a |

Note that full information on the approval of the study protocol must also be provided in the manuscript.

# Field-specific reporting

Please select the one below that is the best fit for your research. If you are not sure, read the appropriate sections before making your selection.

☒ Life sciences　　☐ Behavioural & social sciences　　☐ Ecological, evolutionary & environmental sciences

For a reference copy of the document with all sections, see [nature.com/documents/nr-reporting-summary-flat.pdf]()

# Life sciences study design

All studies must disclose on these points even when the disclosure is negative.

| | |
|---|---|
| Sample size | Preliminary experiments were used to assess variance. Sample sizes were not predetermined but kept consistent within comparison groups and comparable to other studies. Sample sizes are consistent with other recent and comparable studies: Sten et al, 2021, Chiu et al 2021, Roemschied et al, 2023. |
| Data exclusions | No data were excluded with one exception. For the D. yakuba P1 imaging, "splitP1" (71G01-AD 15A01-DBD intersection) had to be used rather than 71G01-Gal4 because this line was weak and bleached before an experiment could be completed. These splitP1 animals showed inter-animal variability in responses. Only ~1 in 5 (5/27) animals exhibited P1 responses to conspecifics, though in responding animals the responses were uniform and predictably evoked each time a male tapped a conspecific female (as shown in Fig. 2). We discovered this likely resulted from stochastic labeling of Fru- Dsx+ P1 subset. Specifically, splitP1 predominantly labels Fru+ P1 neurons, and labels 2-3 Fru- Dsx+ neurons in only some animals, as previously found in D. melanogaster57. Indeed, when we subsequently imaged Dsx∩P1 (71G01-DBD dsx-AD intersection) no inter-animal variability was observed and all animals responded to conspecifics with each tap. The same was found to be true when imaging for all Dsx+ projections in the LPC. Therefore, for simplicity and understanding we have presented have excluded the non-responsive D. yakuba splitP1 males in Fig. 2b. |
| Replication | All experiments were replicated using independent biological replicates over the course of days and typically across multiple months. All experiments were replicated across at minimum 4 animals. In cases where representative traces are depicted, they are shown alongside summary statistics for the entire dataset, typically 4-25 animals, depending on the experiment. All attempts at replication were successful. |
| Randomization | The order of female targets presented to males during imaging experiments was randomized each time an experiment was performed. Behavior experiments were loaded and analyzed by different experimenters. Comparison groups were interleaved on the same day in a order randomized by the loader and unknown to the analyzer until after analysis. |
| Blinding | All manually scored behavioral experiments were analyzed by an observer blind to the genotype, species, or treatment. The experimenter was not blind to male of female identities during functional imaging experiments. |

# Reporting for specific materials, systems and methods

We require information from authors about some types of materials, experimental systems and methods used in many studies. Here, indicate whether each material, system or method listed is relevant to your study. If you are not sure if a list item applies to your research, read the appropriate section before selecting a response.

## Materials & experimental systems

| n/a | Involved in the study |
|---|---|
| ☐ | ☒ Antibodies |
| ☒ | ☐ Eukaryotic cell lines |
| ☒ | ☐ Palaeontology and archaeology |
| ☐ | ☒ Animals and other organisms |
| ☒ | ☐ Clinical data |
| ☒ | ☐ Dual use research of concern |
| ☒ | ☐ Plants |

## Methods

| n/a | Involved in the study |
|---|---|
| ☒ | ☐ ChIP-seq |
| ☒ | ☐ Flow cytometry |
| ☒ | ☐ MRI-based neuroimaging |

## Antibodies

| | |
|---|---|
| Antibodies used | Primary antibodies: Chicken anti-GFP (Abcam ab13970), Mouse anti-brp (Developmental Studies Hybridoma Bank nc82), Rabbit anti-FruM (generated for this study by YenZyme against a synthesized peptide: HYAALDLQTPHKRNIETDV70), and Guinea pig anti-FruM, (gift from D. Yamamoto, Tohoku University).<br>Secondary Alexa Fluor antibodies were purchase from Life Technologies and used as dilution of 1/500. Secondary antibodies used were AF555 goat anti-Rat (A21434, Invitrogen), AF555 goat anti-Rabbit (A-21428, Invitrogen) AF647 goat anti- mouse (A-21235, Invitrogen), AF488 goat anti-chicken (A-11039, Invitrogen) and AF488 goat anti-rabbit (A11034, Invitrogen). |
| Validation | Chicken Polyclonal GFP antibody. Validated in WB, ICC/IF and tested in Tag - Aequorea victoria samples. Cited in 2531 publications.<br>Mouse anti-brp. Validated in IHC, IHC-IF, ICC, and IF. Cited in 1023 publications. Validation described in detail on the manufacturer's website.<br>For anti-FruM, this immunizing peptide was previously validate in Kimura et al, 2005 Nature. |

## Animals and other research organisms

Policy information about studies involving animals; ARRIVE guidelines recommended for reporting animal research, and Sex and Gender in Research

| | |
|---|---|
| Laboratory animals | D. yakuba Wildtype (Ivory Coast)<br>D. melanogaster Wildtype (Canton S)<br>D. simulans Wildtype<br>D. erecta Wildtype (14021-0224.01)<br>D. eugracilis Wildtype (SHL12)<br>D. ananassae Wildtype (14024-0371.34)<br>All experimental animals were tested on days 4-7 post-eclosion. |
| Wild animals | This study does not involve wild animals. |
| Reporting on sex | The courtship behaviors described in this studied are sexually dimorphic and exclusive to male Drosophila |
| Field-collected samples | This study does not involve field-collected samples. |
| Ethics oversight | No ethical guidance or approval required. |

Note that full information on the approval of the study protocol must also be provided in the manuscript.

## Plants

| | |
|---|---|
| Seed stocks | *Report on the source of all seed stocks or other plant material used. If applicable, state the seed stock centre and catalogue number. If plant specimens were collected from the field, describe the collection location, date and sampling procedures.* |
| Novel plant genotypes | *Describe the methods by which all novel plant genotypes were produced. This includes those generated by transgenic approaches, gene editing, chemical/radiation-based mutagenesis and hybridization. For transgenic lines, describe the transformation method, the number of independent lines analyzed and the generation upon which experiments were performed. For gene-edited lines, describe the editor used, the endogenous sequence targeted for editing, the targeting guide RNA sequence (if applicable) and how the editor was applied.* |
| Authentication | *Describe any authentication procedures for each seed stock used or novel genotype generated. Describe any experiments used to assess the effect of a mutation and, where applicable, how potential secondary effects (e.g. second site T-DNA insertions, mosiacism, off-target gene editing) were examined.* |

