## [Peer Review File · Nature]

Manuscript Title: A modular circuit architecture coordinates the evolution of mating strategies

Reviewer Comments & Author Rebuttals

Reviewer Reports on the Initial Version:

Referees' comments:

Referee #1 (Remarks to the Author):

Little is known about the neural substrates that underlie behavioral evolution. *Drosophila* pheromones and sexual behavior offer a great system for investigating this question, given the deep knowledge about neural circuits that control mating in *Drosophila melanogaster*, and the rapid divergence in pheromones and their processing circuits among closely related species to ensure species-specific mate recognition. For productive mating, males need to discriminate females of their own species from females of other species, as well as from males of any species. For *D. melanogaster*, this is facilitated by a sexual dimorphic pheromone (a hydrocarbon called 7,11-HD), which is produced by females but not males and enables males to taste females and mate in the dark. However, *D. simulans* and *D. yakuba*, two species closely related to *D. melanogaster*, do not produce 7,11-HD. *D. simulans* males thus have difficulty following the females in close range in the dark, and require vision for successful mating (in the light). However, *D. yakuba* males could still follow females in the dark despite their females not producing 7,11-HD. Having made these observations on the behavior, Coleman and colleagues tried to figure out how *D. yakuba* males could do it.

Through heroic effort, including generating multiple new genetically modified *D. yakuba* to knock-out genes or knock-in drivers to access specific cell types, Coleman et al. found two properties in *D. yakuba* that could contribute to the observed behavioral results. First, taste sensory neurons that detect 7,11-HD in *D. melanogaster* (and express *ppk25*) are activated by 7-T in *D. yakuba*, a hydrocarbon produced by males of all *Drosophila* species and females of *D. yakuba* (and *D. simulans*) that have lost the enzymes to convert 7-T to 7,11-HD. Thus, 7-T could now serve as a pheromone to promote mating of *D. yakuba* males towards females of their own species in the dark. Indeed, Coleman et al. provided behavioral data to support this. Second, through genetic dissection of P1 neurons, the “master regulators” of sexual behavior in *Drosophila* that integrate multiple sensory inputs and promote male courtship, Coleman et al. found that two subpopulations of P1 neurons expressing two transcription factors implicated in regulating sexual behavior, *Fru* and *Dsx*, have distinct properties: only *Dsx+* P1 neurons but not *Fru+* P1 neurons are activated by tasting female *D. yakuba* (presumably through 7-T activated taste sensory neurons).

Overall, the study addresses an important problem and is technically well executed. The rapid divergence of pheromone detection in sensory neurons in closely related species is fascinating. The story will be more satisfying if the authors could provide clues to one or more of the following

questions: 1) how do ppk25+ neurons selectively transmit signals to Dsx+ but not Fru+ P1 neurons? 2) how do *D. yakuba* males discriminate male and female *D. yakuba*, both having 7-T on their bodies that could activate Dsx+ P1 neurons to promote mating in the dark? 3) how do *D. yakuba* males discriminate against females (and males) of other species that possess 7-T? These questions may seem open-ended, but at least some specific experiments can be performed to address question #2. Does a yakuba male pursue another yakuba male in the dark (experiments in Fig. 1)? Are yakuba Dsx+ P1 neurons or ppk23+ neurons activated by tasting male yakuba (Fig. 2, 3)? If not, why not? Does cVA play a role in discriminating male and female yakuba in the above experiments?

Other issues:

On Fru+ vs. Dsx+ P1 neurons: it seems that Dsx+ neurons also include all Fru+ neurons (line 301). If so, the nomenclature is misleading. For example, the effect of genetic silencing Dsx+ neurons could also be interpreted by Fru+ and Fru- neurons playing redundant functions (rather than implicitly assigning the function to Dsx+Fru- neurons).

Fig. 5 is not very clear and is made more confusing with the issue raised above about the relationship between Fru+ and Dsx+ P1 neurons.

The authors tested whether the distinction of Fru+ vs. Dsx+ P1 neurons in *D. yakuba* also extends to *D. melanogaster*. They found that *D. melanogaster* Fru+ P1 neurons responded more strongly to taste of *D. erecta* females. Does Dsx+ P1 neurons also include Fru+ P1 neurons? If so, would Dsx+Fru- P1 neurons be even less responsive than Dsx+Fru+ P1 neurons given that the data on Dsx+ P1 neurons represent a mixture? In any event, showing that the distinction between Fru+ vs. Dsx+ P1 neurons is a conserved feature (even though their tuning across species may differ) is an important piece of evidence supporting the “modular” nature of the circuit, so a better characterization of the difference between Fru+ vs. Dsx+ P1 neurons in different species will provide stronger support to the authors’ interpretation.

Minor points:

Line 190: insert “other” before “heterospecific”

Line 281: insert “of” after “expression”

Fig. 3: it’s confusing that shade vs. non-shade is used to discriminate conspecific vs. heterospecific in panel h, but to discriminate dark vs. light in other panels.

Referee #2 (Remarks to the Author):

The manuscript by Coleman et al. is an especially impressive example of the generation of genetic tools in non-model species, in which such resources are not readily available, to address non-trivial mechanistic questions around the evolution of mate recognition circuits. Capitalizing on these novel

tools, along with automated behavioral assays and in vivo calcium imaging approaches, this tour-de-force study looks comparatively across *Drosophila* species which vary in their pheromone dimorphisms and courtship strategies. The study reveals two levels at which male courtship circuitry is modulated to allow them to respond to rapidly evolving female pheromonal cues and direct species-specific courtship. They identify changes at the peripheral sensory level (namely, ppk23+ppk25+ neurons) as well as how these signals from the periphery are then differentially conveyed to central courtship-promoting neurons (Fru or Dsx subpopulations of pC1/P1). Selective activation of specialized subsets of ppk and P1 neurons allows the sexually ambiguous pheromone 7-T to act as a sexual arousal cue in *D. yakuba* males, driving courtship towards conspecific females in the dark. This sets *D. yakuba* apart from *D. simulans* in their pheromone detection and integration, allowing them to occupy a specialized sensory niche for mating.

The authors propose a modular circuit configuration through a comparative analysis of sensory and central courtship nodes across *D. yakuba*, *D. simulans*, and *D. melanogaster*. This circuit model allows independent sensory inputs to connect to master courtship neurons, steering divergent courtship strategies across sensory environments. Thus, a few critical switches in an otherwise largely conserved neural circuit may suffice to create behavioral diversity across the *Drosophila* genus. In their previous work, the authors proposed that P1 neurons, functioning as hubs for sensory convergence that trigger the male courtship program, could function as evolutionary hot spots for mate discrimination, enabling subtle neural circuit modifications to have large behavioral effects (Seeholzer et al., *Nature*, 2018). Looking at an expanded set of *Drosophila* species, the evidence in this paper further places P1 as key neural substrates targeted by evolution, emphasizing their coordinated action with peripheral sensory systems. Therefore, the findings presented in this manuscript mark an important advancement in our understanding of the evolution of mate discrimination.

The paper is very well written, and the conclusions are generally supported by the data. However, I'd like to raise several points for consideration, presented in no specific order:

Major

1. Figure 3b shows that *D. yakuba* males display shorter bout lengths in the light compared with the dark (Figure 3d, f), which is surprising. Looking into the methods, the authors detail that a food substrate was used for dark courtship assays but not light ones and hence, males show shorter bout lengths in the light condition. Is the presence of the food absolutely required for *D. yakuba* males to court in the dark? This should be acknowledged in the main text. In addition, poor courtship bout lengths in wildtype flies might mask effects from disrupted pheromonal signaling in ppk23/ppk25 mutants (Figure 3b-f). Finally, is there a potentially confounding effect introduced from adding food in the dark condition? Different species may respond differently to the presence of a food substrate. The presence of food is known to affect female receptivity and male courtship in *D. melanogaster* (Gorter et al., *sci rep*, 2016, Grosjean et al., *Nature*, 2011) as well as change the potentiation of male pheromones (Das et al., *PNAS*, 2017).
2. The authors mention in the methods that due to video quality only a small subset of the original data could undergo automated analysis. Removal of these data points has given a small n for behavioral experiments. For instance, there seems to be variability between behavior of the 6 courting pairs in Fig

- 1b. Hence, increasing the n of flies will allow for stronger conclusions.
3. There are instances of discrepancies between trends seen in automated analyses in the main figures and manual analyses provided in the extended data figures. For example, the authors state that *D.yakuba* ppk23 mutant males show reduced pursuit of conspecifics in the dark (Figure 3d). However, extended data figure 3c looking at % time courting contradicts this. Similarly, in Figure 3f authors state *D. yakuba* ppk25 mutant males display diminished courtship towards conspecifics in the dark; however, extended data figure 4b % time courting contradicts this. This should be either acknowledged in the text or resolved.
4. For Figure 4e, the authors show that responses in the *Dsx* subpopulation are lost in the ppk23 mutant background for *D.yakuba*. This experiment should be repeated in a ppk25 mutant background since ppk23+ppk25+ neurons were identified by the authors as sensing 7-t as an arousal signal (Figure 3).
5. In Figure 4a it is shown that the *Fru* intersection in *D.yakuba* does not elicit reliable wing extension on optogenetic activation. This contrasts with what is seen in *D. melanogaster* (Extended data Figure 6b). Is it possible that non-targeted neurons crucial for eliciting courtship are missing? Since these data are important for supporting the proposed model, could the authors provide movies for the courtship behavior of the *Fru* intersection.
6. Throughout the manuscript, measurements used to represent male courtship behaviors are inconsistent between experiments e.g., average bout length (Figure 2 and 3), % time courting, pursuing, and extending wing (Figure 4). It is understood that perhaps this change in behavioral quantification is due to the authors being interested in either sustained courtship (bout length) or total courtship levels (% courtship). However, the inconsistency in behavioral parameters is at times difficult to follow and sometimes even contradictory to the authors claims (see previous point 3). More explanation of behavioral parameters chosen, and more consistency could be beneficial.
7. For Figure 3e the authors need to directly compare the wildtype *D. yakuba* male results (top panel) to the ppk23 and ppk25 mutants (bottom 2 panels) to make statistical comparisons.
8. The authors should discuss/integrate the model proposed in this manuscript with previous work (Seeholzer et al., *Nature*, 2018, Kallman, Kim and Scott, *eLife*, 2015). For instance, could vAB3/mAL neurons differentially target the subpopulations of P1, changing overall balance/P1 activation?

Minor:

9. P1 is functionally heterogenous and has been defined differently in the literature (e.g. P1, P1a, pC1, etc.), creating confusion (issue described in Asahina et al., *Curr Opin Physiol*, 2018). A better description of how the pC1/P1 populations are named and defined should be included (Figure 4). In the main text there is only reference to 'P1' however, pC1 is mentioned in the extended data figures. The nomenclature should be more clearly explained, for instance describing the *Dsx*+ pC1 population which overlaps with the *Fru*+ P1 cluster at the beginning of the section related to Figure 4.
10. The authors use an optogenetic stimulation protocol of 2 mins dim white light followed by 2 mins red light (627nm), alternated over 10 minutes. However, P1 activation can be sustained for minutes after stimulation (Hoopfer et al., *eLife*, 2015). Raster plots in extended data Figure 7 show that, except between the first 'light off' to 'light on' period, there seems not to be much difference between light on and off periods. In addition, it appears that more males exhibited wing extension after the light stim

period in Figure 4a. The authors should consider doing separate experiments for “light on” and “light off” conditions to avoid confounding effects.

11. The authors have tested for normality in order to apply the appropriate statistical tests however, when non-parametric tests such as Mann Whitney were applied the mean and SEM were plotted. For non-normal data it is more appropriate to use the median.

12. Courtship assays in Figure 1b were recorded for a total of 10 minutes. Was any consideration given to the fact that courtship/copulation latencies vary between different species (Khallaf et al., Nature communications, 2021).

13. The IFD traces in Fig 1b, showing male proximity to the female over the 10-minute courtship assay, are for a single representative pair only. To represent the whole data set, a population trace taking into account all 6 pairs may be more informative.

14. For main and extended data Figure 1, $IFD < 8\text{mm}$ is stated as the threshold used for close proximity. However, no details of how this was decided upon is included. Was this chosen using previous studies or was this the distance threshold that fitted best with the manually scored data (extended data fig 1)?

15. Since *oe-* flies are still able to produce pheromones (e.g., some methyl pheromones, Dweck et al., PNAS, 2015) the authors should add a wildtype female *D. yakuba* control to Figure 1d to account for any contributions of non-ablated pheromones.

16. Is it possible to know for Figure 3a how many neurons make up the *ppk23+/ppk25+* and *ppk23+/ppk25-* subpopulations in *D. yakuba*? Extended data Figure 3d provides total *ppk23* neurons only.

17. The authors may wish to acknowledge in the text (line 328/9) that it has been shown previously that male *D. melanogaster* court *D. erecta* females (Billeter et al., Nature, 2009).

18. In both Figure 3e and Figure 4b it is shown that *D. yakuba* males respond to *D. simulans* females due to the presence of 7-T. Therefore, could the authors discuss how male *D. yakuba* avoid or minimize courtship towards heterogenous females that also carry 7-T. For instance, how does this fit with the authors' current model of *D. yakuba*'s use of 7-T to direct courtship to conspecifics?

19. Comments on Figure presentation:

- Figure 3a: the right panel is not referred to in the text.

- Figure 3h: (i) The authors should add *ppk23* and *ppk25* into the schematic (ii) The grey and white shaded boxes are misleading as these are previously used to represent dark and light conditions (iii) It would be helpful if the authors could make clear on the model schematic what is previous work and what is new data added from this manuscript, including the relevant references in the figure legend (iv) Finally, some of the model may be oversimplified and therefore slightly misleading, for example from the schematic it appears as though 7,11-HD directly inhibits P1 in *D. simulans* males which according to previous work from the same group isn't the case (Seeholzer et al., Nature, 2018).

- Figure 3b,c,d,f: the scale of the y-axis (up to 600 seconds), compresses all the data at the bottom of the axis. This makes it very difficult to see the spread of data and differences between data sets.

- Figure 4i: it would be helpful to have the mean bars in a different color to black, so they are easier to see.

- Extended data Figure 8: it would be helpful to label the relevant species on the phylogeny circle.

Referee #3 (Remarks to the Author):

In their manuscript entitled “A modular circuit architecture coordinates the diversification of courtship strategies in *Drosophila*”, Coleman et al. make use of pheromone mating circuits in multiple *Drosophila* species to investigate evolutionary mechanisms leading to behavioral diversification. They nicely show that males of the *Drosophila* species *D. yakuba*, with a monomorphic pheromone profile, use this pheromone as excitatory mating signal. Peripheral pheromone sensing cells, which lead to aversion in *D. melanogaster* with a dimorphic pheromone profile, switched to induce attractive responses in *D. yakuba*. In addition, they show that functional subdivision in central brain circuits lead to specific responses in *Dsx+* cells in *D. yakuba* triggering a mating response. Overall, they provide evidence that tuning changes at the periphery and modified central processing shape evolution of mating behaviors. Conceptually, they propose that the modular organization at each layer facilitates adaptation and behavioral evolution.

This work nicely illustrates how the transfer of neurogenetic tools from *D. melanogaster* to other species with derived behaviors can deliver functional insights into basic principles of neural circuit evolution. I am impressed by the wealth of data in this manuscript, the degree of experimental details in a non-standard model system and the sophisticated experimental approaches.

I consider the results presented here of immediate interest to a broad audience working in (behavioral) neuroscience and evolution and I think it merits publication in *Nature*. The described evolutionary changes at the periphery and in central brain circuits between species represent a fascinating example of how input-output relationships can be modified and will serve as reference point for future studies in other systems. The work nicely builds on a previous, seminal study from the same lab (Seeholzer, *Nature* 2018) and by smartly choosing selected model species, it identifies novel ways (gain of peripheral excitation, selective activation of P1 subclusters) how sensory system and its central processing centers can evolve.

The manuscript is very well written and follows a nice flow and narrative. The presented results support the conclusions drawn by the authors and the statistical evaluation is convincing. However, there are a few points that I would like the authors to address prior to publication:

Major:

Figure 1: It is not clear to me where in the manuscript IFD-tracking and automatic scoring of bout lengths was performed and where courtship behavior was evaluated manually. As shown in Ext. Data Figure 1b, automatic scoring performs very poorly in scoring average bout length. How data for each figure was processed should be made clearer e.g. in the figure legends.

In this line, I am wondering if data shown in Figure 3b-f was manually scored? Could the authors show the other parameters measured (in addition to bout length) for Figure 3b? It is surprising to me that *D. yakuba* *ppk23* mutants do not show a phenotype in light. Does this indicate an additional role for visual information? In *D. melanogaster*, *ppk23* mutants reduce mating in light (e.g. <https://journals.plos.org/plosgenetics/article?id=10.1371/journal.pgen.1002587>).

One direct consequence of the proposed model would be that *D. yakuba* shows an increased degree of male-male courtship as there is no strong cVA response in the ppk23 population and 7T is an attractive signal. Do the authors observe this? What could prevent this type of interaction?

Related: Line 365: "In *D. yakuba* and other monomorphic species, the conserved male-specific volatile pheromone cis-Vaccenyl acetate (cVA) likely performs this role as it has been to prevent intrasexual courtship, likely through direct inhibitory pathways that impinge on P1 neurons."

But it does not activate ppk23 neurons in *Dyak* (Extended Data Fig. 3). How?

Figure 3f: Why was the the *Dyak*-*Dmel* experiment here performed in light conditions (for ppk25 mutants)? It would be more consistent to perform these experiments in the dark to compare directly to the neighboring data.

Figure 3 and Ext. Data Fig 5: Could the authors please indicate when data was re-plotted across panels?

Figure 2: Left: Here the authors just show a small overview staining of the respective brain labels in multiple species. Could they locate the LPC and indicate where the region of interest was placed for quantification of their calcium imaging data?

Also, I would suggest adding a few words in the material and methods on how the quantification of Calcium imaging data was done (it is apparent that signals are stronger in *Dmel* than in the other species probably due to variation in expression levels of transgenes?!).

Extended Data Fig. 5b-e: Here it is not clear which neurons are counted. Could the authors provide representative pictures of their *Dsx* and *Fru* stainings to facilitate recapitulation of their data for the reader.

Extended Data Fig. 7: Here, the *D. yakuba* data does not look very convincing to me. Could the authors clarify which pre and post-stimulation periods were taken into account for quantification. To me it seems there is a strong initial arousal response in the first activation window but activity remains high in the following OFF phase.

Figure 4e: I was wondering why the authors did not use the ppk25+ mutants for their experiment to more directly support their claim (does the chromosomal location of ppk25 prevent this experiment)? If it is technically possible, could the authors show *dsx*>GCaMP responses in ppk25 mutants?

Related: Line 348: "Exemplifying this evolutionary flexibility, we show that in *D. yakuba* males, one subset of P1 neurons uniquely integrates from the Ppk25+ peripheral sensory neurons with enhanced sensitivity to 7-T. ". This is not really shown.

Minor:

Figure 1 and Extended Data Figure 1: Please add information that the red line indicates courtship initiation

Could you clarify in the figure legend what is the time frame analyzed in b? Experiments were run for 3h

-> the heatmap shows a 10min time window, the histogram the full 3h?

Extended Data Fig 1 b,d: It is not clear what is the sample size in the experiments and what the complete sample refers to.

Line 166/167: "Minor differences in the arborization patterns of P1 neurons were apparent across species. Yet despite their morphological variation,...". Given that different genetic reagents were used in each species (e.g. GCaMP vs. CD8-GFP, split- reagents vs. single driver), I am not sure if one can conclude morphological variation rather than technical reasons. I would suggest removing of this phrase.

Figure 2b: Is this the average tap-evoked response of traces shown to the left or of all experiments (shown at the right)?

Please update Reference 45.

Author Rebuttals to Initial Comments:

A modular circuit architecture coordinates the diversification of courtship strategies in *Drosophila* 2023-10-18659

Referee #1 (Remarks to the Author):

Little is known about the neural substrates that underlie behavioral evolution. *Drosophila* pheromones and sexual behavior offer a great system for investigating this question, given the deep knowledge about neural circuits that control mating in *Drosophila melanogaster*, and the rapid divergence in pheromones and their processing circuits among closely related species to ensure species-specific mate recognition. For productive mating, males need to discriminate females of their own species from females of other species, as well as from males of any species. For *D. melanogaster*, this is facilitated by a sexual dimorphic pheromone (a hydrocarbon called 7,11-HD), which is produced by females but not males and enables males to taste females and mate in the dark. However, *D. simulans* and *D. yakuba*, two species closely related to *D. melanogaster*, do not produce 7,11-HD. *D. simulans* males thus have difficulty following the females in close range in the dark, and require vision for successful mating (in the light). However, *D. yakuba* males could still follow females in the dark despite their females not producing 7,11-HD. Having made these observations on the behavior, Coleman and colleagues tried to figure out how *D. yakuba* males could do it.

Through heroic effort, including generating multiple new genetically modified *D. yakuba* to knock-out genes or knock-in drivers to access specific cell types, Coleman et al. found two properties in *D. yakuba* that could contribute to the observed behavioral results. First, taste sensory neurons that detect 7,11-HD in *D. melanogaster* (and express *ppk25*) are activated by 7-T in *D. yakuba*, a hydrocarbon produced by males of all *Drosophila* species and females of *D. yakuba* (and *D. simulans*) that have lost the enzymes to convert 7-T to 7,11-HD. Thus, 7-T could now serve as a pheromone to promote mating of *D. yakuba* males towards females of their own species in the dark. Indeed, Coleman et al. provided behavioral data to support this. Second, through genetic dissection of P1 neurons, the “master regulators” of sexual behavior in *Drosophila* that integrate multiple sensory inputs and promote male courtship, Coleman et al. found that two subpopulations of P1 neurons expressing two transcription factors implicated in regulating sexual behavior, *Fru* and *Dsx*, have distinct properties: only *Dsx*+ P1 neurons but not *Fru*+ P1 neurons are activated by tasting female *D. yakuba* (presumably through 7-T activated taste sensory neurons).

Overall, the study addresses an important problem and is technically well executed. The rapid divergence of pheromone detection in sensory neurons in closely related species is fascinating.

We thank the reviewer for their enthusiasm and their appreciation of our efforts to overcome the challenges of neurogenetic manipulations in non-model species. We are also very grateful for the reviewer's poignant and helpful suggestions, which we have addressed below.

The story will be more satisfying if the authors could provide clues to one or more of the following questions:

1) how do *ppk25*+ neurons selectively transmit signals to *Dsx*+ but not *Fru*+ P1 neurons?

The reviewer highlights the importance of linking the detection of 7-T by *Ppk25*+ neurons to the selective responses we observe in just the *Dsx*+ subset of P1 neurons in *D. yakuba* males. While we generated transgenic lines to label second and third order pheromone processing neurons in *D. yakuba*—for example, *vAB3* and *mAL* neurons—the genetic drivers that label these populations in *D. melanogaster* were unfortunately too weakly expressed to draw confident conclusions. Therefore, as an alternate inroad to this question we imaged P1 neuron responses in *ppk25* mutant males (now shown in **Fig. 4e**; previously displayed *ppk23* mutant responses have been moved to **Extended Data Fig. 3g**), which confirmed that the

distinct sensitivity of the $dsx \cap P1$ neurons to the *D. yakuba* pheromone is relayed via the Ppk25+ sensory neurons.

2) how do *D. yakuba* males discriminate male and female *D. yakuba*, both having 7-T on their bodies that could activate $Dsx+ P1$ neurons to promote mating in the dark?

The prevalence of sexually monomorphic cuticular pheromones across many members of the *Drosophila* genus poses an interesting challenge to sex discrimination, especially given that in *D. yakuba*, P1 neurons are excited by 7-T, which is produced by both males and females. We have addressed this important question through data presented in a new supplementary figure (**Ext. Data Fig. 8**; now discussed in a new section in the manuscript, lines 350-369). We show that while *D. yakuba* males are less effective than *D. melanogaster* males in discriminating between conspecific males and females, they still exhibit a clear preference for courting females. This indicates that despite *D. yakuba* males and females sharing the same attractive pheromone, 7-T, males of this species have evolved sensory mechanisms to suppress inappropriate courtship pursuit of males. One obvious mechanism we considered is that *D. yakuba* males produce the male-specific volatile pheromone 11-cis-Vaccenyl acetate (cVA), which has been proposed to suppress male courtship in many species. Indeed, we show that perfuming *D. yakuba* females with cVA has a similar inhibitory effect on male courtship as perfuming them with the *D. melanogaster* pheromone, 7,11-HD (**Ext. Data Fig. 8d**). We therefore generated a mutant for the canonical cVA receptor, *Or67d*, in *D. yakuba*. Unexpectedly, however, *Or67d* mutant males did not display enhanced intrasexual courtship (**Ext. Data Fig. 8b,c**), suggesting that cVA likely acts through another, potentially redundant, receptor in this species (e.g., *Or65a*; Naters & Carlson, 2007). Nevertheless, despite the potential complexity of peripheral cVA detection, we demonstrated that cVA dampens the responses of $dsx \cap P1$ neurons, by presenting males with *D. yakuba* females perfumed with cVA (**Ext Data Fig. 8e**). The inhibitory role of cVA at the level of the P1 neurons thus recapitulates its suppressive effect on courtship observed at the behavioral level.

In addition, we also considered that males may avoid intrasexual courtship through behavioral countersignaling in the form of bilateral wing flicks that produce an agonistic song (Hindmarsh Sten, Li, et al., 2023). Indeed, either ‘muting’ a male (by surgically removing their wings) or deafening a male (by removing their arista) increases male-male courtship suggesting this acoustic countersignaling normally suppresses male arousal. Taken together, *D. yakuba* males appear to rely on multimodal signals for sex discrimination and overcome the inherent challenge of using a sexually monomorphic excitatory pheromone to become aroused.

3) how do *D. yakuba* males discriminate against females (and males) of other species that possess 7-T?

Females of many species within the *Drosophila* genus have converged on using 7-T as a potential sex pheromone (e.g. Khallaf et al. *bioRxiv*, 2024). Given the critical role that cuticular pheromones play in mate recognition, in our initial submission we provided evidence that *D. yakuba* males may not be able to distinguish between *D. yakuba* and *D. simulans* females that both carry 7-T. To emphasize this point, we have now moved data from **Extended Data Fig. 1** to **Fig. 1c** which demonstrates that *D. yakuba* males court *D. simulans* females and *D. simulans* males court *D. yakuba* females with comparable vigor to their courtship of conspecific females, reinforcing that they cannot discriminate between these females due to their shared pheromone profile. Additionally, we have now added preference assays to **Ext. Data Fig. 8a** to show that *D. yakuba* males are unable to discriminate between conspecific and *D. simulans* females. Consistent with this, *D. yakuba* and *D. simulans* females evoke equivalent responses in the P1 neurons of *D. yakuba* males (**Fig. 2b** and **Fig. 4b**). Together, these observations reinforce the critical role that female pheromones play in *Drosophila* mate discrimination.

As the reviewer astutely hints, while such indiscriminate pursuit of heterospecific females that carry 7-T may appear to be a maladaptive strategy, it is important to consider the natural histories of *D. yakuba* and *D. simulans* which indicate they largely evolved in geographic isolation from one another. Consequently, *D. yakuba* and *D. simulans* males did not need to evolve a mechanism to discriminate between females that carry 7-T (Lachaise & Silvain, 2004; Dean & Ballard, 2004; Pool & Aquadro, 2006). Moreover, in the rare locales where 7-T producing species do overlap, males do court heterospecific females, but females discriminate against heterospecific males, likely due to differences in their song production (Coyne et al., 2005).

The promiscuous courtship towards all females carrying 7-T further points to the importance of additional sensory mechanisms for sex-discrimination in these sexually monomorphic species. As noted above, discrimination against heterospecific males that produce 7-T is likely mediated by the male-specific volatile pheromone cVA and behavioral countersignaling.

These questions may seem open-ended, but at least some specific experiments can be performed to address question #2. Does a *yakuba* male pursue another *yakuba* male in the dark (experiments in Fig. 1)?

While we did not specifically examine male-male courtship in the dark in *D. yakuba*, we sought to address this question by examining male-male courtship in the light, as detailed above and in **Ext. Data Fig. 8**. Given that male-male courtship occurs at very low rates in the light, we anticipate it would be the same or lower in the dark, where there is even less excitatory drive for courtship.

Are *yakuba* *Dsx+* P1 neurons or *ppk23+* neurons activated by tasting male *yakuba* (Fig. 2, 3)? If not, why not? Does cVA play a role in discriminating male and female *yakuba* in the above experiments? Based on the reviewer's questions, we have carried out additional experiments to gain insight into how the chemosensory system of *D. yakuba* has evolved to navigate mate discrimination in the context of a sexually monomorphic excitatory pheromone.

We have now included the responses of both *Ppk23+* sensory neurons (**Fig. 3e**) and P1 neurons (**Fig. 4a, b**) to *D. yakuba* males. Notably, we show that while *Ppk23+* sensory neurons respond to the taste of males, P1 neurons do not. Guided by the reviewer's suggestion, we tested whether the lack of P1 neuron responses to males might be mediated by cVA and found that perfuming of *D. yakuba* females with cVA was indeed sufficient to suppress *dsx*∩P1 neuron responses in *D. yakuba* males (**Ext. Data Fig. 8e**). Thus, the sensory neurons detecting cuticular pheromones respond indistinguishably to the taste of both *D. yakuba* males and females in accord with their sensitivity to 7T. However, P1 neurons, as a site of multisensory integration (Kohatsu et al., 2011; Clowney et al., 2015; Kallman et al., 2015; Zhou et al., 2015; Kohatsu & Yamamoto, 2015; Hindmarsh Sten et al., 2021; Roemschied et al., 2023), appear inhibited by cVA, thereby suppressing courtship toward other males.

Other issues:

On *Fru+* vs. *Dsx+* P1 neurons: it seems that *Dsx+* neurons also include all *Fru+* neurons (line 301). If so, the nomenclature is misleading. For example, the effect of genetic silencing *Dsx+* neurons could also be interpreted by *Fru+* and *Fru-* neurons playing redundant functions (rather than implicitly assigning the function to *Dsx+**Fru-* neurons).

We thank the reviewer for highlighting this important point. We believe that our division of P1 neurons into a *Fru+* and *Dsx+* subset highlights the role that these two master regulatory transcription factors play in demarcating functionally distinct subsets of the P1 population. We appreciate the need for clarity, however, and now have modified the text and include a diagram (**Ext. Data Fig. 5a**; lines 286-298) that

illustrates the relationship between the Dsx+ and Fru+ P1 neuronal classes in our study. **Ext. Data Fig. 5a** shows that all P1 neurons marked by a widely-used genetic driver (71G01) express Dsx and are part of the broader pC1 cluster, with just a subset expressing Fru. Aligned with the anatomic characterization of neurons marked by our intersectional labeling strategy (**Extended Data Fig. 5b-d**), $Fru \cap P1$ labels only Fru+ neurons while $Dsx \cap P1$ labels both Dsx+Fru+ and Dsx+Fru- neurons, a point we now explicitly state in the text (lines 290-295). In addition, in the text, we now only refer to these two populations by the intersectional strategies that label them ($Fru \cap P1$ and $Dsx \cap P1$) to avoid any confusion that a subset of $Dsx \cap P1$ are Fru+.

Importantly, our functional imaging of all Fru+ neurons demonstrates that they are completely insensitive to pheromones in *D. yakuba*, despite also being Dsx+ (**Figure 4**). This observation strongly suggests that the pheromone responses observed in the $Dsx \cap P1$ intersection in *D. yakuba* arise from the Dsx+Fru- population, while the Dsx+Fru+ neurons are not responsive, supporting our conclusion that these P1 subsets display distinct pheromonal tuning.

Nevertheless, to provide additional evidence that Dsx+ and Fru+ subsets of P1 neurons are modular and display distinct pheromonal responses across species, we further explored their relationship in *D. melanogaster*. In this species, our data suggests that the $Fru \cap P1$ subpopulation exhibits elevated responses to the *D. erecta* pheromone, despite that this subset is also Dsx+ and therefore labeled in both the Dsx-Gal4 and $Dsx \cap P1$ genetic drivers. To isolate the responses of just the Dsx+ population, we generated a *fru*-Gal80 insertion in *D. melanogaster*, thereby allowing us to block GCaMP expression in just the Fru+ contingent of Dsx-expressing P1 neurons. In accord with our model that the Dsx+Fru+ and Dsx+Fru- populations are functionally distinct, we find that the *D. erecta* pheromone responses observed in all Dsx+ neurons are strongly attenuated in males carrying the *fru*-Gal80 allele, with the the Dsx+Fru- subset responsive only to the *D. melanogaster* pheromone, 7,11-HD. These results indicate that, in *D. melanogaster* males, sensitivity to the *D. erecta* pheromone indeed appears restricted to just the Fru+ subset of P1 neurons (**Ext. Data Fig. 6c**) further supporting the functional modularity of P1 neuron subsets.

Finally, while silencing of Fru+ P1 neurons ($Fru \cap P1 > Kir$) had no impact on the vigor of courtship in the dark, we acknowledge we cannot conclusively rule out the possibility that the attenuated courtship in the dark observed in *D. yakuba* $Dsx \cap P1 > Kir$ males is due to silencing of the Fru+Dsx+ neuronal subset. We have therefore added this important caveat to our description of **Fig. 4g** in the main text: “*While we cannot exclude the possibility that $Fru \cap P1$ neurons might function redundantly with the Dsx+/Fru- neurons to promote courtship, our functional and behavioral data suggest that Fru+ and Dsx+ P1 subpopulations play distinct roles in sustaining courtship in the dark, where males become reliant on pheromonal feedback.*” (lines 323-326)

Fig. 5 is not very clear and is made more confusing with the issue raised above about the relationship between Fru+ and Dsx+ P1 neurons.

We thank the reviewer for their feedback and have revised this figure to improve clarity, summarizing the data we present that the $Dsx \cap P1+$ and $Fru \cap P1+$ neuron subsets are differentially tuned to pheromones across species, yet all drive male courtship, suggesting the parallel control of sexual arousal by distinct sensory pathways (lines 286-306).

The authors tested whether the distinction of Fru+ vs. Dsx+ P1 neurons in *D. yakuba* also extends to *D. melanogaster*. They found that *D. melanogaster* Fru+ P1 neurons responded more strongly to taste of *D. erecta* females. Does Dsx+ P1 neurons also include Fru+ P1 neurons?

As noted above, we added a diagram to illustrate the nested nature of the Fru+ P1 contingent within the broader Dsx+ population, which is true for both *D. melanogaster* and *D. yakuba* (Ext. Data Fig. 5a).

If so, would Dsx+Fru- P1 neurons be even less responsive than Dsx+Fru+ P1 neurons given that the data on Dsx+ P1 neurons represent a mixture? In any event, showing that the distinction between Fru+ vs. Dsx+ P1 neurons is a conserved feature (even though their tuning across species may differ) is an important piece of evidence supporting the “modular” nature of the circuit, so a better characterization of the difference between Fru+ vs. Dsx+ P1 neurons in different species will provide stronger support to the authors’ interpretation.

Indeed, as described above and exactly as the reviewer suggests, using a fru-Gal80 allele to isolate the functional responses of the Dsx+Fru- and Dsx+Fru+ P1 neuron subsets in *D. melanogaster*, we and confirmed that the Dsx+Fru- P1 neurons display attenuated responses to the *D. erecta* pheromone in comparison to the full Dsx+ population. This data supports the modular nature of P1 neuron subsets, which are distinctly tuned across species.

Minor points:

Line 190: insert “other” before “heterospecific”

Line 281: insert “of” after “expression”

Thank you for catching these, they have all now been fixed.

Fig. 3: it’s confusing that shade vs. non-shade is used to discriminate conspecific vs. heterospecific in panel h, but to discriminate dark vs. light in other panels.

We agree and have modified this figure for clarity as suggested.

Referee #2 (Remarks to the Author):

The manuscript by Coleman et al. is an especially impressive example of the generation of genetic tools in non-model species, in which such resources are not readily available, to address non-trivial mechanistic questions around the evolution of mate recognition circuits. Capitalizing on these novel tools, along with automated behavioral assays and *in vivo* calcium imaging approaches, this tour-de-force study looks comparatively across *Drosophila* species which vary in their pheromone dimorphisms and courtship strategies. The study reveals two levels at which male courtship circuitry is modulated to allow them to respond to rapidly evolving female pheromonal cues and direct species-specific courtship. They identify changes at the peripheral sensory level (namely, ppk23+ppk25+ neurons) as well as how these signals from the periphery are then differentially conveyed to central courtship-promoting neurons (Fru or Dsx subpopulations of pC1/P1). Selective activation of specialized subsets of ppk and P1 neurons allows the sexually ambiguous pheromone 7-T to act as a sexual arousal cue in *D. yakuba* males, driving courtship towards conspecific females in the dark. This sets *D. yakuba* apart from *D. simulans* in their pheromone detection and integration, allowing them to occupy a specialized sensory niche for mating.

The authors propose a modular circuit configuration through a comparative analysis of sensory and central courtship nodes across *D. yakuba*, *D. simulans*, and *D. melanogaster*. This circuit model allows independent sensory inputs to connect to master courtship neurons, steering divergent courtship strategies across sensory environments. Thus, a few critical switches in an otherwise largely conserved neural circuit may suffice to create behavioral diversity across the *Drosophila* genus. In their previous work, the authors proposed that P1 neurons, functioning as hubs for sensory convergence that trigger the male courtship program, could function as evolutionary hot spots for mate discrimination, enabling subtle neural circuit modifications to have large behavioral effects (Seeholzer et al., *Nature*, 2018). Looking at an expanded set of *Drosophila* species, the evidence in this paper further places P1 as key neural substrates

targeted by evolution, emphasizing their coordinated action with peripheral sensory systems. Therefore, the findings presented in this manuscript mark an important advancement in our understanding of the evolution of mate discrimination.

The paper is very well written, and the conclusions are generally supported by the data. However, I'd like to raise several points for consideration, presented in no specific order:

We thank the reviewer for their enthusiasm of our work and their appreciation of the power of using a broader comparative approach to define the neural substrates of evolutionary variation. The reviewer has raised important points, that we address below, and believe have further strengthened the manuscript.

Major:

1. Figure 3b shows that *D. yakuba* males display shorter bout lengths in the light compared with the dark (Figure 3d, f), which is surprising. Looking into the methods, the authors detail that a food substrate was used for dark courtship assays but not light ones and hence, males show shorter bout lengths in the light condition. Is the presence of the food absolutely required for *D. yakuba* males to court in the dark? This should be acknowledged in the main text. In addition, poor courtship bout lengths in wildtype flies might mask effects from disrupted pheromonal signaling in *ppk23/ppk25* mutants (Figure 3b-f). Finally, is there a potentially confounding effect introduced from adding food in the dark condition? Different species may respond differently to the presence of a food substrate. The presence of food is known to affect female receptivity and male courtship in *D. melanogaster* (Gorter et al., sci rep, 2016, Grosjean et al., Nature, 2011) as well as change the potentiation of male pheromones (Das et al., PNAS, 2017).

We thank the reviewer for raising these important questions. *D. melanogaster* males do indeed court in the dark in the absence of food (Boll & Noll, 2002; Lin et al., 2005; Toda et al., 2012), as do *D. yakuba* males. However, these assays are typically conducted in small chambers (5-10 mm diameter) where a male can never stray far from the female. This set-up makes it difficult to observe differences in pheromone-dependent tracking, presumably because the male continuously reencounters the female by chance. In contrast, our assays are performed in larger 35 mm chambers, where finding and tracking a female represents a significant sensory challenge for males. As the reviewer points out, females are less receptive in the absence of food and spend more time exploring the chamber. In the presence of food, female locomotion is slower, facilitating the ability of males to maintain pursuit of them based on pheromonal feedback. Including food in our assay therefore enables us to focus on the behavioral role that pheromones play in regulating male arousal and courtship pursuit. We believe that the increased locomotion of females in the absence of food accounts for the observed differences in bout courtship length in the light (where females are not on food) versus the dark (where females are on food). We therefore did not perform courtship in the light on food because, as the reviewer notes, female receptivity is strongly enhanced by the presence of food, such that copulation proceeds quickly, precluding detailed analysis of courtship dynamics prior to mating.

Importantly and relevant to the conclusions of our study, all assays in the dark were carried out under identical conditions on food, allowing us to capture the significant impact of sensory mutants in **Fig. 3**. Moreover, given that species-specific courtship strategies are maintained even when pairing males with females of different species in the dark and on food (**Fig. 1c**), we feel confident that the presence of food is not confounding our behavioral interpretation of the role that pheromone detection via *Ppk23* and *Ppk25* play in shaping mate preferences across species. Finally, we have modified the way we plot bout lengths for all assays, using a log scale to better capture the total range of the data across conditions, while also highlighting significant differences between wild type and mutant animals that were previously artificially compressed on a linear scale.

2. The authors mention in the methods that due to video quality only a small subset of the original data could undergo automated analysis. Removal of these data points has given a small n for behavioral experiments. For instance, there seems to be variability between behavior of the 6 courting pairs in Fig 1b. Hence, increasing the n of flies will allow for stronger conclusions.

We appreciate this suggestion and have adjusted our recording setup for higher resolution recordings, which allowed us to double the size of the tracking experiments in Fig. 1 (n=12 for each of 6 species). These new data support our original conclusion with added confidence: species with sexually dimorphic cuticular pheromones, such as *D. melanogaster* and *D. erecta*, vigorously court their conspecific females in the dark, while only a subset of species (*D. ananassae* and *D. yakuba*) producing monomorphic pheromones do (**Fig. 1b, Ext. Data Fig. 1a**), likely due to their ability to use pheromones as instructive cues to promote courtship in the absence of visual feedback.

3. There are instances of discrepancies between trends seen in automated analyses in the main figures and manual analyses provided in the extended data figures. For example, the authors state that *D. yakuba ppk23* mutant males show reduced pursuit of conspecifics in the dark (Figure 3d). However, extended data figure 3c looking at % time courting contradicts this. Similarly, in Figure 3f authors state *D. yakuba ppk25* mutant males display diminished courtship towards conspecifics in the dark; however, extended data figure 4b % time courting contradicts this. This should be either acknowledged in the text or resolved. We thank the reviewers for noting this. We believe that in assays performed in the dark, bout length is a more sensitive readout of pheromone-dependent courtship as it reflects a male's ability to maintain pursuit based on continued excitatory or inhibitory pheromonal feedback. Once a female is out of reach of the male and 'lost' in the dark, males must search for her based on more distal, non-cuticular pheromone cues. For this reason, we consistently used bout duration throughout the main figures. However, we felt it was important to include additional metrics in the extended data for completeness and transparency. As the reviewer notes, there is a consistent downward trend in the percentage of time *ppk23* and *ppk25* mutant males spent courting observed, but these differences were not statistically significant. We believe that this apparent discrepancy reflects the fact that there is variability in the time it takes males to reencounter a female once contact is lost. Consistent with this notion, *ppk23* and *ppk25* mutant *D. yakuba* males engage in a higher number of courtship bouts than wildtype males (**Reviewer Figure 1**), suggesting they partially compensate for the loss of tracking by reencountering females using alternative search strategies. We have revised the text to acknowledge the differences in these metrics: "*Despite their abbreviated courtship bouts, the total amount of time that ppk23 mutant males spent courting was not significantly reduced, likely because they were often able to rapidly reencounter their female (Extended Data Fig. 3b,d).*" (lines 223-225)

Reviewer Fig. 1. *D. yakuba ppk23* (a) and *ppk25* (b) mutant males show an increase in the number of courtship bouts in the dark toward *D. yakuba* females relative to wildtype males.

4. For Figure 4e, the authors show that responses in the *Dsx* subpopulation are lost in the *ppk23* mutant background for *D. yakuba*. This experiment should be repeated in a *ppk25* mutant background since *ppk23+ppk25+* neurons were identified by the authors as sensing 7-t as an arousal signal (Figure 3).

We agree with the reviewer and have now carried out this experiment which was challenging due to the need to make recombinants in *D. yakuba* in the absence of the balancer chromosomes commonly used in *D. melanogaster*. This experiment, shown in the new Fig. 4e, reveals that the *Dsx+* P1 neurons are insensitive to 7-T in *ppk25* mutant *D. yakuba* males, consistent with our model that that the rapid diversification of *Ppk25+* peripheral sensory neurons underlies the distinct pheromonal tuning of the *Dsx+* P1 neuron population. The previous imaging performed in *ppk23* mutants has been moved to **Extended Data Fig. 3g**.

5. In Figure 4a it is shown that the *Fru* intersection in *D. yakuba* does not elicit reliable wing extension on optogenetic activation. This contrasts with what is seen in *D. melanogaster* (Extended data Figure 6b). Is it possible that non-targeted neurons crucial for eliciting courtship are missing? Since these data are important for supporting the proposed model, could the authors provide movies for the courtship behavior of the *Fru* intersection.

We thank the reviewer for pointing this out and now include representative movies of the courtship behaviors elicited by optogenetic activation of the *Fru*∩P1 and *Dsx*∩P1 subsets. Additionally, given the important implications of differences in the motor displays elicited by the two P1 subtypes in *D. yakuba*, we have carried out additional experiments to explore this point further. First, we optimized our optogenetic protocol for each P1 population (as detailed in the updated methods section: lines 1202-1221) by performing a titration of the optogenetic illumination intensity and selecting the light levels that evoked the highest levels of overall courtship. As observed in **Reviewer Figure 2** (below), at the highest light intensities, the amount of courtship evoked in *Fru*∩P1>CsChrimson males is strongly attenuated suggesting the possibility that off-target effects were likely present in the previous stimulation regime we used in our initial submission, confounding our behavioral analyses. In our new dataset, optogenetic activation of *Dsx*∩P1 and *Fru*∩P1 subtypes in *D. yakuba* drives comparable behaviors (**Fig. 4a, b; Ext. Data Fig. 7; Supplementary Videos 4, 5**), analogous to what we observed in *D. melanogaster*, supporting the basic behavioral redundancy of these populations in promoting courtship.

Reviewer Fig. 2. Manually scored average courtship bout length of FruNP1>CsChrimson (a) or DsxNP1>CsChrimson (b) *D. yakuba* males toward a *D. melanogaster* female target during 2 minutes of optogenetic stimulation across a range of light intensities.

6. Throughout the manuscript, measurements used to represent male courtship behaviors are inconsistent between experiments e.g., average bout length (Figure 2 and 3), % time courting, pursuing, and extending wing (Figure 4). It is understood that perhaps this change in behavioral quantification is due to the authors being interested in either sustained courtship (bout length) or total courtship levels (% courtship). However, the inconsistency in behavioral parameters is at times difficult to follow and sometimes even contradictory to the authors claims (see previous point 3). More explanation of behavioral parameters chosen, and more consistency could be beneficial.

The reviewer is correct that we used different courtship metrics in distinct experimental contexts to capture differences in the persistence of courtship bouts or total courtship levels. However, we appreciate that this inconsistency might create confusion. As we note above, we believe that bout length is the most sensitive indicator of pheromone-dependent courtship in assays conducted in the dark, effectively capturing a male's ability to persistently pursue a female, guided by ongoing pheromonal feedback. Consequently, we now use bout length in all figures. The only exceptions to using this metric throughout our study are: (i) in the initial analysis of courtship in the dark (**Fig. 1**), where we plotted inter-fly distance as a readout of courtship bouts and compared these to manually scored courtship metrics (**Ext. Data Fig. 1**) before selecting blinded, manually scored bout length as the most accurate; (ii) in the optogenetic analysis of P1 subtypes in *D. melanogaster* and *D. yakuba*, where it was important to provide a more granular description of courtship dynamics (**Fig. 4a,b**; **Ext. Data Fig. 6a,b**; **Ext. Data Fig. 7**), and thus employ behavioral classifiers to quantify courtship pursuit and song; and (iii) in the extended data, where in addition to bout length, we also plot additional common measurements of courtship (e.g., latency to court, total % time courting) to provide a comprehensive view (**Extended Data Fig. 3b-d**; **Extended Data Fig. 4b**; **Extended Data Fig. 8a,c,d,f,g**). For consistency, we now also detail how activation of P1 neuron subtypes influences courtship bout length (**Fig. 4a,b**; **Ext Data Fig. 6a,b**). This is all now detailed in the **Courtship Quantification** section of the Methods.

7. For Figure 3e the authors need to directly compare the wildtype *D. yakuba* male results (top panel) to the *ppk23* and *ppk25* mutants (bottom 2 panels) to make statistical comparisons.

We agree and have added a statistical comparison between the wildtype and mutant conditions to the supplement (**Ext. Data Fig. 4c**).

8. The authors should discuss/integrate the model proposed in this manuscript with previous work (Seeholzer et al., *Nature*, 2018, Kallman, Kim and Scott, *eLife*, 2015). For instance, could vAB3/mAL neurons differentially target the subpopulations of P1, changing overall balance/P1 activation?

As noted above in response to Reviewer 1, we attempted to gain genetic access to the second and third order pheromone processing neurons in *D. yakuba*, including mAL and vAB3. Unfortunately, none of the driver lines we generated had sufficiently strong expression to perform functional imaging or confidently define their pheromonal tuning. Nevertheless, we agree with the Reviewer that it is important to integrate our results in *D. yakuba* males with models for how pheromone processing circuits have evolved in other species. We have now expanded on the potential roles of these two pathways in the Discussion, in particular noting that, as the Reviewer suggests, changes in how vAB3 neurons connect to different P1 subsets could underlie the observed differences in their pheromone tuning: *“While the structural or functional changes underlying the divergent patterns of P1 integration remain to be elucidated, the homologous ascending pathways that transmit pheromone signals from the foreleg afferents to the P1 neurons are anatomically identifiable across species^{17,25} (Extended Data Fig. 10). In D. melanogaster males, vAB3 neurons receive input from Ppk25+ sensory neurons responsive to 7,11-HD and relay these excitatory signals to P1 neurons to promote courtship towards a conspecific female^{25,26}. The switch in pheromone sensitivity of Ppk25+ neurons in D. yakuba suggests that vAB3 neurons may selectively convey 7-T-mediated excitation solely to the Dsx+ P1 population, indicating that minor changes to an otherwise conserved circuit architecture could give rise to divergent P1 pheromone tuning. Such subtle alterations in how ascending pheromone pathways are integrated by P1 neurons would resemble the changes proposed to underlie the opposite behavioral response of D. melanogaster and D. simulans males to 7,11-HD¹⁷. ”* (lines 438-449)

Minor:

9. P1 is functionally heterogenous and has been defined differently in the literature (e.g. P1, P1a, pC1, etc.) creating confusion (issue described in Asahina et al., *Curr Opin Physiol*, 2018). A better description of how the pC1/P1 populations are named and defined should be included (Figure 4). In the main text there is only reference to ‘P1’ however, pC1 is mentioned in the extended data figures. The nomenclature should be more clearly explained, for instance describing the Dsx+ pC1 population which overlaps with the Fru+ P1 cluster at the beginning of the section related to Figure 4

We thank the reviewer for highlighting this point and appreciate the need for clarity, especially given that the delineation of P1 neurons has been inconsistent in the past, as highlighted by Asahina et al., *Curr Opin Physiol*, 2018 and more recently by Jiang and Pan, *Neurosci. Bull* (2022). For example, P1 neurons have been variously defined as:

1. “The P1 cluster (fru P1), composed of approximately 25 neurons that coexpress fru and dsx.” (Kimura et al., 2008; Koganezawa et al., 2016).
2. “P1 cells are male-specific, FruM interneurons” labeled by “intersection between R15A01-AD and R71G01-DBD (referred to as ‘P1a’)” (Hoopfer et al., 2015).
3. “P1a, a subset of pC1 neurons and pC2 cell types” (Roemschied et al, 2023).
4. “A cluster of up to 60 sexually dimorphic neurons located at the posterior medial part of the male *Drosophila* brain, collectively referred to as ‘P1’ or ‘pC1’” (Ishii et al., 2020).
5. “Dsx-positive pC1 neural cluster and its fru-expressing subset called P1 neurons” (Takayanagi-Kiya et al., 2023).
6. “All P1 a neurons express DsxM , but only some express FruM, thus not all these neurons are P1, but all are pC1, according to the initial terminology” (Jiang and Pan 2022)

As Jiang and Pan (2022) described: “The current use of P1/pC1 terms is somewhat arbitrary.” The complexities in the nomenclature of P1 neurons in the literature reflect the fact that, as the reviewer notes, this population is heterogenous and different genetic methods label distinct subsets of the total population (Asahina, 2018; Jiang and Pan 2022), all of which appear sufficient to promote a male’s arousal and courtship.

In this study, we focused on the relevant functional, anatomic and behavioral differences revealed by intersecting a widely-used P1 neuron driver (71G01) with the endogenous *dsx* and *fru* promoters, highlighting the role these two master regulatory transcription factors play in demarcating functionally distinct subsets of the P1 population. To enhance clarity, we now include a diagram (**Ext. Data Fig. 5a**) that illustrates the relationship between the *Dsx+* and *Fru+* neuronal classes and relates them to the nomenclature used in the literature. Our diagram shows that $Fru \cap P1$ and $Dsx \cap P1$ neurons represent partially overlapping subsets within the broader *Dsx+* pC1 cluster. In the text, we also only refer to these two populations by the intersectional strategies we used to label them ($Fru \cap P1$ and $Dsx \cap P1$), making a clear distinction as to how they are genetically defined in our study. We also recognized that the previous reference to pC1 neurons in **Ext. Data Fig. 5b-d** was confusing and now refer only P1 neurons, aligned with our definition of these neurons in **Ext. Data Fig. 5a**.

10. The authors use an optogenetic stimulation protocol of 2 mins dim white light followed by 2 mins red light (627nm), alternated over 10 minutes. However, P1 activation can be sustained for minutes after stimulation (Hoopfer et al., eLife, 2015). Raster plots in extended data Figure 7 show that, except between the first ‘light off’ to ‘light on’ period, there seems not to be much difference between light on and off periods. In addition, it appears that more males exhibited wing extension after the light stim period in Figure 4a. The authors should consider doing separate experiments for “light on” and “light off” conditions to avoid confounding effects.

By optimizing the light intensity used for optogenetic activation described above (see point 5), we believe that we have effectively addressed this issue. With the more consistent behavioral dynamics elicited with the new activation protocol, we now observe distinct differences between the light-on and light-off periods, with a significant return towards baseline during the “light off” periods as the arousal state wanes (**Fig. 4a,b; Ext. Data Fig. 6a,b; Ext. Data Fig. 7**). These results reinforce our conclusion that activation of different P1 neuron subtypes elicits similar behaviors within and across species.

11. The authors have tested for normality in order to apply the appropriate statistical tests however, when non-parametric tests such as Mann Whitney were applied the mean and SEM were plotted. For non-normal data it is more appropriate to use the median.

We thank the reviewer for pointing out this oversight. We have corrected this throughout the manuscript, ensuring that all data that is not normally distributed now appropriately indicate the median.

12. Courtship assays in Figure 1b were recorded for a total of 10 minutes. Was any consideration given to the fact that courtship/copulation latencies vary between different species (Khallaf et al., Nature communications, 2021).

As the reviewer notes, different *Drosophila* species display variation in the latency to initiate courtship and copulation, which prompted us to conduct our in-the-dark assays over the course of three hours to be sure we captured this variation. Although in the absence of visual cues, the latency of a male to encounter a female in our 35mm diameter assay chamber was highly variable (likely due to the variable delays in initially encountering the female), we observed no significant differences in the latencies to initiate courtship between species or among the sensory mutants we tested. We have now explicitly noted this in the **Courtship assays in the dark** section of the methods.

13. The IFD traces in Fig 1b, showing male proximity to the female over the 10-minute courtship assay, are for a single representative pair only. To represent the whole data set, a population trace taking into account all 6 pairs may be more informative.

We thank the reviewer for their suggestion. We plotted the data as suggested (**Reviewer Fig. 3b**) but found that the variability in the timing of courtship behaviors and the wide spread of IFDs that arise when males are transiently not courting made the plot difficult to interpret. However, we have now expanded our dataset to 12 courtship pairs per species, included the heatmaps for all 12 pairs, and plot a histogram of IFD over the trial as a summary statistic, to provide the reader with an intuitive understanding of the differences in courtship dynamics across species. For comparison, here is the behavior of wildtype *D. yakuba* in the dark:

Reviewer Fig. 3. (a) Inter-fly distance (IFD) heatmaps for 12 *D. yakuba* male-female pairs aligned to courtship initiation in the dark as in Fig. 1b. (b) Individual IFD traces of the same pairs in a overlaid (left) or the mean trace of all pairs (right). Horizontal dotted line indicates 8mm threshold.

14. For main and extended data Figure 1, IFD<8mm is stated as the threshold used for close proximity. However, no details of how this was decided upon is included. Was this chosen using previous studies or was this the distance threshold that fitted best with the manually scored data (extended data fig 1)?

Indeed, as suggested by the reviewer, we selected this threshold because it best matched our manual scoring of courtship. Our aim was to optimize the balance between eliminating false negatives (i.e., the flies are still courting but the female temporarily moves too far away from the male), and reducing false positives (i.e., the flies are nearby but not actually courting). In the methods (lines 1096-1098) and Extended Data Fig. 1, we have now expanded our explanation of using the 8 mm threshold. The update figure legend now includes: “*Comparison of manually confirmed courtship bouts with those estimated by IFD thresholding revealed that an IFD threshold of 8mm most accurately replicated manual scoring, resulting in the lowest incidence of false positives and negatives.*” Importantly, while IFD is not a perfect representation of courtship, it still captures the species-specific variation in courtship dynamics in the dark observed across the 72 pairs represented in **Fig 1**. Notably, in all other figures, where precise quantification of and comparison of courtship metrics was required, we used other methods that do not require any assumptions about IFD.

15. Since *oe-* flies are still able to produce pheromones (e.g., some methyl pheromones, Dweck et al., PNAS, 2015) the authors should add a wildtype female *D. yakuba* control to Figure 1d to account for any contributions of non-ablated pheromones.

We agree that this is an important comparison and have updated the scale of **Fig. 1d** to enable direct comparisons between the courtship rate of *D. yakuba* males towards wild-type *D. yakuba* and *oe-* females displayed in the adjacent panel (**Fig. 1c,d**). Additionally, it is important to note that the key comparison we make in **Fig. 1d** is between *oe-* target flies perfumed with solvent or with 7-T, allowing us to control for contributions of non-ablated pheromones.

16. Is it possible to know for Figure 3a how many neurons make up the *ppk23+*/*ppk25+* and *ppk23+*/*ppk25-* subpopulations in *D. yakuba*? Extended data Figure 3d provides total *ppk23* neurons only. Unfortunately, despite several attempts, we have not been successful in integrating a Gal4 driver into the *ppk25* locus, possibly due to its likely disruption of another gene, *CheB42c*, which has an intron into which *ppk25* is embedded. We also attempted to stain this population using FISH, but the uneven penetration of our probes through the cuticle made it challenging to accurately count cell types with this method. Given the overall conservation observed in the Ppk23+ sensory population, we hypothesize that the Ppk25+ population is similarly conserved and likely represents about half of the Ppk23+ neurons. Notably, in the revised manuscript we now have expanded our functional analysis of *ppk25* mutants to compare the pheromone responses of sensory neurons and P1 neurons, both of which indicate that the Ppk25+ population is sensitive to 7-T and propagates this signal to the P1 neurons (**new Fig. 4e**) to promote male arousal. These data underscore that Ppk25+ sensory neurons, independent of the size of the population, are essential for the detection of 7-T in *D. yakuba* males.

17. The authors may wish to acknowledge in the text (line 328/9) that it has been shown previously that male *D. melanogaster* court *D. erecta* females (Billeter et al., *Nature*, 2009).

We thank the reviewer for their suggestion and agree. We have now cited this previous insight in the results section (lines 337-339).

18. In both Figure 3e and Figure 4b it is shown that *D. yakuba* males respond to *D. simulans* females due to the presence of 7-T. Therefore, could the authors discuss how male *D. yakuba* avoid or minimize courtship towards heterogenous females that also carry 7-T. For instance, how does this fit with the authors' current model of *D. yakuba*'s use of 7-T to direct courtship to conspecifics?

In accord with the reviewer's suggestion, we now demonstrate in **Fig. 1c** that both *D. yakuba* and *D. simulans* males court females of the other species as vigorously as their conspecifics, likely due to their shared pheromone, 7-T. This apparent promiscuity makes sense, considering that *D. simulans* and *D. yakuba* are believed to have evolved in geographic isolation from one another, on opposite sides of the African continent (Lachaise & Silvain, 2004; Dean & Ballard, 2004). Additionally, in the rare locales where species producing 7-T overlap, males do court females from both species, while females mediate pre-mating discrimination (Coyne et al., 2005).

19. Comments on Figure presentation:

-Figure 3a: the right panel is not referred to in the text.

We have now referenced this panel in the text (line 235).

- Figure 3h: (i) The authors should add *ppk23* and *ppk25* into the schematic (ii) The grey and white shaded boxes are misleading as these are previously used to represent dark and light conditions (iii) It would be helpful if the authors could make clear on the model schematic what is previous work and what is new data added from this manuscript, including the relevant references in the figure legend (iv) Finally, some of the model may be oversimplified and therefore slightly misleading, for example from the schematic it appears as though 7,11-HD directly inhibits P1 in *D. simulans* males which according to previous work from the same group isn't the case (Seeholzer et al., *Nature*, 2018).

We appreciate these suggested points of clarification and have made the following modifications to this schematic: (i) we added *ppk23* and *ppk25* to the diagram to denote the sensory populations involved; (ii) modified the color scheme to prevent confusion about whether this refers to differences in courtship in the light or dark; (iii) we added the relevant references to the figure legend to clarify what was previously known, and (iv) revised the 7,11-HD pathway in *D. simulans* to reflect our current understanding that it

does indeed inhibit P1 neurons (Seeholzer et al, *Nature*, 2018). While it is challenging to encapsulate all the data of prior studies within a single diagram, we aimed to summarize the key observations as they relate to our discoveries in *D. yakuba*. In support of our depiction of the *D. yakuba* circuitry in Fig. 3h, we also provide new data to show that 7,11-HD is sufficient to inhibit the P1 responses typically evoked by a conspecific *D. yakuba* female (**Ext Data Fig. 8e**).

- Figure 3b,c,d,f: the scale of the y-axis (up to 600 seconds), compresses all the data at the bottom of the axis. This makes it very difficult to see the spread of data and differences between data sets.

We have now addressed this important point by plotting all bout lengths on a log scale. This adjustment allows us to effectively display the full range of the data more effectively while still revealing the relevant differences.

- Figure 4i: it would be helpful to have the mean bars in a different color to black, so they are easier to see.

We have changed the figure to make the median more visible.

- Extended data Figure 8: it would be helpful to label the relevant species on the phylogeny circle.

We have modified this figure such that all species are labeled.

Referee #3 (Remarks to the Author):

In their manuscript entitled “A modular circuit architecture coordinates the diversification of courtship strategies in *Drosophila*”, Coleman et al. make use of pheromone mating circuits in multiple *Drosophila* species to investigate evolutionary mechanisms leading to behavioral diversification. They nicely show that males of the *Drosophila* species *D. yakuba*, with a monomorphic pheromone profile, use this pheromone as excitatory mating signal. Peripheral pheromone sensing cells, which lead to aversion in *D. melanogaster* with a dimorphic pheromone profile, switched to induce attractive responses in *D. yakuba*. In addition, they show that functional subdivision in central brain circuits lead to specific responses in Dsx+ cells in *D. yakuba* triggering a mating response. Overall, they provide evidence that tuning changes at the periphery and modified central processing shape evolution of mating behaviors. Conceptually, they propose that the modular organization at each layer facilitates adaptation and behavioral evolution.

This work nicely illustrates how the transfer of neurogenetic tools from *D. melanogaster* to other species with derived behaviors can deliver functional insights into basic principles of neural circuit evolution. I am impressed by the wealth of data in this manuscript, the degree of experimental details in a non-standard model system and the sophisticated experimental approaches.

I consider the results presented here of immediate interest to a broad audience working in (behavioral) neuroscience and evolution and I think it merits publication in *Nature*. The described evolutionary changes at the periphery and in central brain circuits between species represent a fascinating example of how input-output relationships can be modified and will serve as reference point for future studies in other systems. The work nicely builds on a previous, seminal study from the same lab (Seeholzer, *Nature* 2018) and by smartly choosing selected model species, it identifies novel ways (gain of peripheral excitation, selective activation of P1 subclusters) how sensory system and its central processing centers can evolve.

The manuscript is very well written and follows a nice flow and narrative. The presented results support the conclusions drawn by the authors and the statistical evaluation is convincing. However, there are a few points that I would like the authors to address prior to publication:

We appreciate the reviewer's support for our work and their recognition that integrative circuit nodes, like the P1 neurons, can serve as facile substrates for evolution, allowing for rapid diversification of input-output relationships. We address the reviewer's thoughtful criticism below.

Major:

Figure 1: It is not clear to me where in the manuscript IFD-tracking and automatic scoring of bout lengths was performed and where courtship behavior was evaluated manually. As shown in Ext. Data Figure 1b, automatic scoring performs very poorly in scoring average bout length. How data for each figure was processed should be made clearer e.g. in the figure legends.

In this line, I am wondering if data shown in Figure 3b-f was manually scored? Could the authors show the other parameters measured (in addition to bout length) for Figure 3b? It is surprising to me that *D. yakuba* *ppk23* mutants do not show a phenotype in light. Does this indicate an additional role for visual information? In *D. melanogaster*, *ppk23* mutants reduce mating in light (e.g. <https://journals.plos.org/plosgenetics/article?id=10.1371/journal.pgen.1002587>).

We agree that it is important clarify how behavioral quantifications were performed and now include a section in the Methods called **Courtship Quantification** describing how data was processed (lines 1091-1119). Notably, we only used IFD as a proxy for courtship in **Fig. 1** since it captured species-specific courtship dynamics in a high-throughput manner, facilitating a broader comparison. In all other instances, courtship behaviors were quantified via blind manual scoring or via semi-supervised approaches, as indicated in the figure legends. Moreover, we have included additional courtship metrics to **Ext. Data Fig. 3** (percent time courting and courtship latency) to provide a more comprehensive evaluation of the effects (or lack thereof) of the *ppk23* mutation on courtship in the light in *D. yakuba*. It is worth noting that, as the reviewer points out, courtship deficits for *D. melanogaster* *Δppk23* mutants in the light have been reported, this phenotype has not been consistently observed (Toda et al, 2012). While we are unsure of the source of these varied results (e.g. different chamber size, strain background), the in the dark phenotype appears robust and has been reported across studies. However, acknowledging that *D. melanogaster* *ppk23* mutants have been previously reported to have courtship deficits in the light, we now more explicitly emphasize the previously documented redundancy between vision and pheromones in the text: "*D. yakuba* *ppk23* mutant males vigorously pursued their conspecific females in the light, likely due to the redundant role that conspecific pheromones and visual cues play in promoting courtship (**Fig. 3b**; **Extended Data Fig. 3c**)."

One direct consequence of the proposed model would be that *D. yakuba* shows an increased degree of male-male courtship as there is no strong cVA response in the *ppk23* population and 7T is an attractive signal. Do the authors observe this? What could prevent this type of interaction? Related: Line 365: "In *D. yakuba* and other monomorphic species, the conserved male-specific volatile pheromone cis-Vaccenyl acetate (cVA) likely performs this role as it has been to prevent intrasexual courtship, likely through direct inhibitory pathways that impinge on P1 neurons." But it does not activate *ppk23* neurons in *Dyak* (Extended Data Fig. 3). How?

We have introduced new experiments to address the interesting question of how *D. yakuba* males perform sex discrimination given that the excitatory pheromone 7-T is carried by both males and females. First, we examined the responses of both *Ppk23+* sensory neurons and both P1 subpopulations to male targets (new **Fig. 3e** and **Fig. 4a,b**). These functional data show that while sensory neurons are activated by male pheromones, P1 neurons are not. This suggests that P1 neurons responses to 7-T are suppressed by other male cues, likely cVA, that counters the 7-T-induced excitation evoked at the periphery by both males and females.

To further explore this, we compared sex-discrimination in *D. yakuba* and *D. melanogaster* males (**Ext. Data Fig. 8**; discussed lines 350-369). Aligned with the reviewer's suggestion, *D. yakuba* males do indeed spend more time courting males than *D. melanogaster* males do. However, *D. yakuba* males still display a preference to court females (**Ext. Data Fig. 8a**), underscoring that they can use additional cues for sex discrimination. As noted above, we reasoned that cVA, as a male-specific volatile pheromone, might play a key role in sex discrimination in *D. yakuba* males by countering 7-T excitation. We therefore perfumed *D. yakuba* females with cVA and found that this led to both attenuated pheromone responses in Dsx+ P1 neurons and reduced courtship towards these female targets (**Ext. Data Fig. 8d,e**). To link the suppression of P1 neuron activity and male arousal to the detection of cVA, we generated a mutant of the canonical cVA receptor Or67d. Unexpectedly, *Or67d* mutant males displayed no apparent phenotype and still rarely courted *D. yakuba* males, suggesting Or67d either acts redundantly with other receptors (e.g., Or65a; Naters & Carlson, 2007) or has evolved a distinct role in this species (**Ext. Data Fig. 8b,c**). Thus while the peripheral sensory receptors detecting cVA in *D. yakuba* remain to be defined, our combined functional imaging and behavioral data now demonstrate that this male-specific volatile pheromone serves to counter 7-T-excitation in P1 neurons and reduce intrasexual courtship.

Figure 3f: Why was the the *Dyak-Dmel* experiment here performed in light conditions (for *ppk25* mutants)? It would be more consistent to perform these experiments in the dark to compare directly to the neighboring data.

The goal of **Fig. 3f** was to assess whether *D. yakuba* males remain averse to courting heterospecific females in the absence of the Ppk25 signaling pathway. We therefore wished to maximize the arousal of males, allowing us to reveal whether the *D. melanogaster* female pheromone is sufficient to suppress visual pursuit. Our findings indicate that, unlike *ppk23* mutant males examined under the same conditions (**Fig. 3b**), *ppk25* mutants remain unwilling to court *D. melanogaster* females (**Fig. 3f**). In addition, it may be helpful to note that we have never observed a context in which a male would court a female in the dark that it would not court in the light. Thus, testing the role of *ppk25* in suppressing heterospecific courtship in the light serves as the most stringent test of this receptor's function in mate recognition.

Figure 3 and Ext. Data Fig 5: Could the authors please indicate when data was re-plotted across panels? We thank the reviewer for pointing out this important omission. We have now updated the figure legends of **Ext. Data Fig. 3 & 4**, which we believe are the figures that the reviewer referred to. These updated legends explain that we have replotted the bout lengths across experimental data shown in the main figure, but here also include courtship latency and percent time courting as a point of comparison, to provide a more complete quantification of courtship parameters.

Figure 2: Left: Here the authors just show a small overview staining of the respective brain labels in multiple species. Could they locate the LPC and indicate where the region of interest was placed for quantification of their calcium imaging data? Also, I would suggest adding a few words in the material and methods on how the quantification of Calcium imaging data was done (it is apparent that signals are stronger in Dmel than in the other species probably due to variation in expression levels of transgenes?!). We have now indicated the region of interest used for functional imaging in **Fig. 2** and added a more detailed description to how calcium signals were normalized in the methods. As the reviewer notes, expression levels of the same transgenes vary across species, with often weaker expression observed in *D. yakuba* males compared to *D. melanogaster* males, which may contribute to the generally higher responses observed in the latter. However, it is worth noting that across all experiments, our interpretations rely on the relative response patterns within a species rather than any direct comparison of absolute responses between species.

Extended Data Fig. 5b-e: Here it is not clear which neurons are counted. Could the authors provide representative pictures of their Dsx and Fru stainings to facilitate recapitulation of their data for the reader.

We have now added representative images of the Fru and Dsx antibody labeling of P1 soma that we used to quantify and characterize the P1 subtypes (**Ext. Data Fig. 5b-e**).

Extended Data Fig. 7: Here, the *D. yakuba* data does not look very convincing to me. Could the authors clarify which pre and post-stimulation periods were taken into account for quantification. To me it seems there is a strong initial arousal response in the first activation window but activity remains high in the following OFF phase.

The data presented in **Figure 4** compares courtship behaviors in the first stimulation period, i.e. naïve flies during the 2 min period prior to stimulation, during the 2 min period of optogenetic stimulation with pulsed red light, and in the subsequent 2 min period post stimulation. The continued courtship males display following stimulation is consistent with previous observations that P1 neurons evoke an enduring arousal state (e.g. Hoopfer et al., eLife, 2015). However, as noted above in response to comments by Reviewers 1 and 2, we have now collected a new, larger dataset using an optogenetic protocol which we optimized for each P1 subtype in *D. yakuba* by performing a titration of the optogenetic illumination intensity and selecting the light levels that evoked the highest levels of overall courtship (**Reviewer Figure 2**). With this improved protocol, we better reveal light-evoked pursuit and singing that remains high after stimulation in accord with prior observations in *D. melanogaster*.

Reviewer Fig. 2. Manually scored average courtship bout length of FruNP1>CsChrimson (a) or DsxNP1>CsChrimson (b) *D. yakuba* males toward a *D. melanogaster* female target during 2 minutes of optogenetic stimulation across a range of light intensities.

Figure 4e: I was wondering why the authors did not use the ppk25+ mutants for their experiment to more directly support their claim (does the chromosomal location of ppk25 prevent this experiment)? If it is technically possible, could the authors show dsx>GCaMP responses in ppk25 mutants? Related: Line 348: “Exemplifying this evolutionary flexibility, we show that in *D. yakuba* males, one subset

of P1 neurons uniquely integrates from the Ppk25+ peripheral sensory neurons with enhanced sensitivity to 7-T. “. This is not really shown.

We agree with the reviewer and have now carried out this experiment despite the associated technical challenges in which we had to generate recombinants in *D. yakuba* in the absence of the balancer chromosomes commonly used in *D. melanogaster*. This experiment, shown in the new **Fig. 4e**, reveals that the Dsx+ P1 neurons are insensitive to 7-T in *ppk25* mutant *D. yakuba* males, consistent with our model that the rapid diversification of Ppk25+ peripheral sensory neurons underlies the distinct pheromonal tuning of the Dsx+ P1 neuron population. The previous results showing the impact of the *ppk23* mutation on pheromone responses have been moved to **Ext. Data Fig. 3e**.

Minor:

Figure 1 and Extended Data Figure 1: Please add information that the red line indicates courtship initiation. Could you clarify in the figure legend what is the time frame analyzed in b? Experiments were run for 3h -> the heatmap shows a 10 min time window, the histogram the full 3h?

Thank you for catching these oversights. We have updated the figure legend to clarify that the dotted line represents the initiation of courtship, and that the experimental window encompasses the 10 min following the first courtship event. Due to the assays being conducted in the dark in relatively large chambers, we observed considerable variation in the latency to courtship initiation among individual flies. Consequently, we used an extended recording period of 3 hours to capture all potential courtship events, as detailed in the methods.

Extended Data Fig 1 b,d: It is not clear what is the sample size in the experiments and what the complete sample refers to.

We have modified both the figure and the figure legend to clarify the exact nature of the sample size.

Line 166/167: “Minor differences in the arborization patterns of P1 neurons were apparent across species. Yet despite their morphological variation,...”. Given that different genetic reagents were used in each species (e.g. GCaMP vs. CD8-GFP, split- reagents vs. single driver), I am not sure if one can conclude morphological variation rather than technical reasons. I would suggest removing of this phrase.

This point is well taken and we have removed this phrase.

Figure 2b: Is this the average tap-evoked response of traces shown to the left or of all experiments (shown at the right)?

The average tap-evoked responses represent the complete data set across biological replicates (as shown at the right). We have now clarified this in the figure legend.

Please update Reference 45.

I believe the reviewer is referring to the fact that this BioRxiv paper (Reference #45) has now been published in Current Biology, which we also cite. However, the P1 driver line relevant to citation 45 was actually only included in the BioRxiv preprint and not included in the published Current Biology paper, hence why we cited the preprint here.

Reviewer Reports on the First Revision:

Referees' comments:

Referee #1 (Remarks to the Author):

In their revised manuscript, Coleman et al. have gone to great lengths in addressing the questions I raised in my critiques of the original manuscript, which I highly appreciate. The nice study has been further improved by the additional experiments, insights, and clearer presentations. I enthusiastically support the publication of this paper in Nature.

One minor point: in response to one of my critiques, the authors use the nomenclature of Fru_AND_P1 and Dsx_AND_P1 to designate the two populations of P1 neurons even though the latter contains the former (which they now clearly show in the new Extended Data Fig. 5a and state explicitly). This is fine for the most parts but in a few cases, this leads to some awkwardness or inaccuracy. For example, in lines 327-328, the authors state that "The molecular subdivision of P1 neurons by their differential expression of the Fru and Dsx transcription factors..." Since Dsx is expressed in all P1 neurons according to Extended Data Fig. 5a, they are not differentially expressed in different subdivision of P1 neurons, so Dsx should be removed from the above sentence (differential expression of Fru distinguishes the two subpopulations). The authors should go through the text and figure/legend closely to avoid such inaccuracy.

Referee #2 (Remarks to the Author):

The authors have addressed all concerns and questions I raised. I am satisfied with their revisions and am pleased to recommend the publication of this paper.

Referee #3 (Remarks to the Author):

The authors addressed all my comments and concerns. Congratulations on this great work.

Author Rebuttals to First Revision:

Referee #1

In their revised manuscript, Coleman et al. have gone to great lengths in addressing the questions I raised in my critiques of the original manuscript, which I highly appreciate. The nice study has been further improved by the additional experiments, insights, and clearer presentations. I enthusiastically support the publication of this paper in *Nature*.

One minor point: in response to one of my critiques, the authors use the nomenclature of Fru_AND_P1 and Dsx_AND_P1 to designate the two populations of P1 neurons even though the latter contains the former (which they now clearly show in the new Extended Data Fig. 5a and state explicitly). This is fine for the most parts but in a few cases, this leads to some awkwardness or inaccuracy. For example, in lines 327-328, the authors state that “The molecular subdivision of P1 neurons by their differential expression of the Fru and Dsx transcription factors...” Since Dsx is expressed in all P1 neurons according to Extended Data Fig. 5a, they are not differentially expressed in different subdivision of P1 neurons, so Dsx should be removed from the above sentence (differential expression of Fru distinguishes the two subpopulations). The authors should go through the text and figure/legend closely to avoid such inaccuracy.

We thank the referee for their insightful comments during the review process.

We appreciate the reviewer’s point and fully agree that describing P1 neuron subsets by “their differential expression of the Fru and Dsx transcription factors” is confusing, given that all P1 neuron classes are marked by Dsx. We have modified the text accordingly in both the results (lines 233-235 and lines 271-272) and in the discussion (lines 369-370) so that we only refer to Fru as being differentially expressed across P1 neuron subsets.

Referee #2:

The authors have addressed all concerns and questions I raised. I am satisfied with their revisions and am pleased to recommend the publication of this paper.

Referee #3:

The authors addressed all my comments and concerns. Congratulations on this great work.